# Relaxed targeting rules help PIWI proteins silence transposons

Ildar Gainetdinov[1✉], Joel Vega-Badillo[1], Katharine Cecchini[1], Ayca Bagci[1], Cansu Colpan[1,3], Dipayan De[2], Shannon Bailey[1], Amena Arif[1,4], Pei-Hsuan Wu[1,5], Ian J. MacRae[2] & Phillip D. Zamore[1✉]

In eukaryotes, small RNA guides, such as small interfering RNAs and microRNAs, direct AGO-clade Argonaute proteins to regulate gene expression and defend the genome against external threats. Only animals make a second clade of Argonaute proteins: PIWI proteins. PIWI proteins use PIWI-interacting RNAs (piRNAs) to repress complementary transposon transcripts[1,2]. In theory, transposons could evade silencing through target site mutations that reduce piRNA complementarity. Here we report that, unlike AGO proteins, PIWI proteins efficiently cleave transcripts that are only partially paired to their piRNA guides. Examination of target binding and cleavage by mouse and sponge PIWI proteins revealed that PIWI slicing tolerates mismatches to any target nucleotide, including those flanking the scissile phosphate. Even canonical seed pairing is dispensable for PIWI binding or cleavage, unlike plant and animal AGOs, which require uninterrupted target pairing from the seed to the nucleotides past the scissile bond[3,4]. PIWI proteins are therefore better equipped than AGO proteins to target newly acquired or rapidly diverging endogenous transposons without recourse to new small RNA guides. Conversely, the minimum requirements for PIWI slicing are sufficient to avoid inadvertent silencing of host RNAs. Our results demonstrate the biological advantage of PIWI over AGO proteins in defending the genome against transposons and suggest an explanation for why the piRNA pathway was retained in animal evolution.

In prokaryotes and eukaryotes, small RNA or DNA guides direct Argonaute proteins to fight viruses, plasmids and transposons[5], regulate gene expression[6–12] or aid DNA replication[5]. Animals produce two distinct types of Argonaute protein: AGO and PIWI. AGO-clade proteins use small interfering RNAs (siRNAs; which are typically 21 nucleotides long) or microRNAs (miRNAs; which are most often 22 nucleotides long) to repress extensively or partially complementary transcripts[6]. AGO proteins initially find their targets through complementarity to a short 5′ region of their guide, the seed (nucleotides g2–g8; Fig. 1a). For miRNA-guided AGO proteins, seed complementarity is sufficient to repress the target RNA[6]. By contrast, PIWI-clade Argonaute proteins use piRNAs (which are 18–35 nucleotides long) as guides[1,2]. Although most eukaryotic genomes encode one or more AGO protein, only animals make PIWI proteins. With few exceptions, all animals use piRNAs to repress transposons[1,2]. The ancestral mechanism of piRNA-guided transposon silencing is PIWI-catalysed endonucleolytic cleavage (slicing) of complementary transposon RNAs in the cytoplasm[13–16]. Moreover, piRNA production itself requires piRNA-directed slicing of piRNA precursor transcripts[13,14,17–21]. In some animals, piRNAs also direct nuclear PIWI proteins to nascent transposon transcripts to silence transcription through repressive histone marks or DNA methylation[22–26].

siRNAs direct AGO proteins and piRNAs direct PIWI proteins to hydrolyse the phosphodiester bond that joins the target nucleotides opposite guide nucleotides g10 and g11 (that is, t10 and t11, respectively; Fig. 1a). Unlike siRNA-directed, AGO-catalysed transcript cleavage, efficient target slicing by PIWI proteins requires the auxiliary factor GTSF1 (ref. 27). GTSF1 accelerates the otherwise slow target cleavage by PIWIs by 10–100-fold, probably by stabilizing the catalytically competent conformation of PIWI proteins. Why PIWI slicing evolved to require an auxiliary protein is unknown.

Here we report that, compared with AGO proteins, the requirements for guide:target complementarity are relaxed for the PIWI proteins MILI (also known as PIWIL2) and MIWI (also known as PIWIL1) from mouse (*Mus musculus*) and Piwi from freshwater sponge (*Ephydatia fluviatilis*; hereafter denoted *Ef*Piwi). PIWI proteins bind RNAs both with and without complementarity to the canonical 5′ seed of their guide. Both in vitro and in vivo, PIWI-catalysed slicing requires at least 15 contiguously paired nucleotides, and longer extents of complementarity tolerate guide:target mismatches at essentially any position. Unlike AGO proteins, guide pairing to any target nucleotide, including those that flank the scissile phosphate (t10 and t11), is dispensable for efficient slicing. Although pairing to at least four piRNA 5′ terminal nucleotides facilitates target finding, in vitro and in vivo abundant

[1]RNA Therapeutics Institute and Howard Hughes Medical Institute, University of Massachusetts Chan Medical School, Worcester, MA, USA. [2]Department of Integrative Structural and Computational Biology, The Scripps Research Institute, La Jolla, CA, USA. [3]Present address: Voyager Therapeutics, Cambridge, MA, USA. [4]Present address: Beam Therapeutics, Cambridge, MA, USA. [5]Present address: University of Geneva, Geneva, Switzerland. ✉e-mail: ildar.gainetdinov@umassmed.edu; phillip.zamore@umassmed.edu

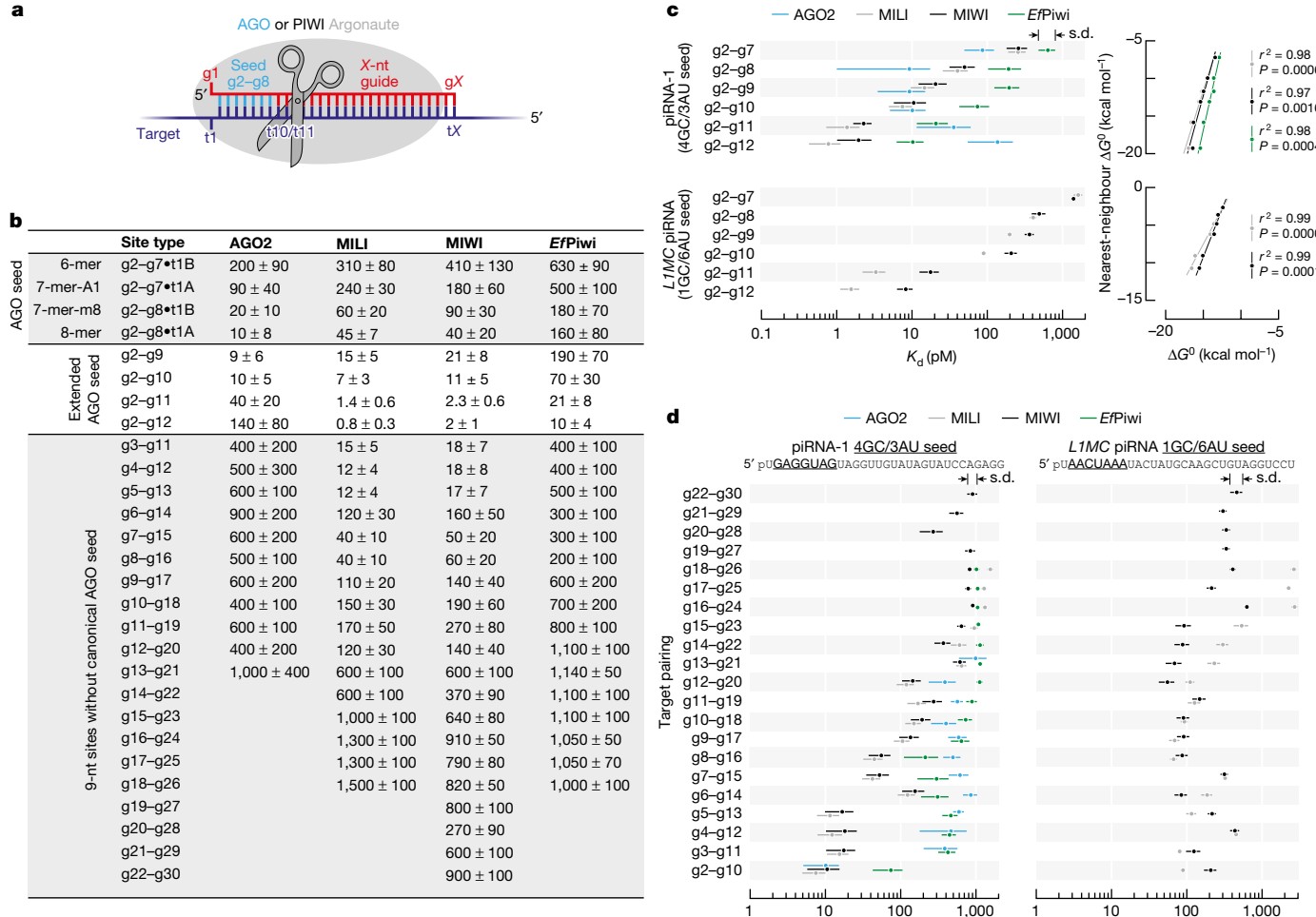

**Fig. 1 | PIWI proteins bind sites containing or lacking canonical seed pairing.** **a**, Small RNA guides direct eukaryotic Argonaute proteins to complementary targets. nt, nucleotide. **b**, Binding affinities ($K_d$ in pM) of MIWI, MILI, *Ef*Piwi and mouse AGO2 loaded with piRNA-1 for canonical and non-canonical target sites. **c**, Left, MIWI, MILI, *Ef*Piwi and mouse AGO2 binding affinities for targets contiguously paired from nucleotide g2. Right, relationship between binding energy $\Delta G^0$ calculated from $K_d$ (mean of three independent trials) and predicted binding energy $\Delta G^0$. Goodness-of-fit for linear regression ($r^2$) and P value for two-tailed permutation test for Pearson's correlation are shown. All data are in Supplementary Fig. 2a. **d**, MIWI, MILI, *Ef*Piwi and mouse AGO2 binding affinities for nine-nucleotide complementary stretches contiguously paired from all guide nucleotides. All data are in Supplementary Fig. 2b. Mean and standard deviation of data from three independent trials are shown (**b**,**c** (left), **d**).

Table b:

| | Site type | AGO2 | MILI | MIWI | *Ef*Piwi |
|---|---|---|---|---|---|
| **AGO seed** 6-mer | g2–g7•t1B | 200 ± 90 | 310 ± 80 | 410 ± 130 | 630 ± 90 |
| 7-mer-A1 | g2–g7•t1A | 90 ± 40 | 240 ± 30 | 180 ± 60 | 500 ± 100 |
| 7-mer-m8 | g2–g8•t1B | 20 ± 10 | 60 ± 20 | 90 ± 30 | 180 ± 70 |
| 8-mer | g2–g8•t1A | 10 ± 8 | 45 ± 7 | 40 ± 20 | 160 ± 80 |
| **Extended AGO seed** | g2–g9 | 9 ± 6 | 15 ± 5 | 21 ± 8 | 190 ± 70 |
| | g2–g10 | 10 ± 5 | 7 ± 3 | 11 ± 5 | 70 ± 30 |
| | g2–g11 | 40 ± 20 | 1.4 ± 0.6 | 2.3 ± 0.6 | 21 ± 8 |
| | g2–g12 | 140 ± 80 | 0.8 ± 0.3 | 2 ± 1 | 10 ± 4 |
| **9-nt sites without canonical AGO seed** | g3–g11 | 400 ± 200 | 15 ± 5 | 18 ± 7 | 400 ± 100 |
| | g4–g12 | 500 ± 300 | 12 ± 4 | 18 ± 8 | 400 ± 100 |
| | g5–g13 | 600 ± 100 | 12 ± 4 | 17 ± 7 | 500 ± 100 |
| | g6–g14 | 900 ± 200 | 120 ± 30 | 160 ± 50 | 300 ± 100 |
| | g7–g15 | 600 ± 200 | 40 ± 10 | 50 ± 20 | 300 ± 100 |
| | g8–g16 | 500 ± 100 | 40 ± 10 | 60 ± 20 | 200 ± 100 |
| | g9–g17 | 600 ± 200 | 110 ± 20 | 140 ± 40 | 600 ± 200 |
| | g10–g18 | 400 ± 100 | 150 ± 30 | 190 ± 60 | 700 ± 200 |
| | g11–g19 | 600 ± 100 | 170 ± 50 | 270 ± 80 | 800 ± 100 |
| | g12–g20 | 400 ± 200 | 120 ± 30 | 140 ± 40 | 1,100 ± 100 |
| | g13–g21 | 1,000 ± 400 | 600 ± 100 | 600 ± 100 | 1,140 ± 50 |
| | g14–g22 | | 600 ± 100 | 370 ± 90 | 1,100 ± 100 |
| | g15–g23 | | 1,000 ± 100 | 640 ± 80 | 1,100 ± 100 |
| | g16–g24 | | 1,300 ± 100 | 910 ± 50 | 1,050 ± 50 |
| | g17–g25 | | 1,300 ± 100 | 790 ± 60 | 1,050 ± 70 |
| | g18–g26 | | 1,500 ± 100 | 820 ± 50 | 1,000 ± 100 |
| | g19–g27 | | | 800 ± 100 | |
| | g20–g28 | | | 270 ± 90 | |
| | g21–g29 | | | 600 ± 100 | |
| | g22–g30 | | | 900 ± 100 | |

piRNAs direct slicing of targets that lack 5′ complementarity. Notably, the minimum 15-nucleotide stretch of complementarity that licenses piRNA-guided target cleavage is sufficient to distinguish host transcripts from transposon RNAs. These findings suggest that the catalytic properties of PIWI proteins evolved to prevent transposons from escaping piRNA silencing through mutation while simultaneously retaining sufficient specificity to spare self-transcripts from inappropriate repression.

## PIWI proteins bind without seed pairing

Mouse piRNAs guide MILI and MIWI to slice extensively complementary transposon transcripts[15,16,28]. piRNAs have also been proposed to direct MIWI to bind and regulate mRNA expression through the same mechanism by which miRNAs guide AGO proteins to their targets[29,30]. In this miRNA-like binding mode, base pairing to the canonical AGO seed sequence (guide nucleotides g2–g8; Fig. 1a) mediates the search for complementary sites and is sufficient to tether the Argonaute protein to its target RNA. We used the RNA Bind-'n-Seq method[31] to measure the affinity of piRNA-guided PIWI Argonaute proteins for a library of 20-nucleotide-long random sequences (Extended Data Fig. 1a). We incubated the target RNA library with purified mouse MILI or MIWI or freshwater sponge *Ef*Piwi loaded with a 5′ monophosphorylated synthetic RNA (26 nucleotides for MILI and *Ef*Piwi, 30 nucleotides for MIWI; Extended Data Fig. 1b–d) and isolated and sequenced RNAs bound to the piRNA–PIWI protein complex (piRISC). The sequencing data were analysed using an approach that estimates the affinity ($K_d$) of piRISC for each binding-site type[32].

Binding of MILI, MIWI or *Ef*Piwi to RNAs with canonical AGO seed sites was weaker than for mammalian AGO2 proteins (Fig. 1b,c and Extended Data Fig. 2a). Compared with AGO proteins, PIWI protein affinity was 4–16-fold lower for an 8-mer (g2–g8•t1A), the canonical site type most effectively repressed by miRNA-guided AGO proteins[6]. In detail, for piRNA-1, $K_d^{AGO2, 8-mer} = 10 \pm 8$ pM compared with $K_d^{MILI, 8-mer} = 45 \pm 7$ pM, $K_d^{MIWI, 8-mer} = 40 \pm 20$ pM and $K_d^{EfPiwi, 8-mer} = 160 \pm 80$ pM (Fig. 1b). The intracellular concentration of the most abundant piRNAs (19 nM) is less that of than the most abundant miRNAs in mouse primary spermatocytes (24 nM)[21], which suggests that the weaker affinity of PIWI proteins for the canonical seed sites will result in lower occupancy of such targets. Our data therefore disfavour a model in which PIWI

proteins find and productively regulate targets through seven-nucleotide canonical seed pairing.

For AGO proteins, extending pairing beyond the canonical seed did not increase the affinity of RISC for a target. For piRNA-1, $K_d^{AGO2,g2-g10} = 10 \pm 5$ pM compared with $K_d^{AGO2,8-mer} = 10 \pm 8$ pM (Fig. 1b,c and Extended Data Fig. 2a). By contrast, extending guide:target complementarity from g2–g8 to g2–g10 increased MILI and MIWI target affinity by 4–6-fold. For piRNA-1, $K_d^{MILI,g2-g10} = 7 \pm 3$ pM compared with $K_d^{MILI,8-mer} = 45 \pm 7$ pM, and $K_d^{MIWI,g2-g10} = 11 \pm 5$ pM compared with $K_d^{MIWI,8-mer} = 40 \pm 20$ pM (Fig. 1b,c). MILI and MIWI therefore bound to g2–g10 complementary sites with an affinity indistinguishable from that of mouse AGO2 for 8-mer sites (Fig. 1b,c). Compared to MILI and MIWI, *Ef*Piwi required longer guide:target pairing (g2–g12) to achieve the affinity of AGO2 for a canonical 8-mer seed site, with $K_d^{EfPiwi,g2-g12} = 10 \pm 4$ pM (Fig. 1b,c). For both a piRNA of synthetic sequence (piRNA-1) and a piRNA found in vivo (*L1MC* piRNA, antisense to the mouse L1MC retrotransposon), the binding affinity of PIWI proteins increased linearly with increasing predicted base pairing energy ($\Delta G^0$; Fig. 1c). Thus, MILI, MIWI and *Ef*Piwi[33] require a longer extent of guide:target base pairing to approach the affinity of AGO2 RISC for a canonical seed match.

Unlike AGO proteins, PIWI proteins bound with similar affinity to sites both with and without full pairing to the canonical seed (Fig. 1b,d and Extended Data Fig. 2b). For instance, MILI and MIWI bound to targets with nine-nucleotide uninterrupted complementarity to the guide starting at g2 or g5 with nearly indistinguishable affinities. For piRNA-1, $K_d^{MILI,g2-g10} = 7 \pm 3$ pM compared with $K_d^{MILI,g5-g13} = 12 \pm 4$ pM, and $K_d^{MIWI,g2-g10} = 11 \pm 5$ pM compared with $K_d^{MIWI,g5-g13} = 17 \pm 7$ pM (Fig. 1b,d). Even when guide:target pairing did not start until g8, MILI and MIWI binding was only fivefold weaker compared with targets paired from g2. For piRNA-1, $K_d^{MIWI,g2-g10} = 11 \pm 5$ pM compared with $K_d^{MIWI,g8-g16} = 60 \pm 20$ pM (Fig. 1b,d). These RNA Bind-'n-Seq data agreed well with direct measurement of binding affinity for individual RNA targets. For piRNA-1, $K_d^{MIWI,g2-g10} = 14 \pm 9$ pM compared with $K_d^{MIWI,g8-g16} = 80 \pm 20$ pM (Extended Data Fig. 2c).

*Ef*Piwi also bound to nine-nucleotide sites starting at g6, g7 or g8 only 3–4-fold less tightly than when complementarity started at g2. For piRNA-1, $K_d^{EfPiwi,g2-g10} = 70 \pm 30$ pM compared with $K_d^{EfPiwi,g8-g16} = 200 \pm 100$ pM (Fig. 1b,d). By contrast, AGO2 bound sites lacking seed pairing 10–100-fold more weakly than those containing a seed match (Fig. 1b,d and Extended Data Fig. 2b). For piRNA-1, $K_d^{AGO2,g2-g10} = 10 \pm 5$ pM compared with $K_d^{AGO2,g8-g16} = 500 \pm 100$ pM. Compared to AGO2, *Ef*Piwi had 3–15-fold higher affinity for longer complementary sites (≥11 nucleotides), with pairing starting at g2, g3, g4, g5 and g6 (Extended Data Fig. 2d). We conclude that PIWI proteins are more flexible than AGO proteins in the types of sites they can bind but require longer complementarity for high-affinity binding.

## Slicing of partially complementary RNA

The modal length of piRNAs is 5–10 nucleotides longer than that of siRNAs (26–31 nucleotides compared with 21 nucleotides)[1], yet MILI, MIWI and *Ef*Piwi do not require pairing to these additional 3′ nucleotides to cleave a target RNA[15,27,33]. In vitro, 16–23-nucleotide-long contiguous complementarity was sufficient for MILI, MIWI and *Ef*Piwi to reach their maximum endonuclease rate (Extended Data Fig. 3a). Moreover, MILI and MIWI, directed by 21-nucleotide-long guides, cleave targets as efficiently as when loaded with full-length 26-nucleotide or 30-nucleotide piRNAs[27]. Nonetheless, we found that extending pairing beyond piRNA nucleotide g20 enabled MILI and MIWI to tolerate guide:target mismatches.

We used the high-throughput Cleave-'n-Seq approach[3] to determine the rates of cleavage for thousands of target variants (Extended Data Fig. 3b). We incubated purified MILI, MIWI or *Ef*Piwi piRISC complexes (1 nM) and the PIWI auxiliary factor GTSF1 (500 nM) with a library containing 7,700–10,400 30-nucleotide-long target RNAs for different lengths of time (60 s to 16 h). Uncleaved RNAs were reverse-transcribed and sequenced, and their abundance at each time point was used to determine their pre-steady-state cleavage rate, $k$ (Extended Data Fig. 3b and Supplementary Table 1).

Consistent with the idea that additional complementarity to piRNA 3′ nucleotides accelerates cleavage by MILI or MIWI of imperfectly paired targets, a mismatch between g2 and g20 decreased the median $k$ by 3.6-fold when all nucleotides after g20 were unpaired, but by only 1.4-fold when g21–g25 were also base paired (Fig. 2a and Extended Data Fig. 3c). Two mismatches between g2 and g20 caused a 30-fold median reduction in $k$ for targets with no pairing beyond g20, but a reduction of only 3.4-fold when g21–g25 were paired (Fig. 2a and Extended Data Fig. 3c). Thus, endonucleolytic cleavage by MILI or MIWI does not require target pairing to piRNA 3′ sequences, but such extended complementarity readily compensates for guide:target mismatches. We did not observe the same compensatory effect for *Ef*Piwi. Compared to MILI and MIWI, slicing by *Ef*Piwi was generally slower (Extended Data Fig. 3a), probably because Gtsf1 from *Ephydatia muelleri* (*Em*Gtsf1) was used to stimulate *Ef*Piwi-catalysed target cleavage (Methods).

AGO-clade Argonaute proteins secure target nucleotide t1 in a pocket that often displays specific nucleotide preferences. Although MILI, MIWI and *Ef*Piwi showed a slight binding preference for t1A and t1U (≤3-fold stronger affinity compared with t1S; Extended Data Fig. 3d), target cleavage showed no t1 preference (Extended Data Fig. 3a).

## PIWI proteins tolerate mismatch at any position

Efficient target RNA cleavage by animal and plant AGO proteins requires uninterrupted base pairing to siRNA nucleotides g9–g13 (refs. 3,4,34). By contrast, MILI, MIWI and *Ef*Piwi slicing tolerated mismatches at any position within the region of complementarity (Fig. 2b,c, Extended Data Fig. 4 and Supplementary Figs. 3 and 4).

For g2–g21-paired targets of AGO2, a single mononucleotide mismatch at g9, g10, g11 or g13 decreased the median $k$ by 10–200-fold[3] (Fig. 2b and Extended Data Fig. 4). For the same extent of pairing, the median reduction in $k$ was ≤5-fold for MILI and MIWI and ≤7-fold for *Ef*Piwi for a mononucleotide mismatch at any position between g2 and g20 (Fig. 2b and Extended Data Fig. 4). For MILI and MIWI, guide:target complementarity from g2 to g25 reduced the median effect of a mononucleotide mismatch to ≤3-fold at any position between g2 and g20 (Fig. 2b and Extended Data Fig. 4). Among mismatch types, GU wobbles had the smallest impact on endonucleolytic rate, decreasing $k$ by 1.2-fold (inter-quartile range (IQR) of 1–1.65; Extended Data Fig. 5a).

Unlike AGO2 RISC, pairing to target nucleotides adjacent to the scissile phosphate was dispensable for target slicing by piRISC (Fig. 2b,c, Extended Data Fig. 4 and Supplementary Figs. 3 and 4). Loss of pairing to either t10 or t11 in a g2–g21 match decreased the median cleavage rate by 4-fold for MILI and MIWI (IQR of 2.5–19 for t10, and IQR of 2.8–5.7 for t11), and by 5–6.4-fold for *Ef*Piwi (IQR of 4–18.9 for t10, and IQR of 3.2–5.3 for t11; Fig. 2b and Extended Data Fig. 4). Purine–purine mismatches at these positions appeared to be the least tolerated by piRISC, perhaps because their greater bulk is poorly accommodated in the PIWI catalytic centre (Extended Data Fig. 5b). piRISC-catalysed slicing was detectable even when both t10 and t11 were unpaired. The median decrease in $k$ was 60-fold for MILI and MIWI (IQR of 40–70) and 67-fold for *Ef*Piwi (IQR of 54–88; Fig. 2b and Extended Data Fig. 4). For the same t10–t11 dinucleotide mismatch, target cleavage was undetectable with AGO2 RISC (Fig. 2b and Extended Data Fig. 4). We conclude that MILI, MIWI and *Ef*Piwi, unlike AGO2, can efficiently cleave partially paired RNAs with mismatches anywhere in a target site.

Sequencing the 3′ products of piRNA-directed, MIWI-catalysed slicing showed that piRISC invariably hydrolysed the RNA at the canonical

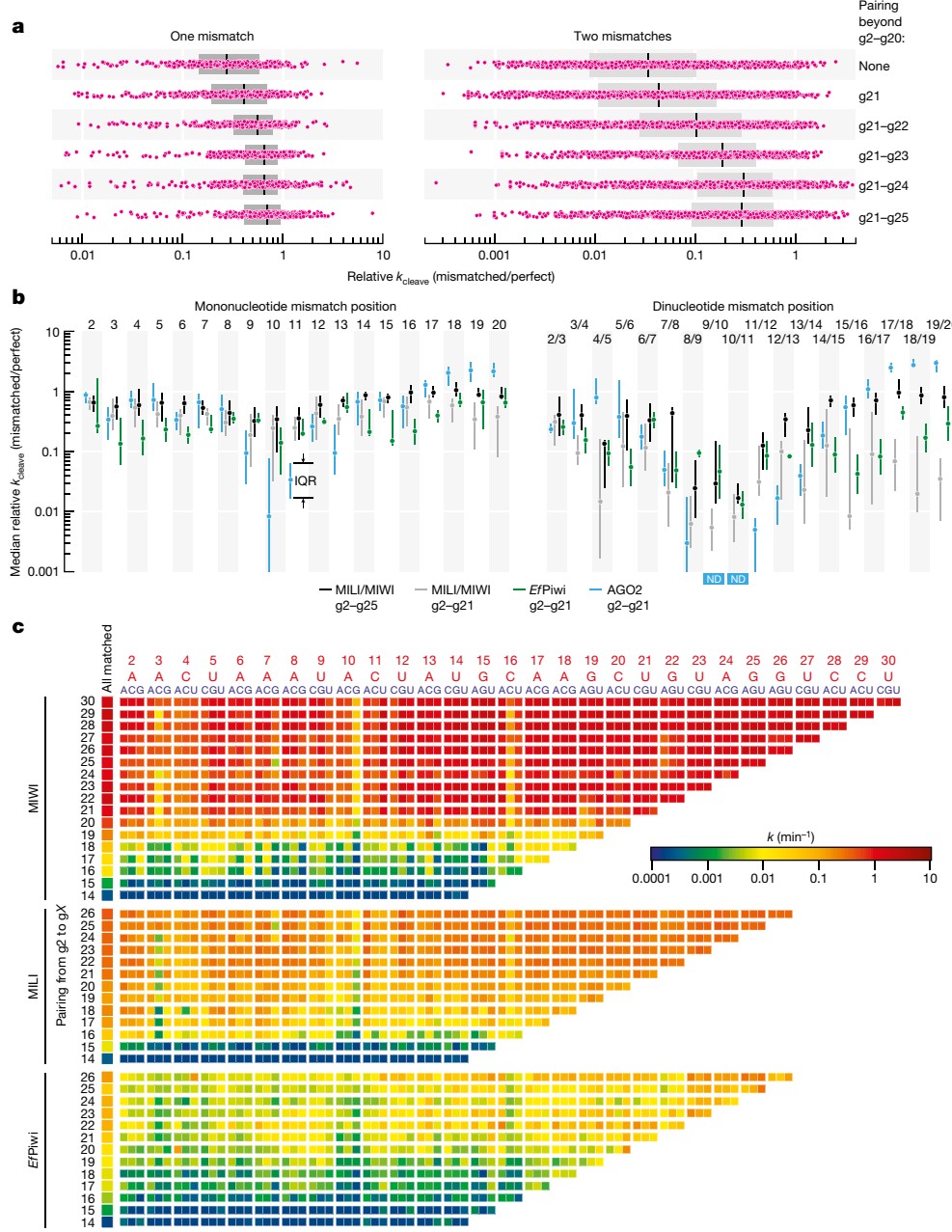

**Fig. 2 | PIWI slicing tolerates mismatches with any target nucleotide.**
**a**, Change in pre-steady-state cleavage rate for one or two mismatches (pink) between g2 and g20. For one mismatch, *n* = 456: all 19 possible positions × 3 geometries × 4 piRNAs × MILI and MIWI. For two mismatches, *n* = 1,368: all 171 possible combinations × 1 geometry × 4 piRNAs × MILI and MIWI. Box plots show the IQR and median. Statistical analyses are in Extended Data Fig. 3c. **b**, Change in pre-steady-state cleavage rate for one or two consecutive mismatches between g2 and g20 for contiguous g2–g21 or g2–g25 pairing for MILI, MIWI, *Ef*Piwi and mouse AGO2. Median and IQR are shown. For one mismatch, *n* = 24 (3 geometries × 4 piRNAs × MILI and MIWI); *n* = 6 for *Ef*Piwi

(3 geometries × 2 piRNAs); *n* = 21 for AGO2 (3 geometries for *L1MC* guide and 3 geometries × 3 contexts for let-7a and miR-21 guides). For two consecutive mismatches, *n* = 8 (1 geometry × 4 piRNAs × MILI and MIWI); *n* = 2 for *Ef*Piwi (1 geometry × 2 piRNAs); *n* = 19 for AGO2 (1 geometry for *L1MC* RISC and 9 geometries for let-7a and miR-21 RISCs). All data and statistical analyses are in Extended Data Fig. 4. ND, not detected. **c**, MIWI, MILI and *Ef*Piwi pre-steady-state cleavage rates (*k*) for targets of *L1MC* piRNA containing a single unpaired nucleotide. Position and identity of mononucleotide mismatch in targets (indicated in blue) of L1MC piRNA (indicated in red) are on the top of the chart.

scissile phosphodiester bond, between target nucleotides t10 and t11, even when both g10 and g11 were unpaired or when contiguous pairing did not start until g11 (Extended Data Fig. 5c). These data suggest that piRNA–target base pairing near the cleavage site has little if any role in positioning the scissile phosphate within the MIWI catalytic centre.

## Mismatch tolerance is intrinsic to PIWI proteins

Unlike AGO-clade Argonaute proteins, PIWI proteins require the auxiliary factor GTSF1 to achieve their maximal catalytic rate[27] (Extended Data Fig. 6a). Without GTSF1, PIWI Argonaute proteins inefficiently slice even perfectly complementary RNAs[27].

Although GTSF1 potentiates target cleavage by PIWI proteins, it is not required for PIWI tolerance of guide:target mismatches. GTSF1 accelerated slicing of fully and partially complementary RNAs to a similar extent (Extended Data Fig. 6b). In the presence of GTSF1, the median increase in MILI cleavage rate was 25-fold (95% confidence interval (CI) of [19, 29]) for perfectly complementary RNAs and 17–50-fold for mismatched targets. GTSF1 enhanced MIWI-catalysed slicing of fully (median increase, 14-fold; CI of [12.8, 15.4]) and partially (11–35-fold) complementary targets with similar efficacy. *Em*Gtsf1 also accelerated *Ef*Piwi slicing for perfectly paired (15-fold, CI of [8, 28]) and mismatched RNAs (4–19-fold) to a comparable degree (Extended Data Fig. 6b). These data suggest that, unlike AGOs, PIWI inherently accommodates unpaired target nucleotides and that GTSF1 only accelerates cleavage.

## Relaxed rules for slicing apply in vivo

In mice, piRNAs direct MILI and MIWI to slice complementary transposon transcripts, mRNAs or long non-coding transcripts (lncRNAs)[7,8,10,11,15,16]. As we observed for purified piRISC, piRNAs directed MILI and MIWI to cleave targets with as few as 15–19-nucleotide complementary nucleotides in vivo in mouse primary spermatocytes.

Mouse primary spermatocytes produce a class of MILI-loaded and MIWI-loaded piRNAs called pachytene piRNAs, which first appear at the pachytene stage of meiosis[1]. Mouse GTSF1 is present in all meiotic male germ cells[27]. Because endonucleolytic cleavage by Argonaute proteins leaves a 5′-monophosphate[35,36], we sequenced 5′-monophosphorylated RNAs from mouse primary spermatocytes purified by fluorescence activated cell sorting (FACS) to identify potential 3′ cleavage products generated in vivo by MILI and MIWI (Fig. 3a). Restricting our analysis to pachytene piRNAs, >80% of which derive from non-repetitive sequences, ensured unambiguous assignment of piRNAs to candidate cleavage products. To identify those RNAs corresponding to 3′ cleavage products generated by piRNA-directed slicing, we searched for 5′-monophosphate-bearing RNAs present in control C57BL/6 mice, but the abundance of which was reduced by ≥8-fold in a triple mutant lacking all piRNAs from three major pachytene piRNA-producing loci on chromosomes 2, 9 and 17: 2-qE1-35981(+); 9-qC-31469(−),10667(+); 17-qA3.3-27363(−),26735(+) (refs. 10,37). For simplicity, we refer to these loci as *pi2*, *pi9* and *pi17*, respectively. The *pi2⁻/⁻ pi9⁻/⁻ pi17⁻/⁻* triple mutation removes approximately 22% of all pachytene piRNAs (Fig. 3a and Extended Data Fig. 7).

Among the 5′-monophosphorylated RNAs detected in the control C57BL/6 mice, we selected candidate cleavage products for which production could be explained by a *pi2*, *pi9* or *pi17* piRNA directing cleavage between nucleotides t10 and t11. For each pattern of piRNA–target complementarity, for example, g2–g*X* pairing with t2–t*X*, we calculated the fraction of cleavage product candidates for which abundance was reduced by ≥8-fold in *pi2⁻/⁻ pi9⁻/⁻ pi17⁻/⁻* mutant primary spermatocytes. We denote this fraction as $f^{pi2,pi9,pi17}_{\text{decreased in mutant}}$. A pairing configuration that can support target slicing is predicted to have a high fraction of such candidate 3′ cleavage products present in the control mice but reduced or absent in the triple mutant mice (Fig. 3a).

To account for sampling error arising from the short-lived nature of 5′-monophosphorylated fragments in vivo, we identified all 5′-monophosphorylated RNAs in C57BL/6 explained by piRNAs not removed in *pi2⁻/⁻ pi9⁻/⁻ pi17⁻/⁻* mice (control piRNAs), and then calculated the fraction of these RNAs reduced by ≥8-fold in *pi2⁻/⁻ pi9⁻/⁻ pi17⁻/⁻* animals. The fraction of cleaved targets for each pairing arrangement g2–g*X* was then calculated as the observed signal minus the sampling error (Fig. 3a) as follows:

$$f^{\text{g2–g}X}_{\text{cleaved}} = f^{pi2,pi9,pi17}_{\text{decreased in mutant}} - f^{\text{control}}_{\text{decreased in mutant}}$$

Cleavage by MILI or MIWI is indistinguishable in our data, thus $f^{\text{g2–g}X}_{\text{cleaved}}$ corresponds to the sum of targets sliced in mouse primary spermatocytes by both PIWI proteins.

These analyses showed that PIWI-catalysed cleavage was detected in primary spermatocytes for piRNA–target base pairing as short as g2–g16, with median $f^{\text{g2–g16}}_{\text{cleaved}} = 0.18$, CI of [0.08, 0.23] (Fig. 3b, Supplementary Table 2 and Supplementary Data 1). The efficiency of piRNA-directed target cleavage increased with longer complementarity. For example, median $f^{\text{g2–g20}}_{\text{cleaved}} = 0.52$ (CI of [0.37, 0.65]; Fig. 3b and Supplementary Table 2). Note that pairing longer than g2–g20 contained too few data points to measure the corresponding $f^{\text{g2–g}X}_{\text{cleaved}}$.

When the piRNA–target complementarity was ≥17 nucleotides long, single mismatches were tolerated at most positions (Fig. 3c, Extended Data Fig. 8a, Supplementary Table 2 and Supplementary Data 1). For example, for g2–g18 complementarity, the median $f_{\text{cleaved}}$ for perfect pairing (0.18, CI of [0.14, 0.23]) was similar to g2–g18 matches bearing a single nucleotide mismatch at positions g2, g5, g6, g9–g14 or g17–g18 (0.15–0.25; Fig. 3c, Supplementary Table 2 and Supplementary Data 1). For targets complementary to piRNA nucleotides g2–g18 or g2–g19, pairing to t10 and t11, the target nucleotides flanking the scissile phosphate, was dispensable for slicing (Fig. 3c, Supplementary Table 2 and Supplementary Data 1). The lower median $f_{\text{cleaved}}$ values for targets mismatched to piRNA 5′ sequences compared with other piRNA regions may reflect slower on-rates for piRNAs of low intracellular concentration (see also the next section). For g2–g18 targets, the median $f^{\text{mismatches to g3–g8}}_{\text{cleaved}} = 0.10$ compared with $f^{\text{mismatches to g9–g18}}_{\text{cleaved}} = 0.18$, whereas for g2–g19 targets, the median $f^{\text{mismatches to g3–g8}}_{\text{cleaved}} = 0.06$ compared with $f^{\text{mismatches to g9–g19}}_{\text{cleaved}} = 0.21$.

Together, these in vivo data corroborate our in vitro target cleavage experiments, which found that the median decrease in cleavage rate was fourfold or less for a mononucleotide mismatch at any position between g2 and g20 (Fig. 2b and Extended Data Fig. 4). Thus, a wide variety of piRNA–target pairing patterns can efficiently direct MILI and MIWI to cleave targets, unlike the relatively limited pairing configurations tolerated by AGO-clade Argonaute proteins.

## Abundant piRNAs slice without seed match

AGO2-catalysed slicing of sites contiguously paired from g4 or g5 (that is, without canonical seed pairing) has been observed in vitro but not detected in vivo[3,38]. By contrast, our data identified piRNA-directed cleavage in vivo in mouse primary spermatocytes of targets for which pairing to the guide starts at g3, g4 or g5.

In vitro, MILI and MIWI did not require target pairing to piRNA 5′ terminal nucleotides for binding (Fig. 1b,d) or slicing (Fig. 3d and Extended Data Fig. 8b). Similarly, we detected in vivo piRNA-directed cleavage of targets that lack complementarity to nucleotides g2–g4 (Fig. 3e, Supplementary Table 2 and Supplementary Data 1). Pre-organization of seed nucleotides g2–g6 in an A-form-like helix accelerates target finding by AGO proteins (for the let-7a 8-mer, target-finding rate constant $k^{\text{guide alone}}_{\text{on}}$ of $5 \times 10^6$ M⁻¹ s⁻¹ compared with $k^{\text{guide in AGO2}}_{\text{on}}$ of $2.4 \times 10^8$ M⁻¹ s⁻¹)[39]. By contrast, PIWI proteins pre-organize only nucleotides g2–g4 (refs. 33,40,41), which suggests that there is slower target finding when piRNA–target complementarity begins after nucleotide g4. In theory, high piRISC concentration could compensate for a slower $k_{\text{on}}$ value. In vivo, piRNA concentrations vary widely, and about 1,500 piRNAs are present in mouse primary spermatocytes at ≥500 pM (Extended Data Fig. 8c). We observed piRNA-directed cleavage of targets with contiguous 14-nucleotide pairing beginning at g4 (median $f^{\text{g4–g17}}_{\text{cleaved}} = 0.43$, CI of [0.29–0.56]) or g5 (median $f^{\text{g5–g18}}_{\text{cleaved}} = 0.33$, CI of [0.2–0.4]) only for highly abundant piRNAs (≥500 pM). Cleavage of targets paired from g3 was detectable for piRNAs for which the in vivo concentration was ≥100 pM, and targets paired from g2 were sliced by piRNAs present at ≥50 pM (Fig. 3e, Supplementary Table 2 and Supplementary Data 1). These in vivo data were consistent with equilibrium

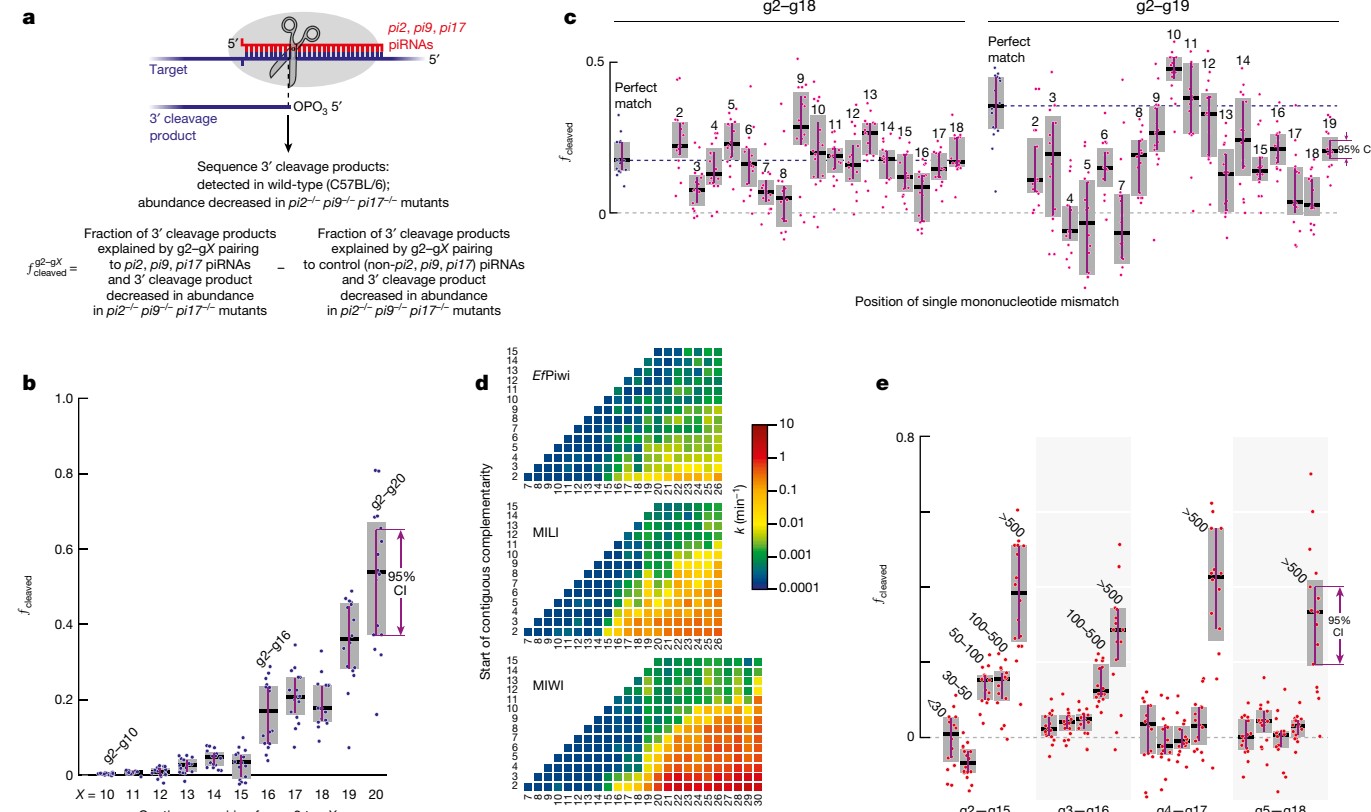

**Fig. 3 | Mouse PIWI proteins cleave partially complementary targets in vivo. a**, Schematic of the strategy used to identify 3′ cleavage products of piRNA-guided PIWI-catalysed slicing and to measure the fraction of targets cleaved by PIWI proteins in FACS-purified mouse primary spermatocytes. **b**, Fraction of cleaved MILI and MIWI targets in FACS-purified mouse primary spermatocytes for contiguous pairing from nucleotide g2. **c**, Fraction of cleaved targets in FACS-purified mouse primary spermatocytes for perfect matches (indicated in blue) and for pairing containing a single-nucleotide mismatch (indicated in pink). Horizontal dotted lines indicate the medians for perfect matches. **d**, MIWI, MILI, and *Ef*Piwi pre-steady-state cleavage rates in vitro for all possible stretches of ≥6-nucleotide contiguous pairing starting from nucleotides g2–g15 of *L1MC* piRNA. **e**, Fraction of cleaved targets in FACS-purified mouse primary spermatocytes for 14-nucleotide contiguous pairing starting from nucleotides g2 to g5. Data are binned by piRNA intracellular concentration (<30, 30–50, 50–100, 100–500, >500 pM). For **b,c** and **e**, box plots show IQR and median; 95% CI was calculated with 10,000 bootstrapping iterations; $n = 16$ permutations of 4 control (C57BL/6) and 4 *pi2[−/−] pi9[−/−] pi17[−/−]* animals.

binding measurements showing that the affinity ($K_d$) of MILI or MIWI for target sites with pairing starting at g2, g3, g4 or g5 was <500 pM (Fig. 1d).

We conclude that because pairing to piRNA 5′ terminal nucleotides is dispensable for both target finding and slicing, piRISC efficiently cleaves targets that lack full complementarity to the canonical 5′ seed.

## Insertions and deletions thwart piRNA-directed slicing

As observed for AGO2 (ref. 3), mononucleotide target insertions between t9 and t15 slowed cleavage catalysed by MILI, MIWI or *Ef*Piwi in vitro by ≥10-fold (Extended Data Fig. 9a). For both AGO2 (refs. 3,42) and PIWI proteins[33,40,41], target nucleotides t9–t15 face the protein surface, which makes insertions likely to distort the catalytic centre.

Single-nucleotide target deletions between t6 and t15 were also poorly tolerated (≥5-fold lower $k$ in vitro for MILI, MIWI and *Ef*Piwi relative to a fully complementary target; Extended Data Fig. 9b). Such target sequence deletions result in mononucleotide bulges in the piRNA guide. Similar to mammalian AGO2 (ref. 42), *Ef*Piwi restricts piRNA nucleotides g6–g10 to the central cleft of the protein[33], and a mononucleotide piRNA bulge between t6 and t10 is unlikely to fit in this narrow furrow, which potentially explains why PIWI proteins

do not tolerate such target deletions. By contrast, single-nucleotide deletions between t11 and t15 create solvent-facing, mononucleotide loops of guide nucleotides g11–g15 that are predicted to be accommodated. Indeed, AGO2 tolerates target deletions between t11 and t15 (ref. 3). Notably, PIWI proteins did not tolerate t11–t15 target deletions (Extended Data Fig. 9b). We speculate that piRNA guide bulges between t11 and t15 specifically affect PIWI proteins because they impair interactions with GTSF1.

Target insertions or deletions near the centre of the piRNA–target duplex were also not tolerated in vivo in mouse primary spermatocytes (Extended Data Fig. 9c, Supplementary Table 2 and Supplementary Data 1). We note that insertions and deletions occur in mammalian genomes 30-fold less frequently than single-nucleotide polymorphisms[43].

## Predicting in vivo target cleavage sites

We used a logistic regression classifier approach to identify factors that predict effective piRNA slicing in vivo. Cleavage data from pachytene spermatocytes were used to fit a logistic function representing the probability of piRNA-guided cleavage, $P$(cleaved), determined by 35 variables ($x_1, x_2, \ldots x_{35}$): the presence or absence of pairing with each guide nucleotide between g2 and g25; the total number of paired nucleotides; the predicted binding energy; the piRNA abundance; the

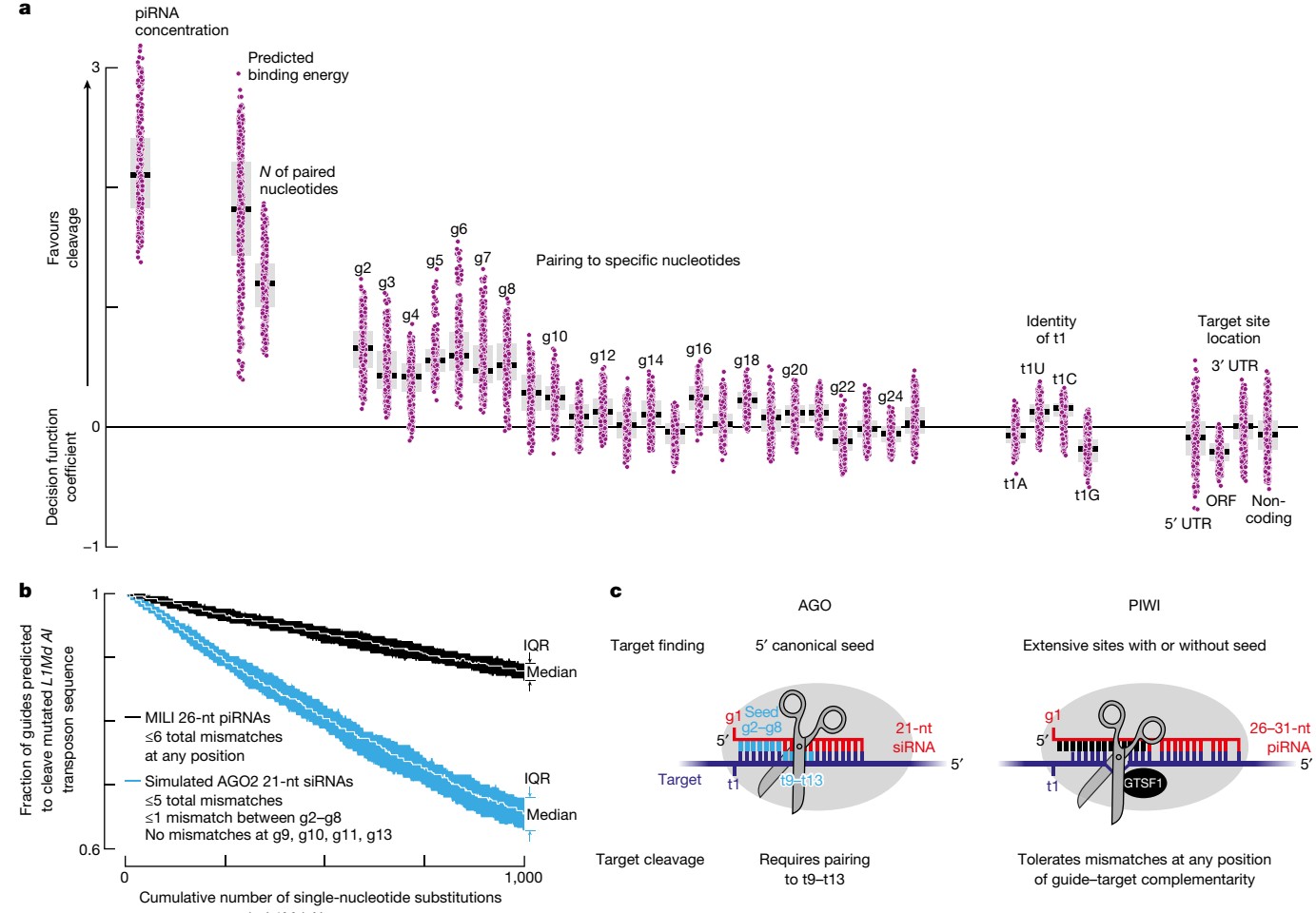

**Fig. 4 | Determinants of PIWI slicing in vivo. a**, Decision function coefficients for 400 logistic function fits (regression models) using around 3,500 distinct piRNA–target pairs detected in mouse primary spermatocytes. *n* = 16 permutations of 4 control (C57BL/6) and 4 *pi2*^−/−^*pi9*^−/−^*pi17*^−/− animals × 5-repeated × 5-fold cross validation. Box plots show IQR and median. **b**, Number of piRNAs and siRNAs predicted to cleave mutated versions of the *L1Md AI* transposon sequence. Data are median and IQR from 100 independent simulations. **c**, AGO and PIWI proteins use different rules to find and slice targets.

target site location in the transcript (5′ untranslated region (UTR), open reading frame (ORF), 3′ UTR or lncRNA; and the identity of target nucleotide t1 (Fig. 4a). The coefficient for each variable in the fitted logistic decision function ($\beta_1, \beta_2, \ldots \beta_{35}$) estimates the importance of each feature as follows:

$$P(\text{cleaved}) = \frac{1}{1 + e^{-(\beta_0 + \beta_1 x_1 + \beta_2 x_2 + \ldots + \beta_{35} x_{35})}}$$

We selected about 3,500 distinct pairs of *pi2*, *pi9* and *p17* piRNAs and target sites for which 5′-monophosphorylated 3′ cleavage products both were detected in the control C57BL/6 mice and had ≥19 nucleotides paired between g2 and g25. Target sites were considered cleaved if the abundance of the 3′ cleavage products decreased by ≥8-fold in the *pi2*^−/−^*pi9*^−/−^*pi17*^−/− mutant mice compared with control mice; all other target sites were assigned as not cleaved. To test whether the logistic regression models created with the *pi2*, *pi9* and *pi17* piRNA data could predict cleavage by non-*pi2*, non-*pi9* and non-*pi17* piRNAs, we generated independent datasets from mice with a mutation disrupting a pachytene piRNA-producing locus on chromosome 7, *pi7* (7-qD2-24830(−)11976(+); Extended Data Fig. 7). Note that >99% of piRNAs eliminated in *pi7*^−/− mice are not found in *pi2*, *pi9* or *pi17*. The performance of the logistic function fitted to *pi2*^−/−^*pi9*^−/−^*pi17*^−/− data

was similar when tested with either *pi2*^−/−^*pi9*^−/−^*pi17*^−/− data (Extended Data Fig. 10a) or *pi7*^−/− data (Extended Data Fig. 10a).

The most predictive features, that is, highest median coefficients in the logistic decision function, were in vivo piRNA concentration (+2.11), predicted energy of piRNA–target base pairing (+1.82) and the total number of paired nucleotides (+1.20; Fig. 4a). These results show that in vivo, piRISC behaves as a conventional enzyme. That is, its concentration and substrate-binding strength determine the efficacy of target cleavage. A high aggregate number of targeting piRNAs was also recently shown to be required for potent transcriptional silencing[44].

Consistent with the idea that PIWI slicing does not rely on complementarity to specific target nucleotides (Figs. 2b and 3c), the median decision function coefficients were ≤+0.6 for guide:target pairing at any individual position (Fig. 4a). Extensive complementarity anywhere in the piRNA 5′ half seems to initiate target binding (Fig. 1b,d), and highly abundant piRNAs (≥500 pM) even direct slicing of targets without pairing to positions g2–g4 (Fig. 3e). In agreement with these data, logistic function coefficients for matches to g2–g10 were higher than those for pairing to other nucleotides ([+0.25, +0.6] compared with [−0.12, +0.25]; Fig. 4a).

The features with the lowest median coefficients were the identity of nucleotide t1 and the location of the target site within a transcript

(−0.18 to +0.13; Fig. 4a). This result suggests that these factors are not rate-determining for piRNA-guided cleavage in vivo.

Together, these analyses show that simple biochemical principles are sufficient to predict efficient piRNA-directed cleavage in vivo. First, piRNA concentration determines how frequently a target encounters piRISC and therefore the concentration of the piRISC–target complex. Second, tighter guide:target base pairing (binding energy) extends the lifetime of the piRISC–target complex, which increases the likelihood of cleavage.

## Implications of piRNA targeting rules

Because PIWI-catalysed slicing does not depend on pairing to a specific piRNA nucleotide position, target mutations are predicted to be better tolerated by PIWI proteins than by AGO proteins. Our computational simulations estimated that when a transposon sequence mutates but the small RNA repertoire does not change, the number of guides capable of directing target slicing decreases by fourfold more slowly for PIWI than AGO proteins (Fig. 4b and Extended Data Fig. 10b). We simulated 1,000 rounds of single-nucleotide substitutions in the *LINE1* consensus sequences, excluding non-synonymous mutations in ORFs. At each round, we recorded the number of embryonic testicular piRNAs or siRNAs (simulated using piRNA 21 nucleotide prefixes) expected to productively slice the mutated transposons. The number of MILI-loaded piRNAs capable of cleavage decreased at 0.01% of guides per single-nucleotide substitution in *LINE1* elements, whereas the number of simulated AGO2-loaded siRNAs decreased at 0.04% of guides per mutation in the transposon sequence (Fig. 4b and Extended Data Fig. 10b).

## Discussion

Our data highlight several distinct features of PIWI proteins that set them apart from AGO-clade Argonaute proteins. First, PIWI proteins do not limit target finding to the seven-nucleotide, 5′ canonical seed. The central cleft that cradles the guide RNA is wider in PIWI than in AGO proteins[33,40,41], which perhaps enables PIWI proteins to productively use piRNA nucleotides 3′ to g8 to initiate pairing with targets. At high concentrations, piRISC efficiently binds and slices RNAs unpaired to nucleotides g2–g4, so targeting capacity is similar for piRNAs for which 5′ ends are several nucleotides apart along a piRNA precursor transcript. This observation explains why piRNAs can tolerate a high degree of 5′ heterogeneity, which is an intrinsic feature of phased (trailing) piRNA biogenesis[17–21]. By contrast, miRNA 5′-isoforms have distinct target repertoires[6], and any change in the 5′ position of a siRNA duplex can invert which strand becomes a guide for an AGO protein.

Second, piRNA-directed slicing tolerates mismatches to any nucleotide of the piRNA guide. Despite more than 900 million years of independent evolution, both mammalian and poriferan PIWI proteins can slice targets even with mismatches adjacent to the scissile bond. In fact, both a perfectly paired and a t10 or t11 mismatched target site were cleaved with similar efficiency when introduced into the 3′ UTR of mouse *Ythdc2* mRNA[45]. Mismatches near the scissile phosphate bond not only block target cleavage by AGO proteins but also promote AGO protein degradation[46,47]. We propose that because piRNAs are generally longer than siRNAs, PIWI proteins can extend the lifetime of the piRISC–target complex through compensatory pairing to piRNA 3′ sequences, which enables piRISC to tolerate multiple guide:target mismatches. Consistent with this model, piRNA 3′ ends are 2′-O-methylated to protect them from decay triggered by extensive pairing to targets[48]. This hypothesis also predicts that cleavage by the ovary-specific mammalian PIWIL3 protein, which uses unusually short, 19-nucleotide-long piRNA guides[49,50], will require full complementarity between their guides and targets.

The catalytic centres of AGO-clade and PIWI-clade Argonaute proteins probably have distinct requirements for effective catalysis of target cleavage. For AGO proteins, perfect pairing between a siRNA and its target moves target nucleotides t10 and t11 into the endonuclease active site[4]. Our data suggest that the catalytically competent geometry of PIWI proteins does not intrinsically rely on perfect complementarity between the target and piRNA near the cleavage site.

Third, the ability of piRNAs to direct cleavage of imperfectly complementary RNAs, but only when the extent of complementarity is more than 15 contiguous base pairs, may explain the rapid evolution of the piRNA target repertoire (Fig. 3b and Extended Data Fig. 3a). Our analyses suggest that a 15-nucleotide contiguous match is sufficient to prevent inappropriate targeting of mRNAs and other self-transcripts. We determined the fraction of all mouse mRNAs and lncRNAs that contain at least one *k*-mer from the consensus sequences of transpositionally active families of mouse LTR or LINE transposons. For $k \geq 15$, less than 5% of mRNAs and lncRNAs shared at least one *k*-mer with transposon sequence (Extended Data Fig. 10c, top). We obtained similar results when these analyses were conducted for intron sequences from the same mouse transcripts (Extended Data Fig. 10c, bottom). This result suggests that strong negative selection against transposon-derived $\geq 15$-mers in mRNAs and lncRNAs is unlikely. Together, our findings (Fig. 4c) provide a plausible explanation for why the piRNA pathway, not RNA interference, was favoured by evolution as the mainstay of transposon defence in animals.

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

## Methods

### Mouse strains and mutants

Mice (Supplementary Table 3; C57BL/6J, International Mouse Strain Resource JAX:000664) were housed in an Association for Assessment and Accreditation of Laboratory Animal Care International-accredited barrier facility at controlled temperature ($22 \pm 2\,°C$), relative humidity ($40 \pm 15\%$) and a 12-h day–light cycle. All experimental animals were 2–6 months old. All procedures were reviewed and performed in compliance with the guidelines of the Institutional Animal Care and Use Committee (IACUC) of the University of Massachusetts Chan Medical School (IACUC protocol number A201900331). Single-guide RNAs (sgRNAs; Supplementary Table 3) were designed using a CRISPR design tool (https://portals.broadinstitute.org/gppx/crispick/public). sgRNAs were transcribed with T7 RNA polymerase and then purified by electrophoresis on a 10% denaturing polyacrylamide gel. gRNA ($20\,ng\,\mu l^{-1}$) and $Cas9$ mRNA ($50\,ng\,\mu l^{-1}$, TriLink Biotechnologies, L-7206) were injected together into the pronucleus of one-cell C57BL/6 zygotes in M2 medium (Sigma, M7167). After injection, the zygotes were cultured in EmbryoMax Advanced KSOM medium (Sigma, MR-106-D) at $37\,°C$ under $5\%\,CO_2$ until the blastocyst stage (3.5 days), then transferred into the uterus of pseudopregnant ICR females 2.5 days post coitum. To screen for mutant founders, gDNA extracted from tail tissues was analysed by PCR using the primers listed in Supplementary Table 3.

No statistical method was used to determine the sample size. For biological samples, the maximum possible sample size ($n = 4$–12) was used for each type of data, which ensured that variability arising from all accountable sources was incorporated in the analyses (animal, day of data collection, reagent lots). No data were excluded from the analyses. Randomization is not relevant to this study because it did not involve treatment or exposure of animals to any agent. Instead, untreated wild-type mice were compared with untreated mutant mice lacking piRNAs from four genomic loci. Blinding is not relevant to this study because during analyses, wild-type control and mutant datasets were easily identified. Blinding was not performed during data acquisition and/or analysis.

### piRNA loading and recombinant piRISC purification for MILI and MIWI

Synthetic piRNA guides (IDT) were purified by electrophoresis through a 15% denaturing polyacrylamide gel. HEK293T cells (American Type Culture Collection) expressing SNAP-tagged, 3×Flag-tagged MILI or MIWI were generated as previously described[27]. Cells were collected at 70% confluency using a TC cell scraper (ThermoFisher, 50809263) into ice-cold PBS and collected by centrifugation at $500g$. Supernatant was removed, and the pellet was stored at $-80\,°C$ until lysed in 10 ml of 30 mM HEPES-KOH, pH 7.5, 100 mM potassium acetate, 3.5 mM magnesium acetate, 2 mM DTT, 0.1% (v/v) Triton X-100, 15% (v/v) glycerol and 1× protease inhibitor cocktail (1 mM 4-(2-aminoethyl)benzenesulfonyl fluoride hydrochloride (Sigma, A8456), 0.3 μM aprotinin, 40 μM betanin hydrochloride, 10 μM E-64 (Sigma, E3132) and 10 μM leupeptin hemisulfate) per g frozen cells. Cell lysis was monitored by staining with trypan blue. Crude cytoplasmic lysate was clarified at $20,000g$, flash frozen in liquid nitrogen and stored at $-80\,°C$.

To capture MILI or MIWI, 1 ml of clarified lysate was incubated with 20 μl anti-Flag M2 paramagnetic beads (Sigma, M8823) for 4 h to overnight rotating at $4\,°C$. Beads were washed four times with extract buffer (30 mM HEPES-KOH, pH 7.5, 3.5 mM magnesium acetate, 2 mM DTT, 15% (v/v) glycerol and 0.01% (v/v) Triton X-100) containing 2 M potassium acetate and four times with extract buffer containing 100 mM potassium acetate. To assemble MILI or MIWI piRISC, beads were resuspended in extract buffer containing 100 mM potassium acetate and 100 nM synthetic piRNA guide (Supplementary Table 4) and incubated with rotation for 30 min at $37\,°C$ or room temperature. After five washes in 2 M potassium acetate extract buffer and five washes in 100 mM

potassium acetate extract buffer, MILI or MIWI piRISC was eluted from the beads twice with $200\,ng\,\mu l^{-1}$ 3×Flag peptide in 100 μl of 100 mM potassium extract buffer with rotation for 1 h at room temperature. The combined 200 μl eluate was used immediately for capture oligonucleotide affinity purification or flash frozen in liquid nitrogen and stored at $-80\,°C$.

To purify MILI or MIWI piRISC loaded with a single synthetic RNA, 200 μl Dynabeads MyOne Streptavidin T1 paramagnetic beads (ThermoFisher, 65601) was washed and incubated with 800 pmol 5′ biotinylated, 2′-O-methyl capture oligonucleotide (Supplementary Table 4) according to the manufacturer's instructions, then resuspended in 100 μl of 100 mM potassium extract buffer. The 200 μl eluate with piRISC from the previous step was added to the capture oligonucleotide-conjugated beads in the extract buffer and incubated with rotation for 1 h at room temperature. The supernatant was removed, and then the beads were washed five times with 100 mM potassium extract, followed by five washes with 2 M potassium extract buffer. piRISC was eluted by rotating the beads for 2 h at room temperature in 200 μl of 100 mM potassium extract buffer containing 1,200 pmol of 5′ biotinylated competitor DNA oligonucleotide (Supplementary Table 4) and S20 testis lysate (total protein $200\,ng\,\mu l^{-1}$ final concentration (f.c., see below)). The supernatant containing eluted piRISC was then incubated for 30 min at room temperature with 300 μl Dynabeads MyOne Streptavidin T1 paramagnetic beads (prewashed according to the manufacturer's instructions followed by two washes in 100 mM potassium extract buffer) to remove excess competitor DNA oligonucleotide. After removing streptavidin beads, 20 μl anti-Flag M2 paramagnetic beads (Sigma, M8823) was added and incubated with the supernatant for 4 h rotating at $4\,°C$ to isolate piRISC from testis lysate. Beads were then washed four times with 2 M potassium acetate extract buffer and four times with 100 mM potassium acetate extract buffer. piRISC was eluted from the beads twice with $200\,ng\,\mu l^{-1}$ 3×Flag peptide in 100 μl of 100 mM potassium extract buffer with rotation for 1 h at room temperature. The combined 200 μl eluate was aliquoted, flash frozen in liquid nitrogen and stored at $-80\,°C$.

### Testis lysate for eluting MILI and MIWI piRISC from capture oligonucleotide

Dissected animal tissue samples were homogenized at $4\,°C$ in 5 volumes of 30 mM HEPES-KOH, pH 7.5, 100 mM potassium acetate, 3.5 mM magnesium acetate, 1 mM DTT and 15% (v/v) glycerol in a dounce homogenizer using 10 strokes of the loose-fitting pestle A, followed by 20 strokes of tight-fitting pestle B to generate crude lysate. S20 was prepared by clarifying the crude lysate at $20,000g$. The protein concentration was estimated using a BCA assay (ThermoFisher, 23200). Crude and fractionated testis lysate were flash frozen in liquid nitrogen and stored at $-80\,°C$.

### piRNA loading and recombinant piRISC purification for *Ef* Piwi

Synthetic piRNA guides (IDT) were purified by electrophoresis through a 15% denaturing polyacrylamide gel. *Ef* Piwi protein was expressed as a $His_6$-TEV-*Ef*Piwi construct using the Bac-to-Bac Baculovirus Expression System (ThermoFisher, 10359016) and Sf9 cells (American Type Culture Collection). Sf9 infection with *Ef*Piwi-expressing baculovirus[33] was performed in 750 ml cultures of $1,275 \times 10^6$ cells for 72 h at $27\,°C$. Each 750 ml culture of Sf9 cells was pelleted and resuspended in 25 ml lysis buffer (50 mM Tris pH 8.0, 300 mM NaCl and 0.5 mM TCEP) and lysed using a high-pressure (18,000 p.s.i.) microfluidizer (Microfluidics M100P). Debris was pelleted by centrifugation, and the clarified lysate was incubated with 1 ml Ni-NTA resin (Qiagen, 30210) per 750 ml culture for 1 h at $4\,°C$, followed by washing twice in nickel wash buffer (50 mM Tris pH 8.0, 300 mM NaCl, 20 mM imidazole and 0.5 mM TCEP). The resin was then washed once with wash buffer supplemented with 5 mM $CaCl_2$ in preparation for micrococcal nuclease treatment to degrade

co-purifying cellular RNAs. The washed resin was resuspended in nickel wash buffer supplemented with 5 mM CaCl$_2$ (final volume of 20 ml). Next, 100 U micrococcal nuclease (Takara Bio, 2910A) was added per 750 ml culture and incubated at room temperature for 1 h, inverting gently every 15 min to resuspend the resin. After three washes with nickel wash buffer without CaCl$_2$, protein was eluted with 6× column volumes of nickel elution buffer (wash buffer supplemented with 300 mM imidazole). Eluted protein was supplemented with 5 mM EGTA to chelate any remaining calcium and dialysed (10,000 MWCO) against 50 mM Tris pH 8.0, 300 mM NaCl, 0.5 mM TCEP buffer overnight at 4 °C.

For each loading procedure, an aliquot of *Ef*Piwi (1/50 of the protein yield from 750 ml of Sf9 culture) was incubated with a synthetic piRNA guide (15 µM f.c.) for 15 min at room temperature and then dialysed into 50 mM Tris pH 8.0, 300 mM NaCl, 0.5 mM TCEP, 0.02% CHAPS buffer overnight at 4 °C (12,000 MWCO). To prepare for capturing guide-loaded *Ef*Piwi, 2.5 nmol of biotinylated capture oligonucleotide was incubated with 40 µl high capacity neutravidin resin (ThermoFisher, 29204) in 1 ml wash A buffer (30 mM Tris pH 8.0, 0.1 M potassium acetate, 2 mM magnesium acetate, 0.02% CHAPS and 0.5 mM TCEP) for 30 min at 4 °C, followed by two washes with 2 ml wash A buffer. *Ef*Piwi–guide complex was captured by incubating with the capture oligonucleotide-conjugated neutravidin resin at room temperature for 1.5 h with rotation. The resin was then washed three times with 2 ml wash A buffer, four times with 2 ml wash B buffer (30 mM Tris pH 8.0, 2 M potassium acetate, 2 mM magnesium acetate, 0.02% CHAPS and 0.5 mM TCEP), and three times with 2 ml wash C buffer (30 mM Tris pH 8.0, 1 M potassium acetate, 2 mM magnesium acetate, 0.02% CHAPS and 0.5 mM TCEP) at 4 °C. The resin was then resuspended in 250 µl wash C buffer containing biotinylated competitor oligonucleotide (50 µM f.c.) and incubated with rotation at room temperature for 3 h. To remove excess competitor oligonucleotide, the supernatant was incubated for 30 min at 4 °C with 60 µl fresh neutravidin resin (prewashed twice in wash C buffer), and the supernatant was dialysed overnight at 4 °C into extract buffer (30 mM HEPES-KOH, pH 7.5, 3.5 mM magnesium acetate, 2 mM DTT, 15% (v/v) glycerol and 0.01% (v/v) Triton X-100). The dialysed *Ef*Piwi–guide RNA complex was aliquoted, flash frozen in liquid nitrogen and stored at −80 °C.

## Recombinant mouse GTSF1 purification

pCold-GST GTSF-expression vectors were transformed into Rosetta-Gami 2 competent cells (Sigma, 71351). Cells were grown to an OD$_{600}$ of 0.6–0.8 in the presence of 1 µM ZnSO$_4$ at 37 °C, then chilled on ice for 30 min to initiate cold shock. Protein expression was induced with 0.5 mM IPTG for 18 h at 15 °C. Cells were collected by centrifugation, washed twice with PBS and cell pellets were flash frozen and stored at −80 °C. Cell pellets were resuspended in lysis/GST column buffer containing 20 mM Tris-HCl pH 7.5, 500 mM NaCl, 1 mM DTT, 5% (v/v) glycerol and 1× protease inhibitor cocktail (1 mM 4-(2-aminoethyl) benzenesulfonyl fluoride hydrochloride (Sigma, A8456), 0.3 µM aprotinin, 40 µM betanin hydrochloride, 10 µM E-64 (Sigma, E3132) and 10 µM leupeptin hemisulfate). Cells were lysed by a single pass at 18,000 p.s.i. through a high-pressure microfluidizer (Microfluidics, M110P), and the resulting lysate clarified at 30,000*g* for 1 h at 4 °C. Clarified lysate was filtered through a 0.22 µm Millex Durapore low-protein-binding syringe filter (EMD Millipore) and applied to glutathione Sepharose 4b resin (Cytiva, 17075604) equilibrated with GST column buffer. After draining the flow through, the resin was washed with 50 column-volumes of GST column buffer. To elute the bound protein and cleave the GST tag in a single step, 50 U HRV3C protease (Millipore, 71493) in 2.5 ml 20 mM Tris-HCl, pH 7.5, 50 mM NaCl, 1 mM DTT and 5% (v/v) glycerol was added to the column, and the column sealed and incubated for 3 h at 4 °C. Next, the column was drained to collect the cleaved protein. The eluate was diluted to 50 mM NaCl and further purified using a HiTrap Q (Cytiva, 29051325) anion exchange column equilibrated with 20 mM Tris-HCl, pH 7.5,

50 mM NaCl, 1 mM DTT and 5% (v/v) glycerol. The bound protein was eluted using a 100–500 mM NaCl gradient in the same buffer. Peak fractions were analysed for purity by SDS–PAGE and the purest were pooled and dialysed into storage buffer containing 30 mM HEPES-KOH, pH 7.5, 100 mM potassium acetate, 3.5 mM magnesium acetate, 1 mM DTT and 20% (v/v) glycerol. Aliquots of the pooled fractions were flash frozen in liquid nitrogen and stored at −80 °C.

## Recombinant *Em*Gtsf1 purification

The high-quality draft genome of *E. muelleri* was used to design the expression construct of *Ephydatia* sp. *Gtsf1* orthologue. The *Em*Gtsf1 expression vector was transformed into BL21(DE3) cells (NEB, C2527H). Transformed cells were grown in LB medium supplemented with 1 µM ZnSO$_4$ at 37 °C until an OD$_{600}$ of 0.6–0.8. The incubation temperature was lowered to 16 °C and protein expression was induced by the addition of 1 mM IPTG for 16 h. Cells were collected by centrifugation and cell pellets flash frozen in liquid nitrogen and stored at −80 °C. Thawed cell pellets were resuspended in lysis buffer (50 mM Tris, pH 8, 300 mM NaCl and 0.5 mM TCEP) and passed through a high-pressure (18,000 p.s.i.) microfluidizer (Microfluidics, M110P) to induce cell lysis. The lysate was clarified by centrifugation at 30,000*g* for 20 min at 4 °C. Clarified lysate was applied to Ni-NTA resin (Qiagen) and incubated for 1 h. The resin was washed with nickel wash buffer (300 mM NaCl, 20 mM imidazole, 0.5 mM TCEP and 50 mM Tris, pH 8.0). Protein was eluted in four column volumes of nickel elution buffer (300 mM NaCl, 300 mM imidazole, 0.5 mM TCEP and 50 mM Tris, pH 8.0). TEV protease was added to the eluted protein to remove the amino-terminal His$_6$ and MBP tags. The resulting mixture was dialysed against HiTrap dialysis buffer (300 mM NaCl, 20 mM imidazole, 0.5 mM TCEP and 50 mM Tris, pH 8.0) at 4 °C overnight. The dialysed protein was then passed through a 5 ml HiTrap chelating column (Cytiva) and the unbound material collected. Unbound material was concentrated and further purified by size-exclusion chromatography using a Superdex 75 Increase 10/300 column (Cytiva) equilibrated in 50 mM Tris, pH 8.0, 300 mM NaCl and 0.5 mM TCEP. Peak fractions were analysed for purity by SDS–PAGE, and the purest were pooled, concentrated to 100 µM, aliquoted and stored at −80 °C.

## Determination of the active fraction of piRISC

In vitro cleavage assays were used to determine the fraction of active piRISC. Target RNA substrates for cleavage assays were prepared as previously described[34]. Fully complementary piRNA target site-containing templates were PCR amplified from pGL2 (primers listed in Supplementary Table 4), in vitro transcribed with T7 RNA polymerase, purified using a 7% denaturing polyacrylamide gel and capped using α-[$^{32}$P] GTP (Perkin Elmer) and a Vaccinia Capping System (NEB, M2080S). Unincorporated α-[$^{32}$P]GTP was removed using a G-25 spin column (Cytiva, 27532501), target RNA was purified using a 7% denaturing polyacrylamide gel, eluted overnight with rotation in 0.4 M NaCl at 4 °C and collected by ethanol precipitation. Radiolabelled target (10 nM f.c.) was added to a mix of purified piRISC and GTSF1 (500 nM f.c.) to assemble a 30 µl cleavage reaction. At 0, 5, 15, 30 and 60 min, a 5 µl sample was quenched in 280 µl 50 mM Tris-HCl, pH 7.5, 100 mM NaCl, 25 mM EDTA and 1% (w/v) SDS, then proteinase K (1 mg ml$^{-1}$ f.c.) was added and the mix incubated at 45 °C for 15 min, followed by extraction with phenol–chloroform–isoamyl alcohol (25:24:1, pH 6.7) and ethanol precipitation. RNA was resuspended in 10 µl 95% (v/v) formamide, 5 mM EDTA, 0.025% (w/v) bromophenol blue and 0.025% (w/v) xylene cyanol, heated at 95 °C for 2 min, and resolved on a 7% denaturing polyacrylamide gel. Gels were dried, exposed to a storage phosphor screen and imaged on a Typhoon FLA 7000 (GE). The raw image file was used to quantify the substrate and product bands, corrected for background. Data were fit to the burst-and-steady-state equation to determine the concentration of active piRISC (see equation and fitting procedure in the section 'Analysis of CNS data').

## RNA bind-'n-Seq for $K_d$ measurements

RNA bind-'n-Seq (RBNS) was performed as previously described[32] with modifications. A library of RNA oligonucleotides containing a central region of 20 random-sequence positions (Extended Data Fig. 1a) was obtained from IDT, 5′[$^{32}$P]-radiolabelled with α-[$^{32}$P]GTP (Perkin Elmer) and T4 PNK (NEB, M0201) and purified using a 15% denaturing polyacrylamide gel, extracted with phenol–chloroform–isoamyl alcohol (25:24:1, pH 6.7) and collected by ethanol precipitation. To sequence the input library, RNA was denatured at 90 °C for 1 min, annealed to a RT primer (Supplementary Table 4) and reverse transcribed with SuperScript III. RNA was degraded by alkaline hydrolysis in 0.4 M NaOH for 1 h at 55 °C, and cDNA was recovered by ethanol precipitation. The sample was then amplified in 25 μl using AccuPrime *Pfx* DNA polymerase (ThermoFisher, 12344024; 95 °C for 2 min, 15 cycles of 95 °C for 15 s, 65 °C for 30 s, 68 °C for 15 s; primers listed in Supplementary Table 4). PCR products were purified with a 2% agarose gel and sequenced on a NextSeq 550 (Illumina) to obtain 79-nucleotide, single-end reads.

For RBNS, DNA-blocking oligonucleotides (Supplementary Table 4) were annealed to the RNA library in 30 mM HEPES-KOH, pH 7.5, 120 mM potassium acetate and 3.5 mM magnesium acetate using a 1:1.2 molar ratio of RNA pool to DNA blockers by first incubating at 95 °C for 1 min, then at 65 °C for 10 min and finally cooled to room temperature. For each trial, the final piRISC concentrations in the six RBNS reactions were 0.003 nM, 0.01 nM, 0.032 nM, 0.1 nM, 0.316 nM and 1 nM active piRISC. Each trial also included a control in which protein storage buffer replaced piRISC. Binding for each piRISC concentration was performed in 20 μl 25 mM HEPES-KOH, pH 7.9, 110 mM potassium acetate, 3.5 mM magnesium acetate, 0.01% (w/v) Triton X-100, 2 mM DTT, 10% (w/v) glycerol and 100 nM (f.c.) RNA library. To reduce non-specific binding, each reaction also included 2.5 mg ml$^{-1}$ BSA and 0.5 mg ml$^{-1}$ yeast tRNA. Reactions were incubated for 2 h at 33 °C (ref. 51) and then filtered through a Whatman Protran nitrocellulose membrane (Sigma, WHA10402506) on top of an Amersham Hybond-XL (Cytiva, RPN2222S) nylon membrane in a Bio-Dot apparatus (Bio-Rad, 1706545). To reduce the retention of free single-stranded RNA, nitrocellulose and nylon membranes were pre-conditioned as previously described[52,53]. In brief, the nitrocellulose filter was pre-soaked in 0.4 M KOH for 10 min, and the nylon filter was incubated in 0.1 M EDTA, pH 8.2 for 10 min, washed three times in 1 M NaCl for 10 min each followed by an 15 s rinse in 0.5 M NaOH. Nitrocellulose and nylon filters were then rinsed in water until the pH returned to neutral and then equilibrated in wash buffer (20 mM HEPES-KOH, pH 7.9, 100 mM potassium acetate, 3.5 mM magnesium acetate and 1 mM DTT) for at least 1 h at 37 °C. After applying seven samples (no-piRISC and six piRISC concentrations; Extended Data Fig. 1a) onto the nitrocellulose and nylon membrane under vacuum, the two membranes were washed with 100 ml wash buffer for 3 s. Membranes were air-dried and signals detected by phosphorimaging to monitor binding. The nitrocellulose membrane areas containing piRISC-bound RNA were excised and incubated with 1 mg ml$^{-1}$ proteinase K in 100 mM Tris-HCl, pH 7.5, 10 mM EDTA, 150 mM sodium chloride and 1% (w/v) SDS for 1 h at 45 °C shaking at 300 r.p.m. After phenol–chloroform extraction and ethanol precipitation, RNA was reverse transcribed, amplified and sequenced as described above for the input RNA pool.

## Determining equilibrium dissociation constants by double-filter binding assays

Binding assays were performed as previously described[34] in 5 μl in 30 mM HEPES-KOH, pH 7.5, 120 mM potassium acetate, 3.5 mM magnesium acetate, 2 mM DTT and 0.01% (w/v) Triton X-100. The 5′[$^{32}$P]-radiolabelled RNA targets (0.1 nM; listed in Supplementary Table 4) were incubated with 0.001–0.8 nM piRISC. The assay also included a control reaction using piRISC storage buffer. Binding reactions were incubated at 33 °C for 2 h. RNA binding was measured by capturing protein–RNA complexes on Protran nitrocellulose (GE, GE10600002) and unbound RNA on a Hybond-XL (Cytiva, 45-001-151) in a Bio-Dot apparatus (Bio-Rad). After applying the sample under vacuum, membranes were washed with 10 μl equilibration buffer (30 mM HEPES-KOH, pH 7.5, 120 mM potassium acetate, 3.5 mM magnesium acetate and 2 mM DTT). Membranes were air-dried and signals detected by phosphorimaging. Because $K_d$ < [RNA target], all binding data were fit to the following equation using IgorPro 6.11 (WaveMetrics):

$$f = \frac{([E_T] + [S_T] + K_d) - \sqrt{([E_T] + [S_T] + K_d)^2 - 4[E_T][S_T]}}{2[S_T]},$$

where $f$ is the fraction target bound, $[E_T]$ is the total piRISC concentration, $[S_T]$ is the total RNA target concentration and $K_d$ is the apparent equilibrium dissociation constant.

## Cleave-'n-Seq to determine target cleavage rates

To the ssDNA oligonucleotide pool of Cleave-'n-Seq (CNS) targets (Extended Data Fig. 3b) obtained from TWIST Bioscience, a T7 promoter was added by PCR (primers listed in Supplementary Table 4). The PCR products were in vitro transcribed with T7 RNA polymerase, then treated with TURBO DNase (ThermoFisher, AM2238), and the CNS target RNA library was purified using a 7% denaturing polyacrylamide gel.

DNA-blocking oligonucleotides (Supplementary Table 4) in 1.2-fold excess were annealed to 100 nM CNS target RNA library in 10 mM Tris-HCl (pH 7.4) and 20 mM NaCl by heating the mixture to 95 °C, cooling it at −0.1 °C s$^{-1}$ to 22 °C and incubating at 22 °C for 5 min. The 100 nM target RNA library was diluted with water to 1 nM, aliquoted and stored at −80 °C. Cleavage assays were performed in 20 mM HEPES-KOH (pH 7.5), 80 mM potassium acetate, 3.5 mM magnesium acetate, 4 mM DTT, 10% (v/v) glycerol and 0.01% (v/v) Triton X-100. Each reaction contained 0.1 nM RNA library, 500 nM GTSF1 and 1 nM active piRISC (single turnover conditions). Reactions were conducted by prewarming components to 33 °C, first mixing piRISC with GTSF1 and then adding target libraries, immediately before incubating at 33 °C (ref. 51) for 0, 1, 2, 4 and 8 min or 0, 20, 60, 120, 240, 480 and 960 min. At each time point, a 5 μl sample was quenched in 280 μl 50 mM Tris-HCl, pH 7.5, 100 mM NaCl, 25 mM EDTA and 1% (w/v) SDS, then proteinase K (1 mg ml$^{-1}$ f.c.) added, and the mix incubated at 45 °C for 15 min followed by extraction with phenol–chloroform–isoamyl alcohol (25:24:1, pH 6.7). The RNA was collected by ethanol precipitation, resuspended in 10 μl water, denatured at 90 °C for 1 min, annealed with 1 μl 50 μM RT primer (Supplementary Table 4) at 65 °C for 5 min, and reverse transcribed with SuperScript III (ThermoFisher, 18080093). cDNA was recovered by ethanol precipitation, and the sample was then amplified in 25 μl using AccuPrime *Pfx* DNA polymerase (ThermoFisher, 12344024; 95 °C for 2 min, 15 cycles of 95 °C for 15 s, 65 °C for 30 s and 68 °C for 15 s; primers listed in Supplementary Table 4). PCR products were purified using a 2% agarose gel and sequenced on a NextSeq 550 (Illumina) to obtain 60-nucleotide, single-end reads. All time points in each trial (that is, both the 0–8 min and the 0–960 min subsets) were sequenced in the same NextSeq 550 run. Data from three trials of each of the 0–8 min and 0–960 min subsets were combined to estimate pre-steady-state cleavage rates.

## Cloning and sequencing 3′ cleavage products from CNS reactions

For Extended Data Fig. 5c, a modified DNA library of CNS targets with 8 nucleotide barcodes (each unique to a target variant) was obtained from TWIST Bioscience (Supplementary Table 5). The procedure was identical to the CNS protocol described in the previous section except for the addition of the 5′ adapter ligation step after annealing the RT primer and before reverse transcription: 3′ cleavage products were ligated to a mixed pool of equimolar amount of two 5′ RNA adapters

(to increase nucleotide diversity at the 5′ end of the sequencing read; Supplementary Table 4) in 20 μl 50 mM Tris-HCl (pH 7.8), 10 mM MgCl$_2$, 10 mM DTT, 1 mM ATP with 60 U high concentration T4 RNA ligase (NEB, M0437M) at 16 °C overnight. Ligation was followed by ethanol precipitation. Cleavage reactions were for performed for 2 h at 33 °C. To account for 5′-to-3′ exonucleolytic digestion or addition of non-templated nucleotides to RNA 5′ ends, a set of five synthetic 5′ monophosphorylated oligonucleotides (Supplementary Table 4) was added to the CNS library before starting the cleavage reaction.

## FACS isolation and immunostaining of mouse germ cells

Testes of 2–6-month-old mice were isolated, decapsulated and incubated for 15 min at 33 °C in 1× Gey's balanced salt solution (GBSS, Sigma, G9779) containing 0.4 mg ml$^{-1}$ collagenase type 4 (Worthington, LS004188) rotating at 150 r.p.m. Seminiferous tubules were then washed twice with 1× GBSS and incubated for 15 min at 33 °C in 1× GBSS with 0.5 mg ml$^{-1}$ trypsin and 1 μg ml$^{-1}$ DNase I, rotating at 150 r.p.m. Next, tubules were homogenized by pipetting through a glass Pasteur pipette for 3 min at 4 °C. FBS (7.5% f.c., v/v) was added to inactivate trypsin, and the cell suspension was then strained through a pre-wetted 70 μm cell strainer (ThermoFisher, 22363548). Cells were collected by centrifugation at 300g for 10 min. The supernatant was removed, cells resuspended in 1× GBSS containing 5% (v/v) FBS, 1 μg ml$^{-1}$ DNase I and 5 μg ml$^{-1}$ Hoechst 33342 (ThermoFisher, 62249) and rotated at 150 r.p.m. for 45 min at 33 °C. Propidium iodide (0.2 μg ml$^{-1}$, f.c.; ThermoFisher, P3566) was added, and cells strained through a pre-wetted 40 μm cell strainer (ThermoFisher, 22363547). Spermatogonia, primary spermatocytes, secondary spermatocytes and round spermatids were purified[48,54] (Supplementary Fig. 5) using a FACSAria II Cell Sorter (BD Biosciences; University of Massachusetts Medical School FACS Core). The 355 nm laser was used to excite Hoechst 33342; the 488 nm laser was used to record forward and side scatter and to excite propidium iodide. Propidium iodide emission was detected using a 610/20 bandpass filter. Hoechst 33342 emission was recorded using 450/50 and 670/50 band pass filters.

Germ cell stages in the unsorted population and the purity of sorted fractions were assessed by immunostaining aliquots of cells. Cells were incubated for 20 min in 25 mM sucrose and then fixed on a slide with 1% (w/v) paraformaldehyde containing 0.15% (v/v) Triton X-100 for 2 h at room temperature in a humidifying chamber. Slides were washed sequentially for 10 min in the following solutions: (1) PBS containing 0.4% (v/v) Photo-Flo 200 (Kodak, 1464510); (2) PBS containing 0.1% (v/v) Triton X-100; and (3) PBS containing 0.3% (w/v) BSA, 1% (v/v) donkey serum (Sigma, D9663), and 0.05% (v/v) Triton X-100. After washing, slides were incubated with primary antibodies in PBS containing 3% (w/v) BSA, 10% (v/v) donkey serum and 0.5% (v/v) Triton X-100 overnight at room temperature in a humidified chamber. Rabbit polyclonal anti-SYCP3 (Abcam, ab15093, RRID:AB_301639; 1:1,000 dilution) and mouse monoclonal anti-γH2AX (Millipore, 05-636, RRID:AB_309864; 1:1,000 dilution) were used as primary antibodies. Slides were washed again as described above and then incubated with secondary donkey anti-mouse IgG (H+L) Alexa Fluor 594 (ThermoFisher, A-21203, RRID:AB_2535789; 1:2,000 dilution) or donkey anti-rabbit IgG (H+L) Alexa Fluor 488 (ThermoFisher, A-21206, RRID:AB_2535792; 1:2,000 dilution) for 1 h at room temperature in a humidified chamber. After incubation, slides were washed three times (10 min each) in PBS containing 0.4% (v/v) Photo-Flo 200 and once for 10 min in 0.4% (v/v) Photo-Flo 200. Finally, slides were dried and mounted in ProLong Gold antifade mountant with DAPI (ThermoFisher, P36931). To assess the purity of sorted fractions, 50–100 cells were staged by DNA, γH2AX and SYCP3 staining[54]. All samples used in this study met the following criteria: spermatogonia, 95–100% pure with ≤5% pre-leptotene spermatocytes; primary spermatocytes, 10–15% leptotene/zygotene spermatocytes, 45–50% pachytene spermatocytes, 35–40% diplotene spermatocytes;

secondary spermatocytes, 100%; round spermatids, 95–100%, ≤5% elongated spermatids.

## Small RNA sequencing library preparation

Total RNA from sorted mouse germ cells was extracted using a mir-Vana miRNA isolation kit (ThermoFisher, AM1560). Small RNA libraries were constructed as previously described[48] with modifications. Before library preparation, an equimolar mix of nine synthetic spike-in RNA oligonucleotides (Supplementary Table 4) was added to each RNA sample to enable absolute quantification of small RNAs (Supplementary Table 6); the median cell volume from ref. 21 was used to calculate the intracellular concentration. To reduce ligation bias and to eliminate PCR duplicates, the 3′ and 5′ adaptors both contained nine random nucleotides at their 5′ and 3′ ends, respectively[55] (Supplementary Table 4) and 3′ adaptor ligation reactions contained 25% (w/v) PEG-8000 (f.c.): 500–1,000 ng total RNA was first ligated to 25 pmol of 3′ DNA adapter (Supplementary Table 4) with adenylated 5′ and dideoxycytosine-blocked 3′ ends in 30 μl of 50 mM Tris-HCl (pH 7.5), 10 mM MgCl$_2$, 10 mM DTT and 25% (w/v) PEG-8000 (NEB) with 600 U of homemade T4 Rnl2tr K227Q at 16 °C overnight. After ethanol precipitation, the 50–90 nucleotide (14–54 nucleotide small RNA + 36 nucleotide 3′ adapter containing unique molecular identifiers) 3′ ligated product was purified from a 15% denaturing urea–polyacrylamide gel (National Diagnostics). After overnight elution in 0.4 M NaCl followed by ethanol precipitation, the 3′ ligated product was denatured in 14 μl water at 90 °C for 60 s, 1 μl of 50 μM RT primer (Supplementary Table 4) was added and annealed at 65 °C for 5 min to suppress the formation of 5′-adapter–3′-adapter dimers during the next step. The resulting mix was then ligated to a mixed pool of equimolar amount of two 5′ RNA adapters (to increase the nucleotide diversity at the 5′ end of the sequencing read; Supplementary Table 4) in 20 μl of 50 mM Tris-HCl (pH 7.8), 10 mM MgCl$_2$, 10 mM DTT and 1 mM ATP with 20 U of T4 RNA ligase (ThermoFisher, EL0021) at 25 °C for 2 h. The ligated product was precipitated with ethanol, cDNA synthesis was performed in 20 μl at 42 °C for 1 h using AMV reverse transcriptase (NEB, M0277), and 5 μl of the RT reaction was amplified in 25 μl using AccuPrime Pfx DNA polymerase (ThermoFisher, 12344024; 95 °C for 2 min, 15 cycles of 95 °C for 15 s, 65 °C for 30 s and 68 °C for 15 s; primers listed in Supplementary Table 4). Finally, the PCR product was purified in a 2% agarose gel. Small RNA sequencing (RNA-seq) libraries samples were sequenced using a NextSeq 550 (Illumina) to obtain 79 nucleotide, single-end reads.

## RNA-seq library preparation

Total RNA from sorted germ cells was extracted using a mirVana miRNA isolation kit (ThermoFisher, AM1560) and used for library preparation as previously described[56] with modifications, including the addition of the ERCC spike-in mix to enable absolute quantification of RNAs and the use of unique molecular identifiers in adapters (Supplementary Table 4) to eliminate PCR duplicates[55]. Before library preparation, 1 μl of 1:100 diluted ERCC spike-in mix 1 (ThermoFisher, 4456740) was added to 1 μg total RNA. To remove rRNA, 1 μg total RNA was hybridized in 10 μl to a pool of 186 rRNA antisense oligos (0.05 μM f.c. each) in 10 mM Tris-HCl (pH 7.4), 20 mM NaCl by heating the mixture to 95 °C, cooling at −0.1 °C s$^{-1}$ to 22 °C, and incubating at 22 °C for 5 min. RNase H (10 U; Lucigen, H39500) was added and the mixture incubated at 45 °C for 30 min in 20 μl containing 50 mM Tris-HCl (pH 7.4), 100 mM NaCl and 20 mM MgCl$_2$. The reaction volume was adjusted to 50 μl with 1× Turbo DNase buffer (ThermoFisher, AM2238) and then incubated with 4 U Turbo DNase (ThermoFisher, AM2238) for 20 min at 37 °C. Next, RNA was purified using RNA Clean & Concentrator-5 (Zymo Research, R1016) to retain ≥200 nucleotide RNAs, followed by the stranded, dUTP-based RNA-seq protocol as previously described[56]. RNA-seq libraries were sequenced using a NextSeq 550 (Illumina) to obtain 79+79 nucleotide, paired-end reads.

## Sequencing of 5′-monophosphorylated long RNAs

Total RNA from sorted mouse germ cells was extracted using a mir-Vana miRNA isolation kit (ThermoFisher, AM1560) and used to prepare a library of 5′-monophosphorylated long RNAs as previously described[21,36] with modifications. rRNA was depleted as described above for RNA-seq libraries. RNA was ligated to a mixed pool of equimolar amount of two 5′ RNA adapters (to increase the nucleotide diversity at the 5′ end of the sequencing read; Supplementary Table 4) in 20 μl of 50 mM Tris-HCl (pH 7.8), 10 mM MgCl$_2$, 10 mM DTT and 1 mM ATP with 60 U of high concentration T4 RNA ligase (NEB, M0437M) at 16 °C overnight. The ligated product was isolated using RNA Clean & Concentrator-5 (Zymo Research, R1016) to retain ≥200 nucleotide RNAs and reverse transcribed in 25 μl with 50 pmol RT primer (Supplementary Table 4) using SuperScript III (ThermoFisher, 18080093). After purification with 50 μl Ampure XP beads (Beckman Coulter, A63880), cDNA was PCR amplified using NEBNext High-Fidelity (NEB, M0541; 98 °C for 30 s; 4 cycles of 98 °C for 10 s, 59 °C for 30 s, 72 °C for 12 s; 6 cycles of 98 °C for 10 s, 68 °C for 10 s, 72 °C for 12 s; and 72 °C for 3 min; primers listed in Supplementary Table 4). PCR products between 200 and 400 bp were isolated from a 1% agarose gel, purified using a QIAquick Gel Extraction kit (Qiagen, 28706), and amplified again with NEBNext High-Fidelity (NEB, M0541; 98 °C for 30 s; 3 cycles of 98 °C for 10 s, 68 °C for 30 s, 72 °C for 14 s; 6 cycles of 98 °C for 10 s, 72 °C for 14 s; and 72 °C for 3 min; primers listed in Supplementary Table 4). The PCR product was purified from a 1% agarose gel and sequenced using a NextSeq 550 or NovaSeq 6000 (Illumina) to obtain 79+79 nucleotide or 150+150 nucleotide, paired-end reads.

## Analysis of RBNS data

To analyse RBNS[31] data, the sequence of the 3′ adapter (5′-TGGA ATTCTCGGGTGCCAAGG-3′) was removed using fastx toolkit (v.0.0.14), then each sequencing read in the RNA input library and piRISC-bound libraries was interrogated for the presence of all binding sites of interest. The entire single-stranded 20 nucleotide random-sequence region flanked by four nucleotides of constant primer-binding sequence on either side (GATCNNNNNNNNNNNNNNNNNNNNTGGA) was searched for the presence of piRISC-binding sites. The sequencing depth of the input library (about $50 \times 10^6$ reads) allowed measurement of input frequencies for ≤12 nucleotide motifs. To interrogate non-overlapping target sets, each ≤10 nucleotide contiguous binding site was required to be flanked by nucleotides that not complementary to the guide: for example, a g4–g12 contiguous target site did not pair to guide positions g3 and g13. Each 11-nucleotide-long contiguous complementary site was required to be flanked by a non-matching nucleotide only at its 5′ end: for example, a g4–g14 contiguous target site did not pair to guide position g3. To eliminate interference from potential piRISC cleavage activity, GTSF1 was omitted from binding reactions; we also relied on the fact that, in our analyses, we do not interrogate sites that are long enough (≥15 nucleotides) to be cleaved by piRISC.

A read was assigned to a site category if it contained one single binding motif. Reads containing multiple instances of binding sites (from the same or a different site category) and reads containing partially overlapping sites were not included in the analysis. Reads that did not have any of the binding motifs of interest were classified as reads with no site. Fitting of the binding model from a previously described method[32] to estimate $K_d$ values for binding sites was performed using a Python-based implementation (MLE_KD.py from https://figshare.com/articles/software/MicroRNA-binding_thermodynamics_and_kinetics_by_RNA_Bind-n-Seq/19180952) on each of the 49 different combinations of 7 initial guesses of piRISC concentration (0.1, 0.2, 0.5, 1, 2, 5 or 10 nM) and 7 initial guesses of $K_d$ for RNA with no enriched site (0.1, 0.2, 0.5, 1, 2, 5 or 10 nM). For each trial, the median of the 49 estimates was reported. Two trials of mouse AGO2 RBNS data for let-7a (piRNA-1 in Fig. 1) are from a previous study[32]; the third trial was conducted separately for

this study. All other mouse AGO2 RBNS data are from a previous study[32]. All human AGO2 RBNS data are from a previous study[57]. AGO2 RBNS data were downloaded from the National Center for Biotechnology Information and analysed using the same binding model as previously described[32]. Predicted binding energy, $\Delta G^0$, was estimated using the RNAplex nearest neighbour algorithm[58].

## Analysis of CNS data

After the sequence of the 3′ adapter (5′-TGGAATTCTCGGGTGCCAAGG-3′) was removed using fastx toolkit (v.0.0.14), CNS library target sites (Supplementary Table 1) were identified without allowing mismatches or insertions or deletions. The 8 nucleotide barcodes were used when 3′ cleavage products were cloned and sequenced (Supplementary Table 5). Sequencing data (representing the abundance of uncleaved targets) were first normalized to the sequencing depth (parts per million (ppm)). To adjust for the decrease in total abundance of the library over the course of cleavage reaction, each ppm value was divided by the sum of ppm values of targets that contained ≤7 nucleotide complementarity to the piRISC piRNA guide. Next, the relative abundance of cleaved product at non-zero time points was inferred as follows: $[P_{relative}] = (ppm_{0\,min} - ppm_{X\,min})/ppm_{0\,min}$. $[P_{relative}]$ ranged from 0 (no cleaved product) to 1 (all substrate cleaved). The combined $[P_{relative}]$ data from three independent trials of each 0–8 min and 0–960 min subsets (that is, three trials of each 1, 2, 4, 8, 20, 60, 120, 240, 480 and 960 min) were used to fit the burst-and-steady-state scheme $E + S \underset{k_{-1}}{\overset{k_1}{\rightleftharpoons}} ES \overset{k_2}{\rightarrow} EP \overset{k_3}{\rightarrow} E + P$, using equation:

$$[P_{relative}] = f(t) = [E_{relative}]([k_2/(k_2 + k_3)]^2 \times (1 - e^{-[k2+k3]t}) + [k_2 k_3/(k_2 + k_3)t])([E_{relative}]).$$

The fit was performed using the Trust Region Reflective algorithm implemented in the optimize.curve_fit function from Python module scipy (v.1.8.1)[59] for the maximum number of 10,000 function evaluations before the termination. The following physically meaningful constraints on the parameters were used: $0.5 \leq [E_{relative}] \leq 1$; $0 \leq k_2 \leq 100$ min$^{-1}$; and for the single turnover experiment setup, $0 \leq k_3 \leq 0.0001$ min$^{-1}$. For each fitting procedure, the mean and the standard deviation of the estimate for each parameter are reported in a Supplementary Table 1. The resulting $(k_2 + k_3)$ was reported as the pre-steady-state cleavage rate $(k)$.

Mouse AGO2 CNS data for let-7a and miR-21 RISCs are from a previous study[3], and mouse AGO2 CNS data for L1MC RISC was generated for this study.

## Analysis of small RNA datasets

The 3′ adapter (5′-TGGAATTCTCGGGTGCCAAGG-3′) was removed using fastx toolkit (v.0.0.14), and PCR duplicates were eliminated as previously described[55]. rRNA matching reads were removed using bowtie (parameter -v 1; v.1.0.0)[60] against the *M. musculus* set in the SILVA rRNA database[61]. Deduplicated and filtered data were analysed using Tailor[62] to account for non-templated tailing of small RNAs. Sequences of synthetic spike-in oligonucleotides (Supplementary Table 4) were identified, allowing no mismatches with bowtie (parameter -v 0; v.1.0.0)[60], and the absolute abundance of small RNAs calculated (Supplementary Table 6). Because piRNA 3′ trimming by PNLDC1 results in piRNA 3′ end heterogeneity, sequencing reads were next grouped by their 5′, 25 nucleotide prefix. For further analyses, we kept only prefix groups that met three criteria. First, the 25 nucleotide prefix unambiguously mapped to a single genomic position (>80% of the 25 nucleotide piRNA prefixes met this criterion). Second, the prefix group total abundance was ≥1 ppm (that is, ≥10 piRNAs/mouse primary spermatocyte), ensuring that, assuming a Poisson or a negative binomial distribution for piRNA concentration in different cells, ≥99.99% of primary spermatocytes contained at least 1 molecule of the piRNA 25 nucleotide prefix. Third, the prefix group total abundance was ≥1 ppm in all 12 replicates

of control C57BL/6 samples (Supplementary Table 6). piRNAs were considered undetectable in $pi2^{-/-} pi9^{-/-} pi17^{-/-}$ mutants if their mean abundance in mutants ($n = 9$) was ≤0.1 ppm.

### RNA-seq analysis

RNA-seq analysis was performed using piPipes for genomic alignment[63]. Before starting piPipes, sequences were reformatted to extract unique molecular identifiers[55]. The reformatted reads were then aligned to rRNA using bowtie2 (v.2.2.0)[64]. Unaligned reads were mapped to mouse genome mm10 using STAR (v.2.3.1)[65] and PCR duplicates were removed[55]. Transcript abundance was calculated using StringTie (v.1.3.4)[66]. Differential expression analysis was performed using DESeq2 (v.1.18.1)[67]. In parallel, reformatted reads were aligned to an index of ERCC spike-in transcripts (ThermoFisher, 4456740) using bowtie (v.1.0.0)[60], PCR duplicates were removed as previously described[55] and the absolute quantity of transcripts calculated (Supplementary Table 7).

### Analysis of 5′-monophosphorylated long RNA-sequencing data

Sequencing data for 5′-monophosphorylated long RNAs was aligned to the mouse genome using piPipes[63]. Before starting piPipes, the degenerate portions of the 5′ adapter sequences were removed (nucleotides 1–15 of read 1). Because each library was sequenced at least twice to increase the sequencing depth, to harmonize the length of paired-end reads from different runs, sequences were trimmed to 64 nucleotide (read 1) + 79 nucleotide (read 2) paired reads. The trimmed reads were then aligned to rRNA using bowtie2 (v.2.2.0)[64]. Unaligned reads were mapped to mouse genome mm10 using STAR (v.2.3.1)[65], alignments with soft clipping of ends were removed using SAMtools (v.1.0.0)[68] and reads with the same 5′ end were merged to represent a single 5′-monophosphorylated RNA species. For further analyses, only unambiguously mapping 5′-monophosphorylated RNA species for which abundance was ≥0.04 ppm were used. For 5′-monophosphorylated RNAs mapped in annotated transcripts, the nucleotide sequence of the corresponding transcript was used to find piRNAs potentially explaining the cleavage, and we used the genomic sequence for 5′-monophosphorylated RNAs mapped outside any annotated transcript. To ensure that piRNA–target combinations for all pairing configurations did not overlap, the piRNA nucleotide immediately after the paired region was required to be unpaired with the target: for example, for g2–g10, g11 was unpaired and thus did not overlap with g2–g11, g2–g12, an so on. Calculating of the fraction of cleaved sites was performed for a collapsed, non-redundant list of cleavage sites, that is, even if a cleavage site was explained by several piRNAs, it was counted only once. Cumulative abundance of all piRNAs explaining each site was used to assess the effect of piRNA concentration.

### Logistic regression classifier implementation

For each of the 16 permutations of 4 C57BL/6 control and 4 $pi2^{-/-} pi9^{-/-} pi17^{-/-}$ mutant primary spermatocyte datasets, we identified 3,150–3,750 5′-monophosphorylated RNAs (that is, potential 3′ cleavage products of piRNA-guided slicing) for which abundance was ≥0.1 ppm and that were explained by ≥19 paired nucleotides between g2 and g25 of $pi2$, $pi9$ and $pi17$ piRNAs (target insertions or deletions were not allowed). Note that although abundance and binding energy remained the best predictive features regardless of the minimum number of paired nucleotides used as a threshold, requiring <19 paired nucleotides produced too few piRNA–target data points to inform the importance of pairing to piRNA 5′ terminal nucleotides. A target site was considered cleaved, that is, $P$(cleaved) = 1, if the abundance of the 5′-monophosphorylated RNAs decreased by ≥8-fold in $pi2^{-/-} pi9^{-/-} pi17^{-/-}$ mutants compared with C57BL/6 controls. All other sites were assigned as uncleaved, that is, $P$(cleaved) = 0.

$$P(\text{cleaved}) = \frac{1}{1 + e^{-(\beta_0 + \beta_1 x_1 + \beta_2 x_2 + \ldots + \beta_{35} x_{35})}}$$

The logistic function representing the probability of target site cleavage, $P$(cleaved), contained 35 independent variables as follows: $x_1$–$x_{24}$: absence (0) or presence (1) of pairing to g2–g25; $x_{25}$: total number of paired nucleotides between g2–g25, rescaled to [0,1]; $x_{26}$: piRNA abundance, that is, the total abundance of all piRNAs with the same 25 nucleotide, 5′ prefix (see the section 'Analysis of small RNA datasets'), rescaled to [0,1]; $x_{27}$: negative of the predicted energy of piRNA–target pairing $\Delta G^0$ estimated with RNAplex[58], rescaled to [0,1] (use of the negative of $\Delta G^0$ creates a positive relationship between strength of binding and probability of cleavage). Moreover, $x_{28}$: equals 1 if t1A, 0 if t1B; $x_{29}$: equals 1 if t1U, 0 if t1V; $x_{30}$: equals 1 if t1C, 0 if t1D; $x_{31}$: equals 1 if t1G, 0 if t1H; $x_{32}$: equals 1 target site is in the 5′ UTR, 0 if outside the 5′ UTR; $x_{33}$: equals 1 if the target site is in the ORF, 0 if outside the ORF; $x_{34}$: equals 1 if the target site is in the 3′ UTR, 0 if outside the 3′ UTR; $x_{35}$: equals 1 if the target site is in lncRNA, 0 if in mRNA.

The logistic function was fit using the Limited-memory Broyden–Fletcher–Goldfarb–Shanno algorithm (L-BFGS) implemented in LogisticRegression from the Python module scikit-learn[69] using L2-regularization ($\lambda = 1$) with default parameters on acceptance of convergence and the maximum number of iterations set at 1,000. To balance cleaved and uncleaved classes, weights inversely proportional to class frequencies were used. RepeatedStratifiedKFold and cross_validate from scikit-learn were used to perform 5× repeated 5-fold cross validation, resulting in 5 × 5 = 25 logistic function fits for each of the 16 permutations of 4 control 4 mutant datasets, generating the total of 25 × 16 = 400 logistic regression models. To assess model performance, area under the precision-recall curve (AUC) for each of the 400 logistic functions was calculated either with the corresponding $pi2^{-/-} pi9^{-/-} pi17^{-/-}$ dataset (400 AUC values total) or with each of the 16 permutations of 4 C57BL/6 and 4 $pi7^{-/-}$ mutant datasets (6,400 AUC values total).

### Simulation of transposon sequence mutagenesis

The consensus sequences of active mouse LINE transposons[70,71] were mutagenized by adding 1,000 random single-nucleotide substitutions at ratios that reflect the established mouse germline mutation rates[72]. Only synonymous substitutions were allowed in LINE ORFs, and 100 independent simulations were performed for each consensus sequence. piRNAs from fetal mouse testis (embryonic day 16.5) were sequenced and used for the analyses, and 21 nucleotide siRNAs were simulated using piRNA 5′ prefixes. piRNA and siRNA guides were predicted to cleave the mutated transposon sequence using the following rules: ≤6 total mismatches at any position were allowed for 26 nucleotide piRNAs; ≤5 total mismatches, ≤1 mismatch between g2 and g8, and no mismatches at g9, g10, g11 and g13 were allowed for 21 nucleotide siRNAs.

### Reporting summary

Further information on research design is available in the Nature Portfolio Reporting Summary linked to this article.

### Data availability

Sequencing data are available from the National Center for Biotechnology Information Small Read Archive under the accession number PRJNA848233. Mouse genome sequence and annotation (build mm10/GRCm38.92) were downloaded from the ftp sites https://ftp.ensembl.org/pub/release-92/fasta/mus_musculus/dna/ and https://ftp.ensembl.org/pub/release-92/gtf/mus_musculus/, respectively. Transposon consensus sequences were obtained from Repbase (v.27.02; https://www.girinst.org/repbase/).

### Code availability

Code used in this work has been deposited at GitHub (https://github.com/ildargv/Gainetdinov_et_al_2023).

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

**Acknowledgements** We thank staff at the University of Massachusetts FACS Core for help sorting mouse germ cells; staff at the University of Massachusetts Transgenic Animal Modeling Core for help generating *pi2*$^{-/-}$, *pi7*$^{-/-}$, *pi9*$^{-/-}$ and *pi17*$^{-/-}$ mice; and members of the Zamore Laboratory for discussions and critical comments on the manuscript. This work was supported in part by NIGMS R35 GM136275 grant to P.D.Z. and 1S10 OD028576 to the University of Massachusetts Flow Cytometry Core Facility. P.D.Z. is an investigator of the Howard Hughes Medical Institute.

**Author contributions** Conceptualization: I.G. and P.D.Z. Methodology: I.G., I.J.M. and P.D.Z. Software: I.G. Investigation: I.G. (purification and loading MILI and MIWI; loading *Ef*Piwi; determination of fraction active for all piRISCs; analyses of RBNS data; CNS experiments and analyses; FACS; RNA, small RNA, 5'-monophosphate RNA library preparation and analyses; logistic regression classifier; and in silico mutagenesis); J.V.-B. (RBNS experiments and library preparation, double-filter binding assays); K.C. (mouse breeding); A.B. (RNA-seq library preparation); C.C. (FACS, RNA-seq library preparation); D.D. (*Ef*Piwi and *Em*Gtsf1 purification); A.A. (mouse GTSF1 purification, generation of HEK293 cells overexpressing MIWI and MILI); and P.-H.W. (generation of *pi2*$^{-/-}$, *pi7*$^{-/-}$, *pi9*$^{-/-}$ and *pi17*$^{-/-}$ mutant mice). Formal analysis: I.G. Funding acquisition: I.J.M. and P.D.Z. Writing original draft: I.G. Writing, review and editing: I.G. and P.D.Z. All the authors discussed the results and approved the manuscript.

**Competing interests** The authors declare no competing interests.

**Additional information**
**Correspondence and requests for materials** should be addressed to Ildar Gainetdinov or Phillip D. Zamore.

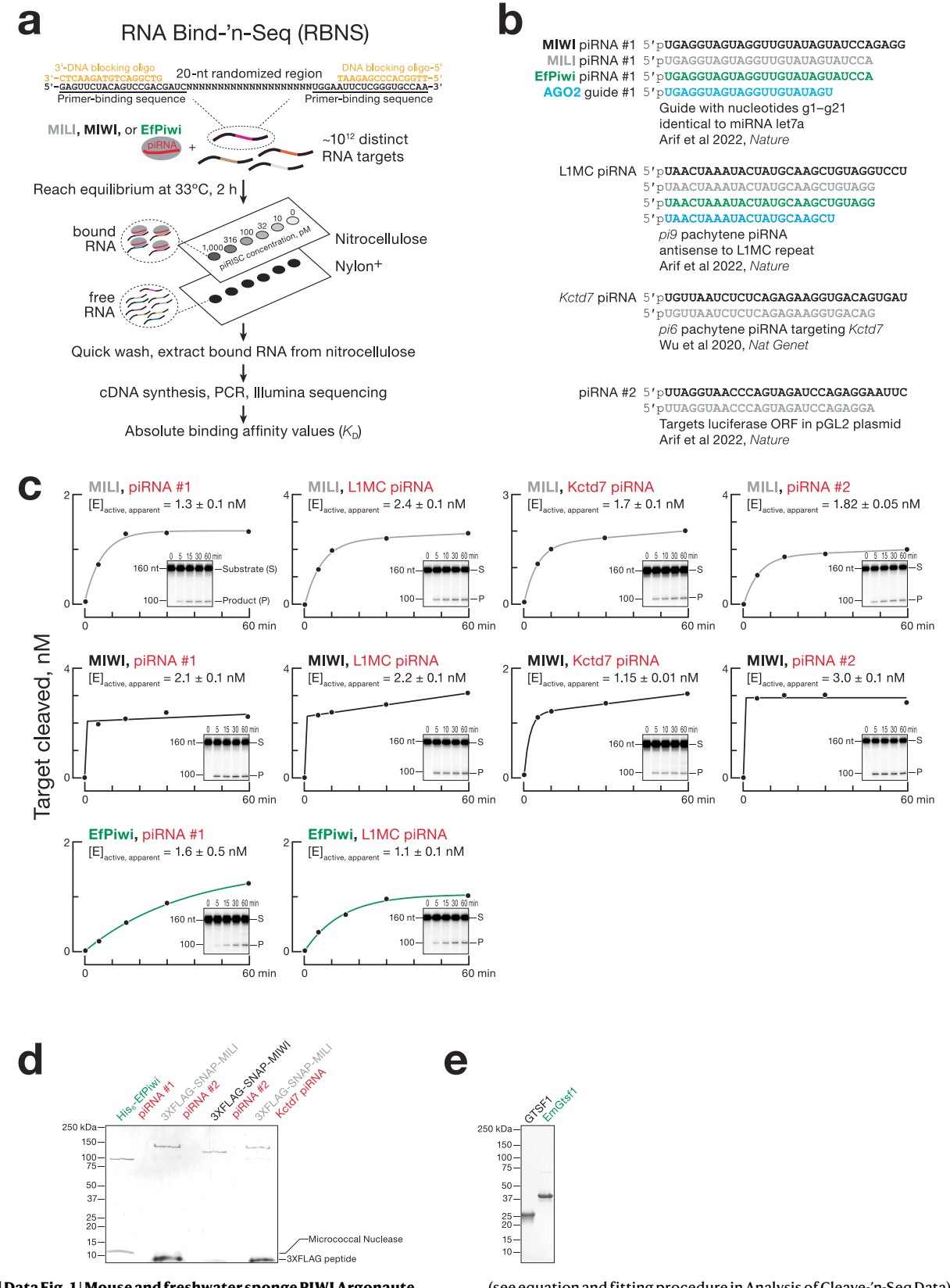

**Extended Data Fig. 1 | Mouse and freshwater sponge PIWI Argonaute proteins. a**, Overview of RNA Bind-'n-Seq. **b**, Sequences of small RNA guides used in this study. **c**, $[E]_{active, apparent}$, the apparent concentration of active MILI, MIWI, or EfPiwi piRISC, determined for each piRISC purification by fitting the cleavage data (from one experiment) to the burst-and-steady-state equation (see equation and fitting procedure in Analysis of Cleave-'n-Seq Data). **d,e**, Silver-stained SDS-PAGE gel showing representative purified MILI, MIWI, and EfPiwi piRISCs (d) and Coomassie-stained SDS-PAGE gel showing the purified EmGtsf1 and mouse GTSF1 used in the study (e). For gel source data, see Supplementary Fig. 1.

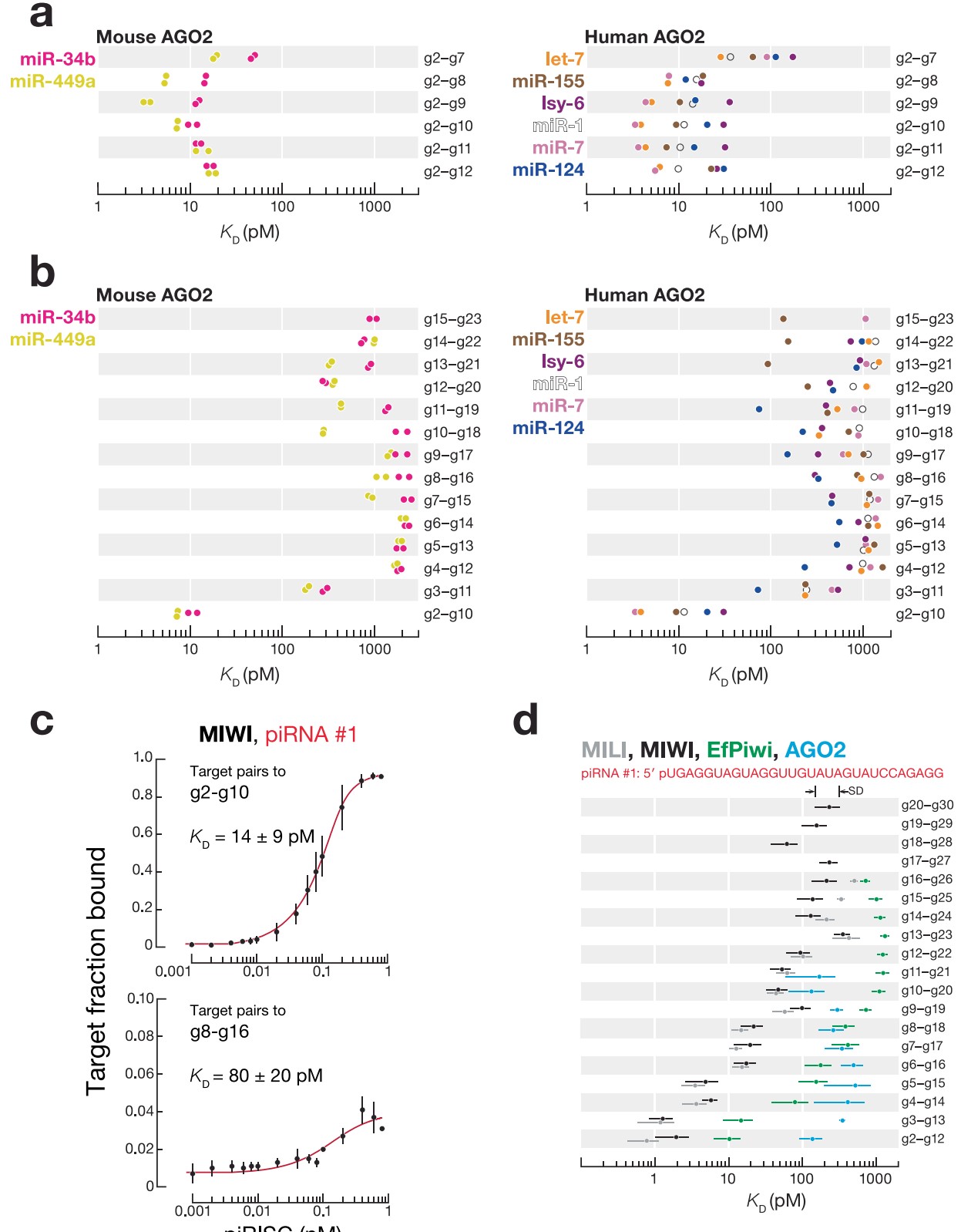

**Extended Data Fig. 2 | Binding affinities of Argonaute proteins. a,b**, Mouse and human AGO2 affinities for 6–11-nt complementary stretches contiguously paired from g2 (**a**) and for 9-nt complementary stretches contiguously paired from all guide nucleotides (**b**) for miR-34b, miR-449a, let-7, miR-1, miR-7, miR-124, miR-155, and lsy-6. Mouse AGO2 data are from ref. 32. Human AGO2 data are from ref. 57. **c**, Measurements of MIWI piRISC affinity for its targets (0.1 nM) using nitrocellulose filter binding assay. Mean and the standard deviation of the data from three independent trials are shown. **d**, MIWI, MILI, EfPiwi, and mouse AGO2 binding affinities for ≥11-nt complementary stretches contiguously paired from all guide nucleotides. Mean and standard deviation from three independent trials are shown.

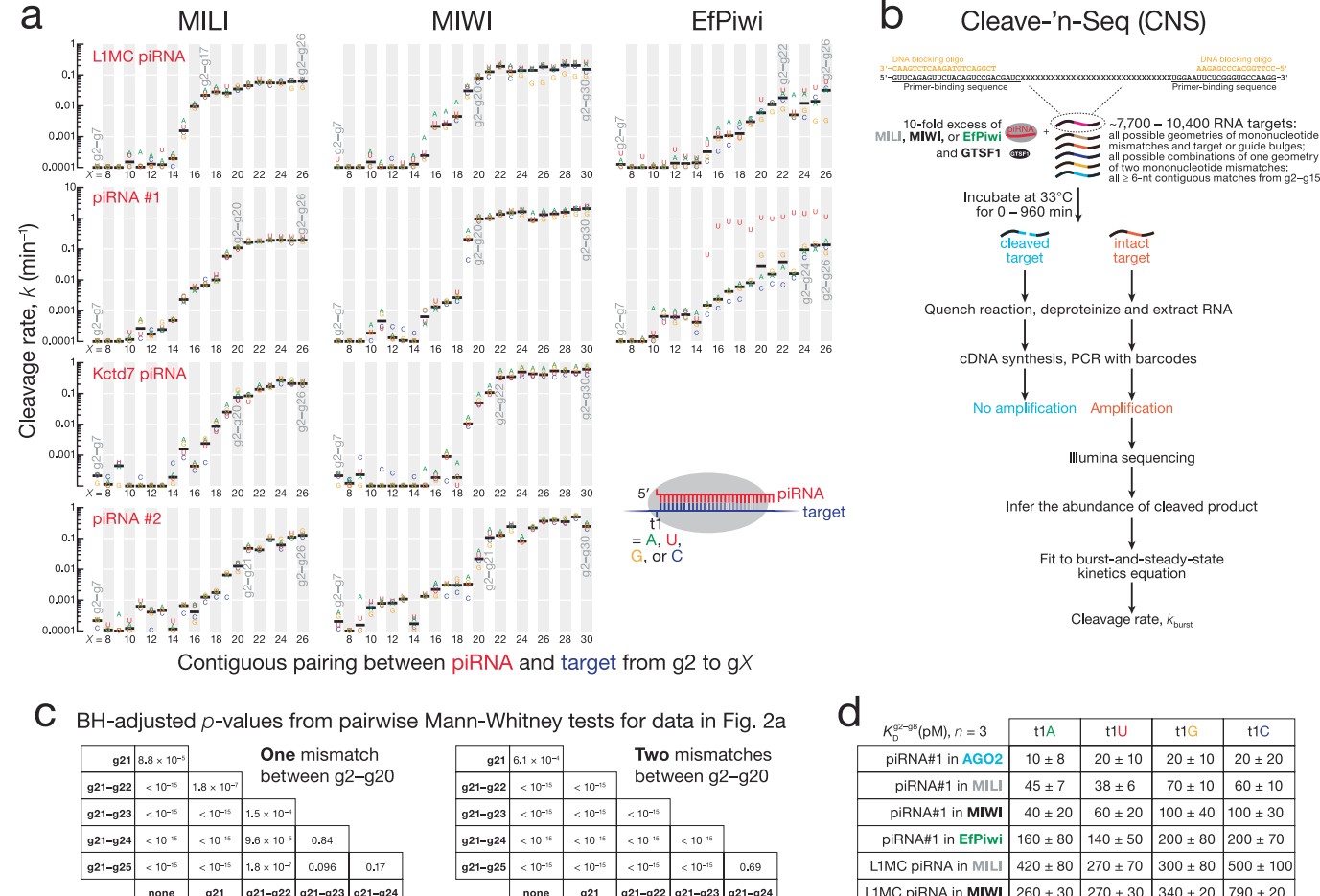

**c** BH-adjusted *p*-values from pairwise Mann-Whitney tests for data in Fig. 2a

| g21 | 8.8 × 10⁻⁵ | | | **One** mismatch |
|---|---|---|---|---|
| **g21–g22** | < 10⁻¹⁵ | 1.8 × 10⁻⁷ | | between g2–g20 |
| **g21–g23** | < 10⁻¹⁵ | < 10⁻¹⁵ | 1.5 × 10⁻⁴ | |
| **g21–g24** | < 10⁻¹⁵ | < 10⁻¹⁵ | 9.6 × 10⁻⁵ | 0.84 |
| **g21–g25** | < 10⁻¹⁵ | < 10⁻¹⁵ | 1.8 × 10⁻⁷ | 0.096 | 0.17 |
| | none | g21 | g21–g22 | g21–g23 | g21–g24 |

| g21 | 6.1 × 10⁻¹ | | | **Two** mismatches |
|---|---|---|---|---|
| **g21–g22** | < 10⁻¹⁵ | < 10⁻¹⁵ | | between g2–g20 |
| **g21–g23** | < 10⁻¹⁵ | < 10⁻¹⁵ | < 10⁻¹⁵ | |
| **g21–g24** | < 10⁻¹⁵ | < 10⁻¹⁵ | < 10⁻¹⁵ | < 10⁻¹⁵ |
| **g21–g25** | < 10⁻¹⁵ | < 10⁻¹⁵ | < 10⁻¹⁵ | < 10⁻¹⁵ | 0.69 |
| | none | g21 | g21–g22 | g21–g23 | g21–g24 |

**d**

| $K_D^{g2-g8}$ (pM), $n = 3$ | t1A | t1U | t1G | t1C |
|---|---|---|---|---|
| piRNA#1 in AGO2 | 10 ± 8 | 20 ± 10 | 20 ± 10 | 20 ± 20 |
| piRNA#1 in MILI | 45 ± 7 | 38 ± 6 | 70 ± 10 | 60 ± 10 |
| piRNA#1 in MIWI | 40 ± 20 | 60 ± 20 | 100 ± 40 | 100 ± 30 |
| piRNA#1 in EfPiwi | 160 ± 80 | 140 ± 50 | 200 ± 80 | 200 ± 70 |
| L1MC piRNA in MILI | 420 ± 80 | 270 ± 70 | 300 ± 80 | 500 ± 100 |
| L1MC piRNA in MIWI | 260 ± 30 | 270 ± 30 | 340 ± 20 | 790 ± 20 |

**Extended Data Fig. 3 | Pairing to piRNA 3′ end is dispensable for PIWI slicing. a**, MILI, MIWI, and EfPiwi pre-steady-state cleavage rates for targets of piRNAs contiguously paired from nucleotide g2. Data are for targets with all possible identities of nucleotide t1. **b**, Overview of Cleave-'n-Seq. **c**, Benjamini-Hochberg corrected *p-value*s for *post hoc* pairwise, two-tailed Mann-Whitney tests for difference in pre-steady-state cleavage rate of targets in Fig. 2a. Kruskal-Wallis test (one-way ANOVA on ranks) *p*-values are < 10⁻¹⁵ for data with one and two mismatches. **d**, MILI, MIWI, EfPiwi, and mouse AGO2, binding affinities ($K_D$) for a g2–g8 match with different t1 nucleotide identities. Mean and standard deviation from three independent trials are shown.

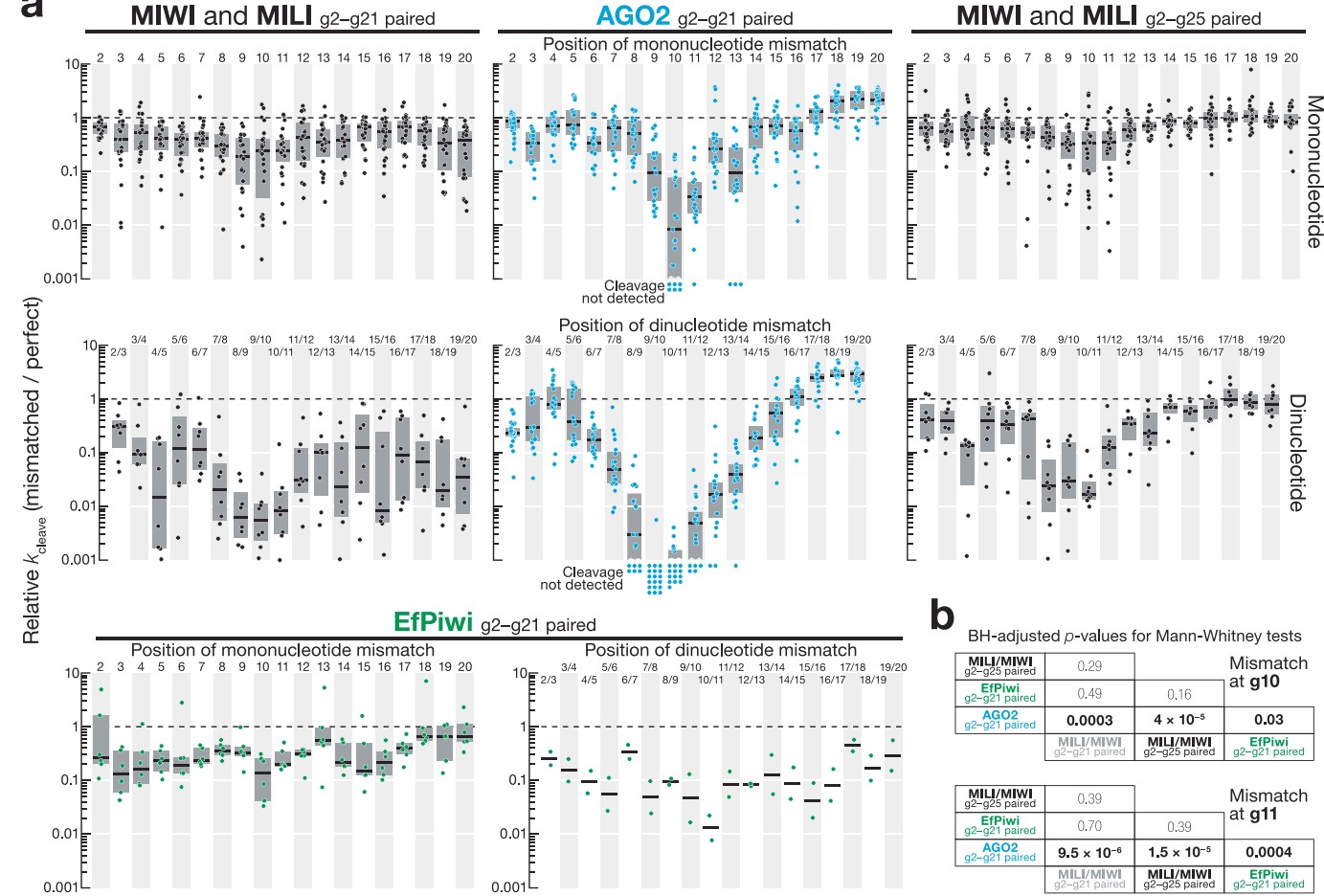

**Extended Data Fig. 4 | PIWI slicing tolerates mononucleotide and dinucleotide mismatches at any position. a**, Change in EfPiwi, MILI, MIWI, mouse AGO2 pre-steady-state cleavage rate for one or two consecutive mismatches between g2–g20. Box plots show IQR and median: for one mismatch, $n = 24$ (three geometries × four piRNAs × MILI and MIWI), $n = 6$ for EfPiwi (three geometries × two piRNAs), $n = 21$ for AGO2 (three geometries for L1MC guide and three geometries × three contexts for let-7a and miR-21 guides); for two consecutive mismatches, $n = 8$ (one geometry × four piRNAs × MILI and

MIWI), $n = 2$ for EfPiwi (one geometry × two piRNAs), $n = 19$ for AGO2 (one geometry for L1MC RISC and nine geometries for let-7a and miR-21 RISCs). **b**, Benjamini-Hochberg corrected *p-values* for *post hoc* pairwise, two-tailed Mann-Whitney tests for difference in pre-steady-state cleavage rate of targets with a mononucleotide mismatch either at g10 or g11 among EfPiwi, MILI, MIWI, mouse AGO2 in panel a. Kruskal-Wallis test (one-way ANOVA on ranks) *p*-value $= 10^{-5}$ for mismatch at g10 and *p*-value $= 4.1 \times 10^{-6}$ for mismatch at g11.

## a

**MILI, MIWI,** and **EfPiwi** (g2–g25 paired)
Mismatches **between g2 and g20**

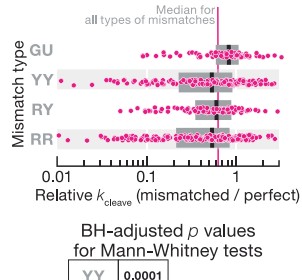

BH-adjusted *p* values
for Mann-Whitney tests

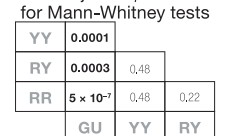

| | GU | YY | RY |
|---|---|---|---|
| YY | 0.0001 | | |
| RY | 0.0003 | 0.48 | |
| RR | 5 × 10⁻⁷ | 0.48 | 0.22 |

## b

**MILI, MIWI,** and **EfPiwi** (g2–g25 paired)
Mismatches **at g10 or g11**

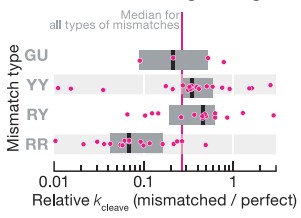

BH-adjusted *p* values
for Mann-Whitney tests

| | GU | YY | RY |
|---|---|---|---|
| YY | 0.61 | | |
| RY | 0.61 | 0.29 | |
| RR | 0.13 | 0.0007 | 0.0005 |

## c

**MIWI,** L1MC piRNA
g2–g25 target with mono- or dinucleotide mismatch:          Contiguous pairing:

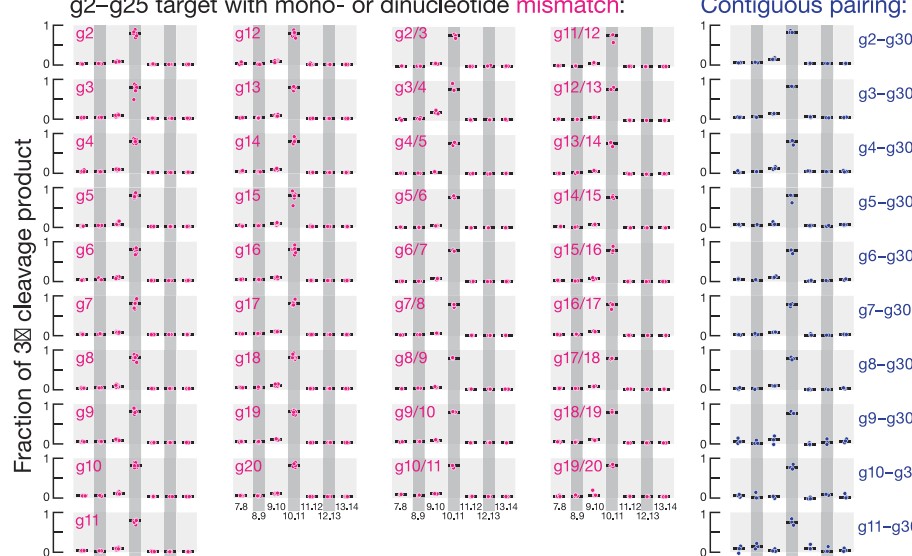

Position of hydrolyzed phosphodiester bond in target RNA

Spike-in controls:

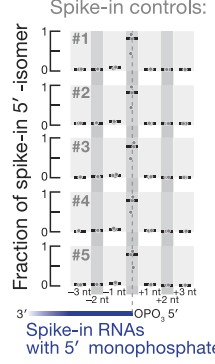

**Extended Data Fig. 5 | PIWI slicing tolerance for different geometries of mismatches. a,** Change in MILI, MIWI, and EfPiwi pre-steady-state cleavage rate for mismatches between g2–g20. Data are binned by mismatch geometry. Data are for all possible mononucleotide mismatch geometries at all 19 positions between g2–g20 for ten piRISCs (Extended Data Fig. 1c). Box plots show IQR and median. Kruskal-Wallis test (one-way ANOVA on ranks) *p*-value = 1.3 × 10⁻⁶. Benjamini-Hochberg corrected *p*-values for *post hoc* pairwise Mann-Whitney tests are shown. **b,** Change in MILI, MIWI, and EfPiwi pre-steady-state cleavage rate for mismatches at g10 or g11. Data are binned by mismatch geometry. Data are for all possible mononucleotide mismatch

geometries at g10 and g11 for ten piRISCs (Extended Data Fig. 1c). Box plots show IQR and the median. Kruskal-Wallis test (one-way ANOVA on ranks) *p*-value = 0.0004. Benjamini-Hochberg corrected *p-value*s for *post hoc* pairwise Mann-Whitney tests are shown. **c,** Relative abundance of 3' cleavage products generated by L1MC-guided MIWI (2 h at 33 °C). All data and the median from three independent trials are shown. Data are also shown for spike-in RNAs that contained no target sites and were added to the reaction to account for 5'-to-3' exonucleolytic trimming or non-templated addition of nucleotides to RNA 5' ends.

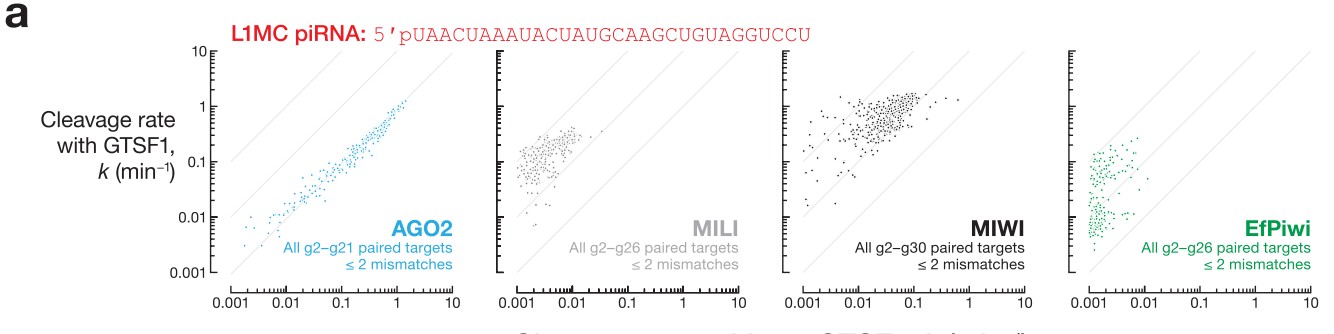

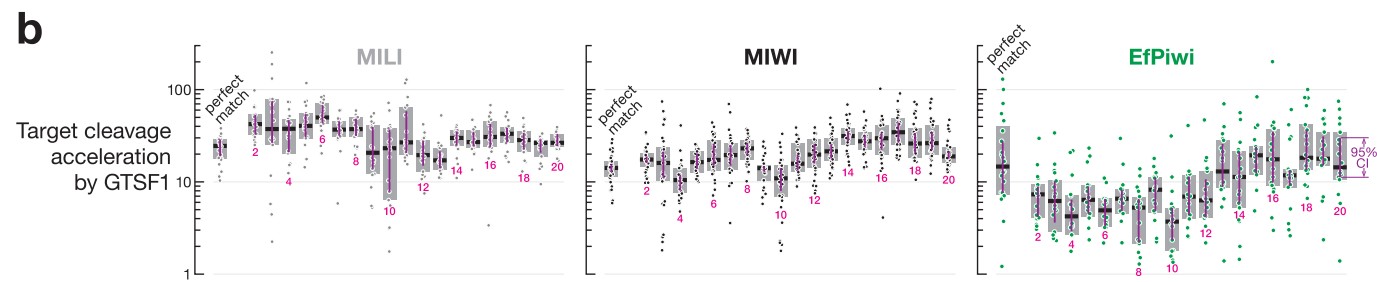

**Extended Data Fig. 6 | GTSF1 accelerates cleavage by PIWI proteins.**
**a**, Pre-steady-state cleavage rate in absence or presence of mouse GTSF1 for mouse AGO2, MILI and MIWI, and *Em*Gtsf1 for EfPiwi. **b**, Acceleration of pre-steady-state cleavage rates by mouse GTSF1 for MILI and MIWI, and by EmGtsf1 for EfPiwi. Data are for four different piRNA guide sequences bound to MILI or MIWI and for two piRNA guide sequences bound to EfPiwi (Extended Data Fig. 1c). Box plots show IQR and median; 95% confidence interval was calculated with 10,000 bootstrapping iterations.

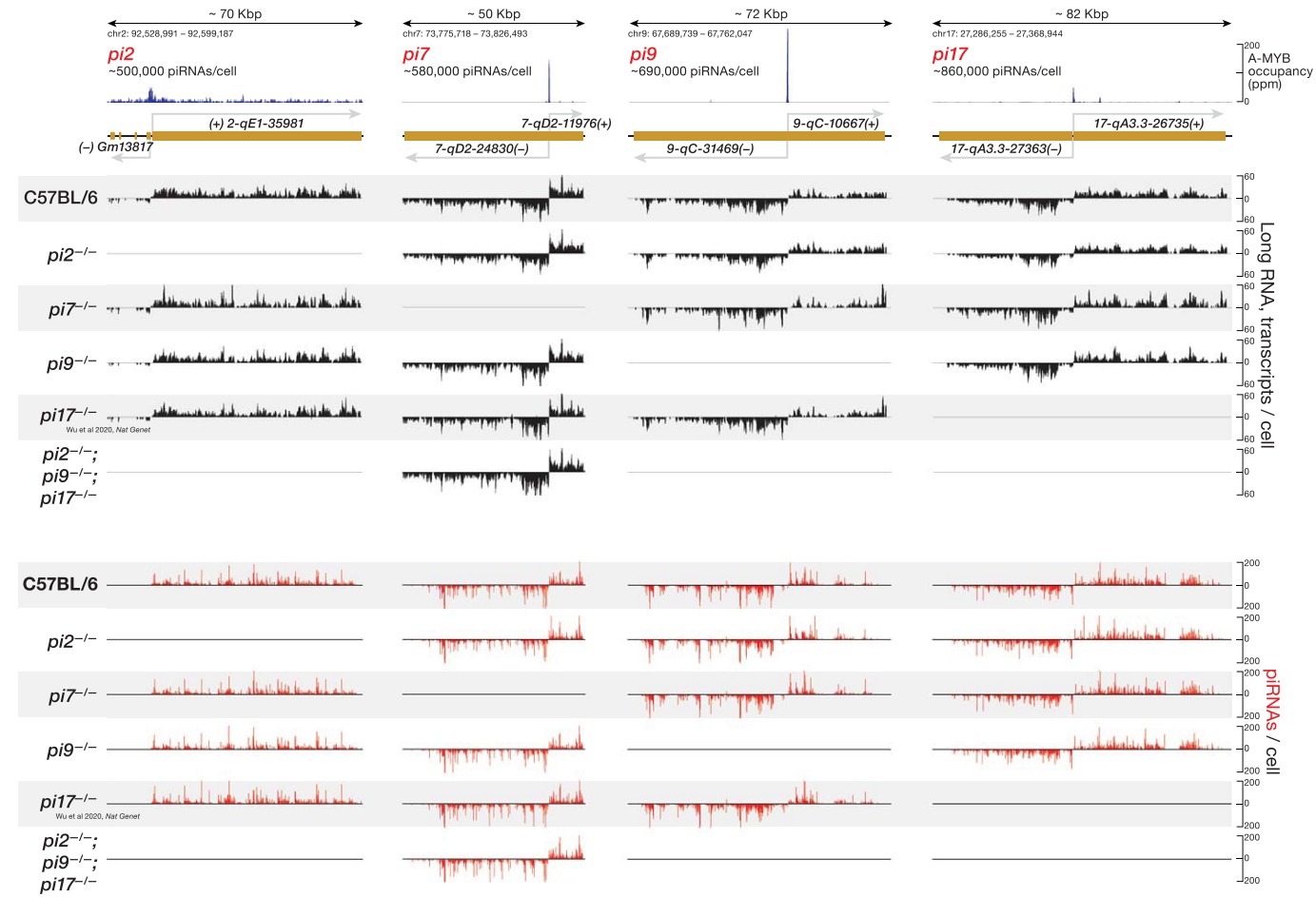

**Extended Data Fig. 7 | *pi2*^–/–, *pi7*^–/–, *pi9*^–/–, and *pi17*^–/– promoter deletions in mice.** Steady-state abundance of piRNA precursor transcripts and mature piRNAs in FACS-purified mouse primary spermatocytes.

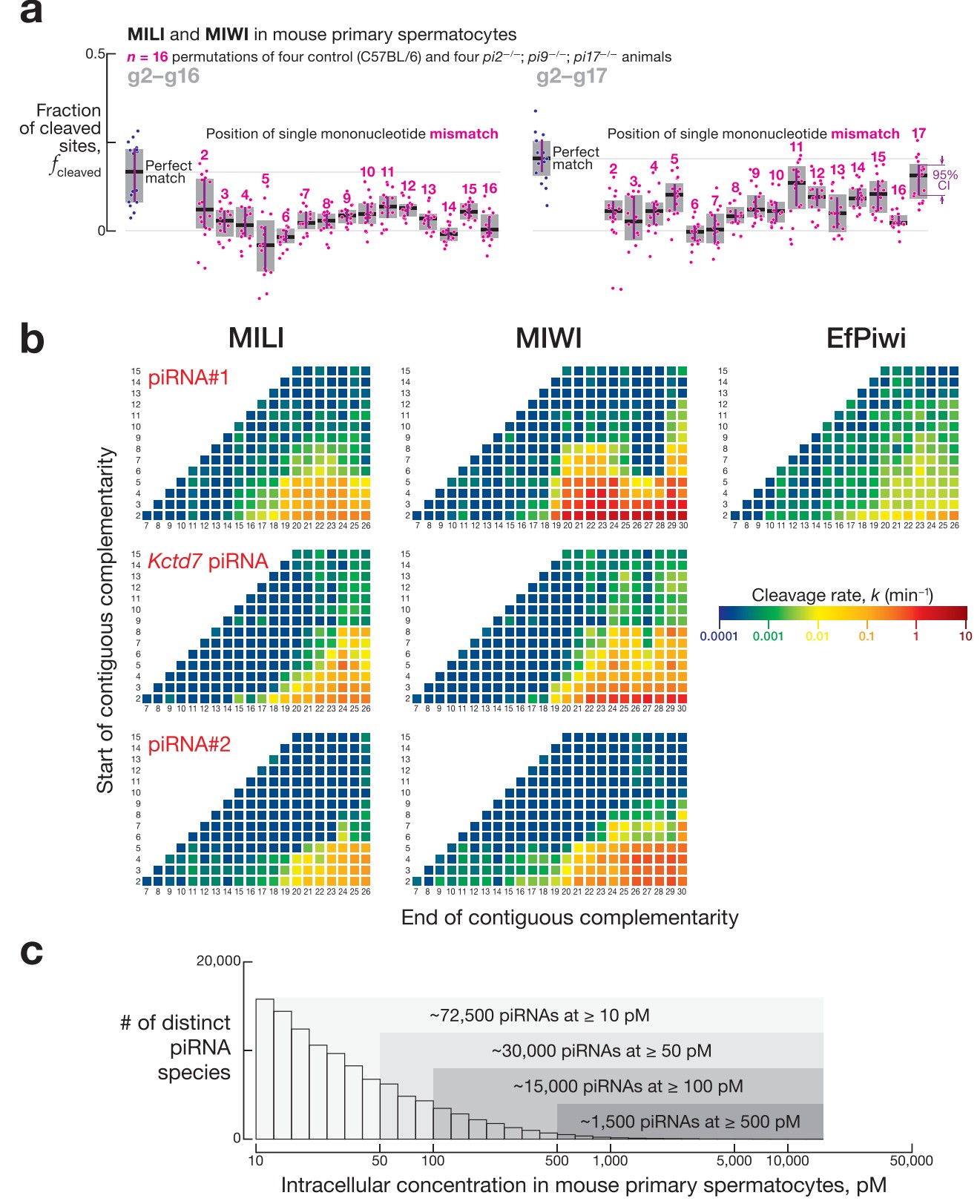

**Extended Data Fig. 8 | PIWI slicing does not require pairing to piRNA 5′ end.**
**a**, Fraction of cleaved targets in FACS-purified mouse primary spermatocytes for pairing containing a single mononucleotide mismatch. Box plots show IQR and median; 95% confidence interval was calculated with 10,000 bootstrapping iterations. **b**, MILI, MIWI, and EfPiwi pre-steady-state cleavage rates in vitro for all possible stretches of ≥ 6-nt contiguous pairing starting from nucleotides g2–g15 of piRNA #1, *Kctd7* piRNA, and piRNA #2. **c**, Intracellular concentration of pachytene piRNAs in mouse primary spermatocytes. Data are the mean of 12 biologically independent samples.

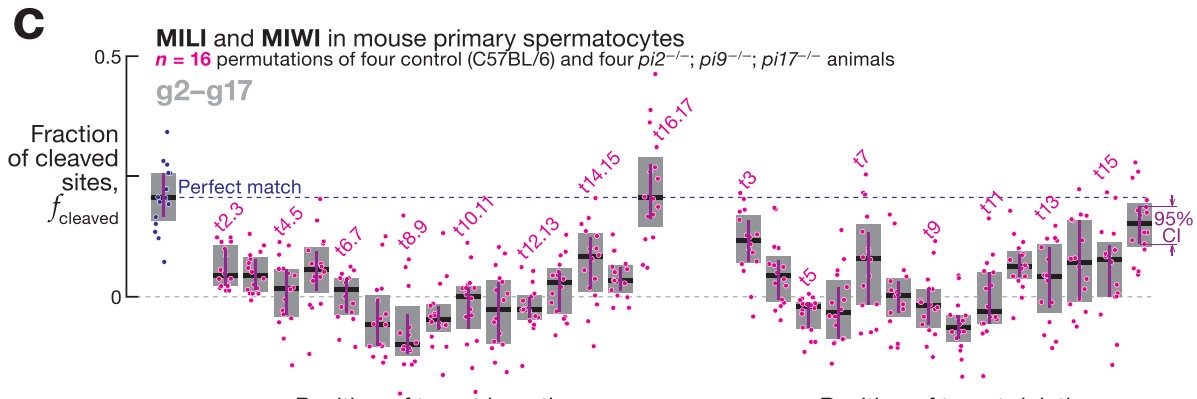

**Extended Data Fig. 9 | Target insertions and deletions in the center of piRNA:target duplex are detrimental for PIWI slicing. a,b,** Change in pre-steady-state cleavage rate for mononucleotide target insertions (**a**) or target deletions (**b**). Target insertion data are for 16 or 40 targets: four insertion geometries for ten piRISCs (MILI and EfPiwi for pairing up to g26, or MIWI for pairing up to g30). Guide bulge data are for four or ten targets: one deletion geometry for ten piRISCs (MILI and EfPiwi for pairing up to g26 or MIWI for pairing up to g30). Box plots show IQR and median. **c,** Fraction of cleaved targets in FACS-purified mouse primary spermatocytes for pairing containing a single mononucleotide bulge in target or guide sequence. Box plots show IQR and median; 95% confidence interval was calculated with 10,000 bootstrapping iterations.

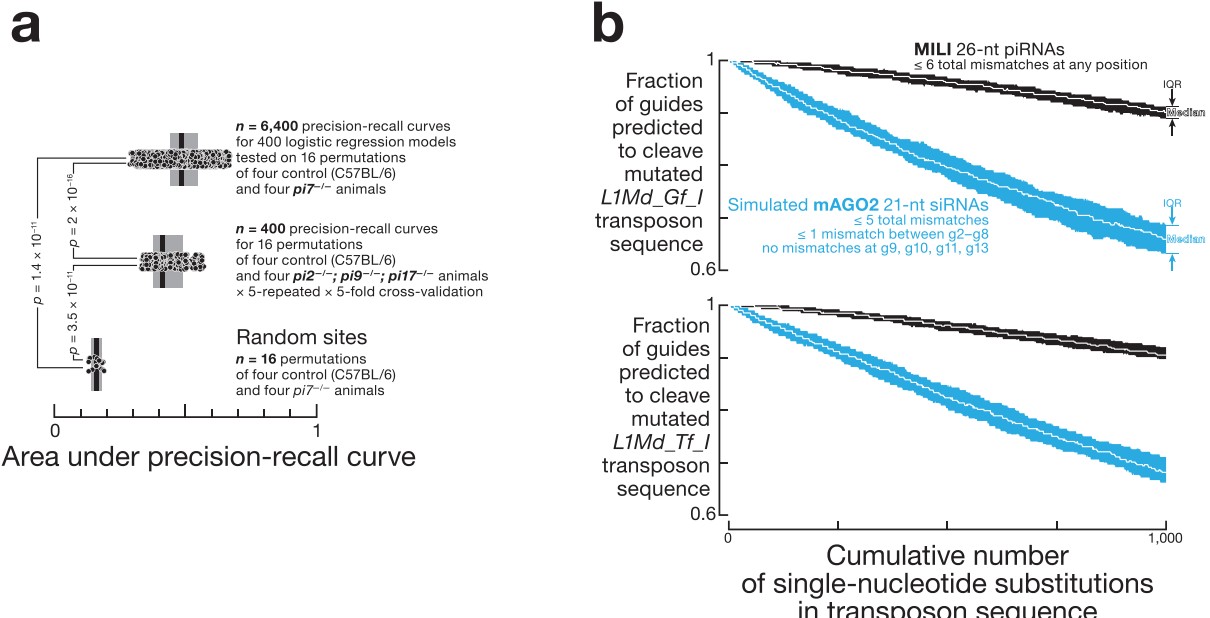

**a**

*n* = **6,400** precision-recall curves for 400 logistic regression models tested on 16 permutations of four control (C57BL/6) and four *pi7*⁻/⁻ animals

*n* = **400** precision-recall curves for 16 permutations of four control (C57BL/6) and four *pi2*⁻/⁻; *pi9*⁻/⁻; *pi17*⁻/⁻ animals × 5-repeated × 5-fold cross-validation

Random sites
*n* = **16** permutations of four control (C57BL/6) and four *pi7*⁻/⁻ animals

$p = 1.4 \times 10^{-11}$

$p = 2 \times 10^{-16}$

$p = 3.5 \times 10^{-16}$

0 1

Area under precision-recall curve

**b**

**MILI** 26-nt piRNAs
≤ 6 total mismatches at any position

IQR
Median

Simulated **mAGO2** 21-nt siRNAs
≤ 5 total mismatches
≤ 1 mismatch between g2–g8
no mismatches at g9, g10, g11, g13

IQR
Median

Fraction of guides predicted to cleave mutated *L1Md_Gf_I* transposon sequence

1
0.6

Fraction of guides predicted to cleave mutated *L1Md_Tf_I* transposon sequence

1
0.6

0 1,000

Cumulative number of single-nucleotide substitutions in transposon sequence

**c**

LINEs:
A_I
A_II
A_III
A_IV
A_V
A_VI
A_VII
Tf_I
Tf_II
Tf_III
Gf_I
Gf_II

LTR-transposons:
IAP1−MM_I
IAP1−MM_LTR
IAPA_MM
IAPEY2_LTR
IAPEY3_I
IAPEY3_LTR
IAPEY3B_I
IAPEY3C_LTR
IAPEY3D_LTR
IAPEY4_I
IAPEY4_LTR
IAPEY5_I
IAPEY5_LTR
IAPEYI
IAPEZI
IAPLTR1_Mm
IAPLTR1a_I_MM
IAPLTR1a_LTR_MM
IAPLTR1b_I_MM
IAPLTR1b_LTR_MM
IAPLTR3_LTR
IAPLTR3_I
IAPLTR3b_I
IAPLTR4_LTR
IAPLTR4_I

Fraction of transcripts, whose **exons** contain ≥ 1 *k*-mer from transposon consensus

1
0.05
0

*k* = 6 8 10 12 14 16 18 20 22 24 26 28 30

Fraction of transcripts, whose **introns** contain ≥ 1 *k*-mer from transposon consensus

1
0.05
0

*k* = 6 8 10 12 14 16 18 20 22 24 26 28 30

**Extended Data Fig. 10 | Determinants of efficient PIWI slicing in vivo.**
**a**, Area under the precision-recall curves for random control, 400 logistic regression classifier models trained with *pi2*⁻/⁻; *pi9*⁻/⁻; *pi17*⁻/⁻ data, and 6,400 tests of the 400 models using *pi7*⁻/⁻ data. Box plots show IQR and median. Kruskal-Wallis test (one-way ANOVA on ranks) *p*-value = $4.5 \times 10^{-12}$. FDR (Benjamini-Hochberg) corrected *p-value*s for *post hoc* pairwise Mann-Whitney tests are shown. **b**, Number of piRNAs and siRNAs predicted to cleave L1Md_Gf and L1Md_Tf transposon sequences. Data are median and IQR from 100 independent simulations. **c**, Sequence overlap between transpositionally active families of LINEs and LTR-transposons and exons or introns of mouse mRNAs and lncRNAs.

# Reporting Summary

## Statistics

For all statistical analyses, confirm that the following items are present in the figure legend, table legend, main text, or Methods section.

| n/a | Confirmed | |
|---|---|---|
| ☐ | ☒ | The exact sample size (*n*) for each experimental group/condition, given as a discrete number and unit of measurement |
| ☐ | ☒ | A statement on whether measurements were taken from distinct samples or whether the same sample was measured repeatedly |
| ☐ | ☒ | The statistical test(s) used AND whether they are one- or two-sided<br>*Only common tests should be described solely by name; describe more complex techniques in the Methods section.* |
| ☒ | ☐ | A description of all covariates tested |
| ☐ | ☒ | A description of any assumptions or corrections, such as tests of normality and adjustment for multiple comparisons |
| ☐ | ☒ | A full description of the statistical parameters including central tendency (e.g. means) or other basic estimates (e.g. regression coefficient) AND variation (e.g. standard deviation) or associated estimates of uncertainty (e.g. confidence intervals) |
| ☐ | ☒ | For null hypothesis testing, the test statistic (e.g. *F*, *t*, *r*) with confidence intervals, effect sizes, degrees of freedom and *P* value noted<br>*Give P values as exact values whenever suitable.* |
| ☒ | ☐ | For Bayesian analysis, information on the choice of priors and Markov chain Monte Carlo settings |
| ☒ | ☐ | For hierarchical and complex designs, identification of the appropriate level for tests and full reporting of outcomes |
| ☐ | ☒ | Estimates of effect sizes (e.g. Cohen's *d*, Pearson's *r*), indicating how they were calculated |

*Our web collection on statistics for biologists contains articles on many of the points above.*

## Software and code

Policy information about availability of computer code

| | |
|---|---|
| Data collection | Typhoon FLA 7000; BD FACSAria II Cell Sorter; Illumina NextSeq 550; Illumina NovaSeq 6000. |
| Data analysis | python2 (v2.7.9) and python3 (v3.8.2) including the following modules: Trust Region Reflective algorithm in optimize.curve_fit from scipy (v1.8.1), RepeatedStratifiedKFold, cross_validate, and Limited-memory Broyden–Fletcher–Goldfarb–Shanno algorithm (L-BFGS) in LogisticRegression from scikit-learn; RNAplex from ViennaRNA (v2.5.1); fastx toolkit (v0.0.14); bowtie2 (v2.2.0); STAR (v2.3.1); StringTie (v1.3.4); DESeq2 (v1.18.1); bowtie (v1.0.0); SAMtools (v1.0.0); Microsoft Excel 2013; CRISPR design tool (crispr.mit.edu/); IgorPro 6.11 (WaveMetrics); Tailor (https://github.com/jhhung/Tailor); piPipes (https://github.com/bowhan/piPipes) |

For manuscripts utilizing custom algorithms or software that are central to the research but not yet described in published literature, software must be made available to editors and reviewers. We strongly encourage code deposition in a community repository (e.g. GitHub). See the Nature Portfolio guidelines for submitting code & software for further information.

## Data

Policy information about availability of data

All manuscripts must include a data availability statement. This statement should provide the following information, where applicable:

- Accession codes, unique identifiers, or web links for publicly available datasets
- A description of any restrictions on data availability
- For clinical datasets or third party data, please ensure that the statement adheres to our policy

Sequencing data are available from the National Center for Biotechnology Information Small Read Archive using accession number PRJNA848233. Code used in this work was deposited at github.com/ildargv/Gainetdinov_et_al_2023; mouse genome sequence and annotation (build mm10/GRCm38.92) were downloaded from https://ftp.ensembl.org/pub/release-92/fasta/mus_musculus/dna/ and https://ftp.ensembl.org/pub/release-92/gtf/mus_musculus/; transposon consensus sequences were obtained from Repbase (v27.02; https://www.girinst.org/repbase/).

## Human research participants

Policy information about studies involving human research participants and Sex and Gender in Research.

| Reporting on sex and gender | N/A |
|---|---|
| Population characteristics | N/A |
| Recruitment | N/A |
| Ethics oversight | N/A |

Note that full information on the approval of the study protocol must also be provided in the manuscript.

# Field-specific reporting

Please select the one below that is the best fit for your research. If you are not sure, read the appropriate sections before making your selection.

☒ Life sciences  ☐ Behavioural & social sciences  ☐ Ecological, evolutionary & environmental sciences

For a reference copy of the document with all sections, see nature.com/documents/nr-reporting-summary-flat.pdf

# Life sciences study design

All studies must disclose on these points even when the disclosure is negative.

| Sample size | No statistical method was used to determine the sample size. For biological samples, the maximum possible sample size (n = 4–12) was used for each type of data, ensuring that variability arising from all accountable sources was incorporated in the analyses (animal, day of data collection, reagent lots). For biochemical experiments, sample size was n = 3 to ensure reproducibility, i.e., for effect sizes of >2-fold, Relative Standard Deviation was <50% for >90% of data. |
|---|---|
| Data exclusions | No data were excluded from the analyses. |
| Replication | All data were collected during independent trials conducted on separate days. When using several types of data for analyses, all possible permutations of samples were analyzed (e.g., 4 control × 4 mutant data sets produced 16 permutations). All attempts at replication were successful. |
| Randomization | This study did not involve treatment or exposure of animals to any agent. Instead, the goal of this work was to compare untreated wild-type mice and untreated mutant mice lacking piRNAs from four genomic loci: all wild-type animals were compared to all mutant mice. Therefore, randomization is not relevant to this study. |
| Blinding | Blinding is not relevant to our study, because during analyses wild-type control and mutant data sets are easily identified. Blinding was not performed during data acquisition and/or analysis. |

# Reporting for specific materials, systems and methods

We require information from authors about some types of materials, experimental systems and methods used in many studies. Here, indicate whether each material, system or method listed is relevant to your study. If you are not sure if a list item applies to your research, read the appropriate section before selecting a response.

## Materials & experimental systems

| n/a | Involved in the study |
|---|---|
| ☐ | ☒ Antibodies |
| ☐ | ☒ Eukaryotic cell lines |
| ☒ | ☐ Palaeontology and archaeology |
| ☐ | ☒ Animals and other organisms |
| ☒ | ☐ Clinical data |
| ☒ | ☐ Dual use research of concern |

## Methods

| n/a | Involved in the study |
|---|---|
| ☒ | ☐ ChIP-seq |
| ☐ | ☒ Flow cytometry |
| ☒ | ☐ MRI-based neuroimaging |

# Antibodies

| Antibodies used | Anti-FLAG antibody (M2, Sigma M8823); Anti-SCP3 antibody (Abcam, ab15093); Anti-phospho-Histone H2A.X (Ser139) antibody, clone JBW301 (Millipore, 05-636, clone JBW301); Donkey anti-Mouse IgG (H+L) Highly Cross-Adsorbed Secondary Antibody, Alexa Fluor 594 (ThermoFisher, A-21203); Donkey anti-Rabbit IgG (H+L) Highly Cross-Adsorbed Secondary Antibody, Alexa Fluor 488 (ThermoFisher, A-21206) |
|---|---|
| Validation | Anti-FLAG antibody (https://www.sigmaaldrich.com/content/dam/sigma-aldrich/docs/Sigma/Bulletin/f1804bul.pdf); Anti-SCP3 antibody (https://www.abcam.com/products/primary-antibodies/scp3-antibody-ab15093.pdf); Anti-phospho-Histone H2A.X (Ser139) antibody, clone JBW301 (https://www.emdmillipore.com/US/en/product/Anti-phospho-Histone-H2A.X-Ser139-Antibody-clone-JBW301,MM_NF-05-636#anchor_COA) |

# Eukaryotic cell lines

Policy information about cell lines and Sex and Gender in Research

| Cell line source(s) | HEK293T and Sf9 cells (lab stock) were obtained from ATCC. Primary mouse spermatocytes were from male mice. |
|---|---|
| Authentication | The cell lines were not authenticated; the cell lines were only to produce recombinant proteins. |
| Mycoplasma contamination | Not tested. |
| Commonly misidentified lines (See ICLAC register) | No commonly misidentified cell lines were used in the study. |

# Animals and other research organisms

Policy information about studies involving animals; ARRIVE guidelines recommended for reporting animal research, and Sex and Gender in Research

| Laboratory animals | C57BL/6 wild-type and mutant adult male mice. |
|---|---|
| Wild animals | The study did not involve wild animals. |
| Reporting on sex | Only males have testes. |
| Field-collected samples | No field-collected samples were used in the study. |
| Ethics oversight | (1) PI on IACUC protocol: Phillip D. Zamore<br>(2) Name of IACUC: UMass Medical School Institutional Animal Care and Use Committee<br>(3) IACUC Docket: A2222-17, "Investigation of mechanisms of small RNA function in vivo" |

Note that full information on the approval of the study protocol must also be provided in the manuscript.

# Flow Cytometry

## Plots

Confirm that:

☒ The axis labels state the marker and fluorochrome used (e.g. CD4-FITC).

☒ The axis scales are clearly visible. Include numbers along axes only for bottom left plot of group (a 'group' is an analysis of identical markers).

☒ All plots are contour plots with outliers or pseudocolor plots.

☒ A numerical value for number of cells or percentage (with statistics) is provided.

## Methodology

**Sample preparation**

Testes of 2–6-month-old mice were isolated, decapsulated, and incubated for 15 min at 33°C in 1× Gey's Balanced Salt Solution (GBSS, Sigma, G9779) containing 0.4 mg/ml collagenase type 4 (Worthington, LS004188) rotating at 150 rpm. Seminiferous tubules were then washed twice with 1× GBSS and incubated for 15 min at 33°C in 1× GBSS with 0.5 mg/ml Trypsin and 1 µg/ml DNase I, rotating at 150 rpm. Next, tubules were homogenized by pipetting through a glass Pasteur pipette for 3 min at 4°C. Fetal bovine serum (FBS; 7.5% f.c., v/v) was added to inactivate trypsin, and the cell suspension was then strained through a pre-wetted 70 µm cell strainer (ThermoFisher, 22363548); cells were collected by centrifugation at 300 × g for 10 min. The supernatant was removed, cells resuspended in 1× GBSS containing 5% (v/v) FBS, 1 µg/ml DNase I, and 5 µg/ml Hoechst 33342 (ThermoFisher, 62249) and rotated at 150 rpm for 45 min at 33ºC. Propidium iodide (0.2 µg/ml, f.c.; ThermoFisher, P3566) was added, and cells strained through a pre-wetted 40 µm cell strainer (ThermoFisher, 22363547).

**Instrument**

FACSAria II Cell Sorter (BD Biosciences; UMass Medical School FACS Core)

**Software**

BD FACSDiva (v9.0)

**Cell population abundance**

Spermatogonia: ~100,000 cells/animal; ~95–100% pure with ≤ 5% pre-leptotene spermatocytes;
Primary spermatocytes: ~1,000,000 cells/animal; ~10–15% leptotene/zygotene spermatocytes, ~45–50% pachytene spermatocytes, ~35–40% diplotene spermatocytes;
Secondary spermatocytes: ~1,000,000 cells/animal; ~100%;
Round spermatids: ~1,500,000 cells/animal; ~95–100%, ≤ 5% elongated spermatids.

**Gating strategy**

The gating strategy used to sort mouse primary germ cells is detailed in Supplementary Figure 5. Briefly, propidium iodide was used to label dead cells (top left panel in Supplementary Figure 5), forward and side scatter were used to isolate single cells (two top middle panels in Supplementary Figure 5), Hoechst 33342 emission in 450/50 and 670/50 bandpass filters was used to separate spermatogonia, spermatocytes, and spermatids (bottom left panel in Supplementary Figure 5). Forward scatter was then used to isolate round spermatids form the mixed population of round and elongated spermatids top right panel in Supplementary Figure 5. The percentages for each subpopulation are shown in the bottom right panel in Supplementary Figure 5.

☒ Tick this box to confirm that a figure exemplifying the gating strategy is provided in the Supplementary Information.

