## [Peer Review File · Nature]

Manuscript Title: Relaxed targeting rules help PIWI proteins silence transposons

Reviewer Comments & Author Rebuttals

Reviewer Reports on the Initial Version:

Referees' comments:

Referee #1 (Remarks to the Author):

Gainetdinov et al. investigate the targeting rules of mouse PIWI-piRNA complexes. The authors observe relaxed pairing rules with extensive pairing (more than just seed-pairing) and mismatch tolerance at different positions, in line with previous observations for worm and fungal PIWI-piRNA silencing complexes (Shen et al., and Mello, Cell 2018; Zhang et al., and Lee, Science 2018; Anzelon et al., and MacRae, Nature 2021). Surprisingly, the authors observe mismatch tolerance across t10/11 and suggest that the PIWI co-factor GTSF facilitates slicing despite mismatch t10/11. This hypothesis should be experimentally tested (see points below). Combining in vitro bind/cleave-n-seq experiments (McGeary et al., and Bartel, Science 2019; Lambert et al., and Burge, Mol Cell 2014), in vivo degradom-seq data, and computational modeling, the authors suggest that PIWI-piRNA complex identify their targets with diverse mismatch tolerance and conclude that piRNA abundance (piRNA concentration) and the total predicted binding energy (~total number of paired nts) positively correlate with target silencing, in line with previous observations (Genzor et al., and Hasse, Genome Res 2021). Finally, the authors speculate that the difference in mismatch tolerance between AGOs and PIWIs might prevent transposons from developing resistant mutants (that evade regulation by piRNAs).

Additional PIWI- and AGO-clade proteins should be tested to evaluate whether the observed differences are representative of the two Argonaute clades -as suggested in the main conclusion-. Mouse AGO2 might not be the best example to contrast PIWIs. AGOs (RNAi) efficiently controls transposons in mouse oocytes (Flemer et al., and Svoboda, Cell 2013). Similarly, MIWI might not be the best example to test transposon regulation, because it does not participate in transposon regulation in vivo (Wu et al., and Zamore, Nat genet 2021; Choi et al., and Dean, PloS Genet 2021). For bind and cleave-n-seq experiments, additional replicates (n>=3) should be added to test the significance of the observed differences (McGeary et al., and Bartel, Science 2019; Lambert et al., and Burge, Mol Cell 2014).

There in an ex vivo system for piRNA-guided post-transcriptional regulation in silkworm to directly test the authors' hypotheses using reporter assays (Izumi et al., and Tomari, Cell 2016).

Major points:

1. Mismatch tolerance in target binding and slicing:

This idea is not novel, but interesting to test for different model organisms (Shen et al., and Mello, Cell 2018; Zhang et al., and Lee, Science 2018; Anzelon et al., and MacRae, Nature 2021). However, the authors should test additional Argonaute proteins in their in vitro assays (bind-n-seq and cut-n-seq). This is particularly important, because MIWI is not involved in transposon silencing in vivo, and is thus not the best example to test the authors' hypothesis on transposon biology. Drosophila

Aubergine and Argonaute-3, and Bombyx Siwi and BmAgo3 are better suited as they are well established in slicing of transposon transcripts in vivo. Binding and cleavage preference of these PIWI proteins will be helpful to understand conserved mismatch flexibilities.

The Drosophila Ago proteins AGO2 and AGO1 would be good controls for AGO-clade Argonaute proteins. They are specialized in siRNA-guided target-slicing and miRNA (non-slicing) respectively.

2. The importance of the GTSF-cofactor for mismatch tolerance

The hypothesis that GTSF1 enables slicing without base-pairing across t10/11 is intriguing and should be tested: mismatch tolerance of different PIWI proteins should be tested with and without GTSF1 in vitro. In an orthogonal approach, reporter assays could be performed in the existing ex vivo model for piRNA-guided slicing (BmN4). If the authors' hypothesis is true, PIWIs should only tolerate mismatches at t10/11 in the presence of GTSF.

Drosophila Piwi would be a good control because it does not rely on its slicer activity for function in vivo, but still associates with GTSF in vivo (Donertas et al., and Brennecke, Genes Dev 2013).

Was GTSF also added to the AGO experiments? If so, is there any effect? If not, it should be tested for comparable experimental conditions.

3. Experimental robustness and statistical significance

All experiments should be performed in triplicates to allow for appropriate statistical analyses. It is hard to know whether the differences described in Fig.1 (for example Fig.1c g2-8) are statistically significant because they are only based on two replicates with rather extensive variability. There is really no reason for not showing three replicates for these in vitro assays.

4. Interpretation of Degradom seq data

The authors calculation of slicing kinetics based on sequencing of degradation intermediates from mouse testes (Degradom-seq; Fig. 3 and 4) rely on the assumption that degradation kinetics are the same for all targets and constant over time. Is there any evidence?

Minor points:

Fig.1: It looks like L1MC is used as second guide RNA in this in vitro assay. Why is it only used with MIWI? Do the differences in KD measurements between MIWI-piRNA#1 and MIWI L1MC piRNA indicate additional sequence specific effects in different positions?

Fig. 1b could be moved to the supplement.

Fig. 2b: The authors should show quartiles and statistics in addition to the median kcut. A box plot would be a more traditional depiction. 2C could be moved to the supplement.

Fig. 3. Different in vivo degradation rates of different cleavage fragments make the interpretation of this experiment difficult. The authors could complement it with direct reporter assays in BmN4 cells (a well-established ex vivo system for piRNA-guided slicing).

Fig.4b: Additional AGO-clade and PIWI-clade proteins should be tested (see major points). Are models for MIWI and AGO2 representative of the entire subfamilies?

The code for all computational experiments should be provided in the supplement.

Referee #2 (Remarks to the Author):

Animals express two separate types of Argonaute proteins, AGOs and PIWIs. Whereas AGO proteins utilize siRNAs or miRNA (21-22 nt long) to repress target mRNAs, PIWI proteins have a specialized, yet essential, function in counteracting transposable elements in the germline of most animals.

Gainetdinov et al. systematically analyze the target RNA binding and cleavage rates of two mouse PIWI proteins, MILI and MIWI. They do so for thousands of RNAs in vitro and in vivo. Moreover, the authors then compare their results to human Argonaute-2, a bona fide member of the AGO clade of Argonaute proteins. The authors demonstrate that, in defying the rules that were established for human Argonaute-2 (hAgo2), the mouse PIWI proteins obey relaxed rules of piRNA:target RNA complementarity for catalyzing target RNA cleavage. In more detail, the authors uncover that PIWI proteins do not rely on 5'-seed:target pairing for selecting RNA targets. Instead, guide:target mismatches (even across the scissile phosphate at target nucleotides 10-11) are tolerated at literally any position across the piRNA:target RNA duplex, as long as at least 16 contiguously paired nucleotides are formed. Importantly, these relaxed targeting rules allow the authors to rationalize the importance of PIWI proteins for fighting their ever-mutating targets, transposable elements. Last, the authors also offer a rationale why 16 contiguous base pairs do not trigger significant off-target cleavage events that would have detrimental effects on mRNA surveillance in PIWI-expressing cells.

I congratulate the authors on their exciting data coupled to a convincing and insightful interpretation. It was a pleasure to go through their story. Because of its outstanding technical quality and the intriguing new insights, which will be relevant to a broad scientific audience, I strongly recommend publication of this manuscript in Nature after my few concerns have been adequately addressed.

Major points:

1. Given the reliance of the authors' entire story on highly active and well-behaved protein preparations, I urge the authors to include quality control measures to convince the reader of the high quality of their protein preparation, e.g SDS gels etc. in the extended data section. In my view this is important as PIWI proteins are known to be prone to precipitation and aggregation. The manuscript would also profit from knowing what percentage of purified PIWI is actually active in target RNA binding and cleavage.
2. Fig. 3c: I was struck by the significant difference between g2-g18 and g2-g19. To me the visible drop in cleaved fraction when incorporating mismatches between nucleotides 2 and 9 looks fairly reminiscent of what I would expect for hAgo2. Can the authors please explain this drop and the difference between g2-g19 and g2-g18?
3. Whereas the manuscript greatly profits from the close comparison of the authors' PIWI data with data on hAgo2 in Figs. 1+2, I do miss such comparative analysis for the logistic regression classifier model in Fig. 4. Here, I ask the authors to compute such model for hAgo2 as well, which would, once more, highlight the important functional differences between AGO and PIWI proteins despite their - on protein level - astounding similarity.

Minor points:

1. Page 4/5: It remained unclear to me from where the hAgo2 data for comparison were taken? Please add this information both to the text and to all appropriate figure legends. In case they were experimentally added for this manuscript, please describe them in the Methods section.
2. Fig. 1 and accompanying text: I was missing information regarding RNA cleavage by the studied PIWI proteins. Was this activity somehow suppressed, e.g by EDTA? Please add this information to

the manuscript. I am aware of the fact that GTSF1 was not added here, but nonetheless slicing will happen to some degree, which could hamper data interpretation.

3. P.10: Please add information regarding GTSF1 expression in primary spermatocytes. Which GTSF1 variant is expressed there? What is the evidence of the protein being present there?

4. Fig. 3e: Please name all bins and the area of abundance they encompass. If this is confusing in the figure directly, please incorporate into the figure legend.

5. Page 16: The authors note that for $k > 16$ fewer than 3% of mRNAs shared a k-mer with a transposon sequence. However, how does that value develop for values $k < 16$? Is there a sudden drop e.g. at $k = 15$?

6. As mentioned in major point #1 for MILI and MIWI, please also convince the reader of the quality of the recombinant GTSF1 protein purification.

Referee #3 (Remarks to the Author):

In this study, Zamore and colleagues explore the sequence complementarity requirements for target recognition and cleavage by the piRNA associated MIWI and MILI proteins in mice. Through an elegant and exhaustive analysis using complementary biochemical and in vivo approaches they identify that target cleavage by both MIWI and MILI is surprisingly permissive to mismatches between the piRNA and its target, even at the exact bases where cleavage occurs. The results change are fundamental understanding of how Piwi class proteins recognize and silence their targets relative to their sibling Argonautes and will have important implications in our understanding how piRNAs and siRNAs function in genome defense. Indeed, the authors demonstrate that piRNAs possess greater capacity than siRNAs to silence transposable elements because of relaxed complementarity requirements for target cleavage, without compromising their ability to distinguish and avoid silencing self genes.

-Tai Montgomery

Comments

This is a rigorous study with sound conclusions supported by a combination of sophisticated and superbly executed biochemical and in vivo experiments.

I had some concerns initially when reading through the in vitro results because of the limited number of piRNA sequences analyzed and the variability in their effectiveness in the different target contexts but these were alleviated by the more exhaustive in vivo data that followed.

Suggestions (all issues noted are minor, I saw no need for additional experiments)

Figure 1. It's clear what the reported pairing includes, but it wasn't clear what it excludes. For example, does g2-g10 mean that there is no complementarity or no extended complementarity outside of that region? Perhaps it's irrelevant given how much sequence depth you have but I was

curious if in these assays 9 nt is sufficient for target recognition or if it means 9 nt of continuous complementarity as well as maybe 25% on average additional scattered random complementarity.

Figure 3a. How many piRNA-target pairs are included in this assessment? Also, what are these targets? It seems like this would be a useful resource for follow up studies identifying the functional roles of pachytene piRNAs.

Figure 3c. It's interesting that both g2-g18 and g2-g19 seem to weakly favor mismatches around the cleavage site but mismatches in the 5' and 3' end regions have a seemingly substantial negative impact on cleavage, particularly for g2-g19, which differs somewhat from the in vitro results. It's surprising to me also that targets representing every possible mononucleotide mismatch exist but presumably that's due to the sheer abundance of piRNA sequences as noted in Extended Data Figure 7c. In Wu et al I believe only 6 mRNAs were identified as cleaved targets of pi6 piRNAs, which are I assume at similar abundance to those produced from the loci you examine here. It would be useful to know for each mismatch how many unique target-piRNA pairs (n=?) are included in the analysis to help put these results in context.

Figure 3d. Does MIWI still cleave between g10-11 when basepairing doesn't begin until g11-13?

Gainetdinov et al.
Nature Manuscript 2022-08-12673A

Responses to Reviewers' Critiques

We thank the Reviewers for their useful comments and suggestions that have substantially improved our manuscript. In particular, we are grateful for the suggestion that we examine the sequence requirements for target cleavage by invertebrate PIWI proteins. These experiments, performed using sponge Piwi and GTSF1, revealed that the remarkable tolerance to mismatches observed for target cleavage by mammalian PIWI proteins was likely present in the PIWI protein present in the last common ancestor of all animals >900 million years ago. Experiments suggested by the Reviewers also helped uncover additional mechanistic details of PIWI tolerance for mismatches. Altogether, our revised manuscript adds >350 new sequencing datasets.

Referee #1

Gainetdinov et al. investigate the targeting rules of mouse PIWI-piRNA complexes. The authors observe relaxed pairing rules with extensive pairing (more than just seed-pairing) and mismatch tolerance at different positions, in line with previous observations for worm and fungal PIWI-piRNA silencing complexes (Shen et al., and Mello, *Cell* 2018; Zhang et al., and Lee, *Science* 2018; Anzelon et al., and MacRae, *Nature* 2021).

Anzelon et al. did not systematically investigate where mismatches are tolerated; instead, they looked at “how mismatches towards the piRNA 3' end influence cleavage” and found that 1–2 mismatches are tolerated by the sponge PIWI. Worm PRG-1 does not slice (Lee et al., *Cell* 2012), and its binding to targets was reported to require a canonical seed match (Shen et al., *Cell* 2018; Zhang et al., *Science* 2018). In contrast, we found that, both in vitro and in vivo, highly abundant mouse piRNAs direct binding and slicing of targets lacking complementary to a canonical seed.

Surprisingly, the authors observe mismatch tolerance across t10/11 and suggest that the PIWI co-factor GTSF facilitates slicing despite mismatch t10/11. This hypothesis should be experimentally tested (see points below). Combining in vitro bind/cleave-n-seq experiments (McGeary et al., and Bartel, *Science* 2019; Lambert et al., and Burge, *Mol Cell* 2014), in vivo degradome-seq data, and computational modeling, the authors suggest that PIWI-piRNA complex identify their targets with diverse mismatch tolerance and conclude that piRNA abundance (piRNA concentration) and the total predicted binding energy (~total number of paired nts) positively correlate with target silencing, in line with previous observations (Genzor et al., and Hasse, *Genome Res* 2021). Finally, the authors speculate that the difference in mismatch tolerance between AGOs and PIWIs might prevent transposons from developing resistant mutants (that evade regulation by piRNAs).

Additional PIWI- and AGO-clade proteins should be tested to evaluate whether the observed differences are representative of the two Argonaute clades -as suggested in the main conclusion-. Mouse AGO2 might not be the best example to contrast PIWIs.

AGOs (RNAi) efficiently controls transposons in mouse oocytes (Flemr et al., and Svoboda, *Cell* 2013).

We respectfully disagree. AGO2 protein sequences are nearly identical among mammals. Golden hamster AGO2 does not participate in transposon silencing and differs from its mouse ortholog at just three of its 860 amino acids: two Thr→Ser and one Ala→Gly substitution in its N-terminal sequence. Human AGO2 differs from its mouse ortholog by only seven amino acids. Target binding preferences of human and mouse AGO2 proteins are virtually indistinguishable (Wee et al., *Cell* 2012; Schirle et al., *Science* 2014; Salomon et al., *Cell* 2015; Sheu-Gruttadauria et al., *EMBO J* 2019; McGeary et al., *Science* 2019; Jouravleva et al., *Cell Rep Methods* 2022). Mouse RNAi participates in transposon silencing in oocytes because of a specialized isoform of Dicer, not AGO2 (Flemr et al., *Cell* 2013). Moreover, the RNAi and piRNA pathways have overlapping but distinct sets of transposon targets in mouse oocytes (Taborska et al., *PLoS Genet* 2019).

Similarly, MIWI might not be the best example to test transposon regulation, because it does not participate in transposon regulation in vivo (Wu et al., and Zamore, *Nat genet* 2021; Choi et al., and Dean, *PLoS Genet* 2021).

MIWI is, in fact, required for transposon regulation in vivo, using piRNA guides not derived from pachytene piRNA loci (Reuter et al., MIWI catalysis is required for piRNA amplification-independent LINE1 transposon silencing. *Nature* 2011). Wu et al. and Choi et al. showed that pachytene piRNAs derived from genomic loci conserved across placental mammals (Ozata et al., *Nat Ecol Evol* 2020) are not required for transposon silencing. However, pachytene piRNAs co-exist with transposon-silencing piRNAs derived from mobile elements, and these piRNAs direct MIWI to silence active LINE1 elements in the mouse genome during meiosis (Reuter et al., *Nature* 2011). Transposon silencing by catalytically active MIWI is required in vivo in mice, underscoring the importance of the relaxed targeting rules for MIWI in protecting the male germ cell genome from transposon expression.

In addition to MIWI, all of our in vitro experiments were also performed for MILI. Catalytically active MILI is required to silence LINE1 transposons in fetal male germ cells and postnatal spermatocytes and spermatids (De Fazio et al., The endonuclease activity of Mili fuels piRNA amplification that silences LINE1 elements. *Nature* 2011; Di Giacomo et al., *Mol Cell* 2013).

To make evolutionary comparisons, we performed RBNS and CNS for the freshwater sponge PIWI protein EfPiwi. Remarkably, despite >900 million years of independent evolution, EfPiwi follows target binding and target slicing rules similar to those of mouse PIWI proteins: (1) EfPiwi binds extensively complementary sites containing or lacking a canonical seed match equally well (Extended Data Fig. 2d), and (2) RNA cleavage by EfPiwi tolerates mismatches at any target nucleotide, including those flanking the scissile bond (Fig. 2b).

For bind and cleave-n-seq experiments, additional replicates ($n \geq 3$) should be added to test the significance of the observed differences (McGeary et al., and Bartel, *Science* 2019; Lambert et al., and Burge, *Mol Cell* 2014).

We have added an additional replicate (i.e., experiments previously performed in duplicate have now been performed independently three times) to both our RBNS and CNS experiments. To the best of our knowledge, this is the first time RBNS analysis has been performed in triplicate: Lambert et al. performed each RBNS experiment once; McGeary et al. also performed a single trial of RBNS for let-7a and lsy-6, and, for miR-124 and miR-7, two technical trials were merged for further analyses.

There in an ex vivo system for piRNA-guided post-transcriptional regulation in silkworm to directly test the authors' hypotheses using reporter assays (Izumi et al., and Tomari, *Cell* 2016).

Our in vivo data from FACS-purified mouse primary spermatocytes provide direct evidence for the remarkable tolerance of MIWI and MILI to mismatches with their cleavage targets. It is difficult to see what additional information would be gleaned using an exogenous reporter assay in an immortalized cell line.

Major points:

1. Mismatch tolerance in target binding and slicing:

This idea is not novel, but interesting to test for different model organisms (Shen et al., and Mello, *Cell* 2018; Zhang et al., and Lee, *Science* 2018; Anzelon et al., and MacRae, *Nature* 2021).

As noted above, we find that both mammalian and sponge PIWI proteins behave quite differently from worm PRG-1, which requires a canonical seed match to bind its targets (Shen et al., and Mello, *Cell* 2018; Zhang et al., and Lee, *Science* 2018). To the best of our knowledge, no PIWI protein has previously been reported to efficiently cleave its targets in the absence of a seed match. We also note that Anzelon et al. investigated "how mismatches towards the piRNA 3' end influence cleavage" and not the efficiency of target slicing in the absence of a seed match or in the presence of dinucleotide mismatches flanking the scissile phosphate.

However, the authors should test additional Argonaute proteins in their in vitro assays (bind-n-seq and cut-n-seq). This is particularly important, because MIWI is not involved in transposon silencing in vivo, and is thus not the best example to test the authors' hypothesis on transposon biology.

To the contrary, catalytically active MIWI is essential for transposon silencing in vivo in mice (Reuter et al., MIWI catalysis is required for piRNA amplification-independent LINE1 transposon silencing. *Nature* 2011). Please also see above.

Drosophila Aubergine and Argonaute-3, and *Bombyx* Siwi and BmAgo3 are better suited as they are well established in slicing of transposon transcripts in vivo. Binding

and cleavage preference of these PIWI proteins will be helpful to understand conserved mismatch flexibilities.

We are unaware of any laboratory that has produced recombinant Aub, Ago3, or BmAgo3 that can be loaded with a defined piRNA guide. Despite multiple attempts over the past 20 years, we have been unable to produce recombinant fly Piwi, Aub, or Ago3. Nonetheless, the Reviewer asks an important question: how deeply conserved are the properties of mouse MIWI and MILI among animals? To address this, we performed RBNS and CNS experiments for the freshwater sponge PIWI protein EfPiwi. These new data show that the binding and target cleavage preferences of PIWI proteins have been maintained through >900 million years of animal evolution.

The *Drosophila* Ago proteins AGO2 and AGO1 would be good controls for AGO-clade Argonaute proteins. They are specialized in siRNA-guided target-slicing and miRNA (non-slicing) respectively.

The tolerance of *Drosophila* Ago2 for mismatches between an siRNA guide and its target RNA has been studied in detail (Wee et al., *Cell* 2012) and are essentially indistinguishable from those of mouse AGO2 (by co-localization single-molecule spectroscopy: Salomon et al., *Cell* 2015; by Cleave-'n-Seq: Becker et al., *Mol Cell* 2019). Target binding by mouse or human AGO2 guided by miRNAs has been examined in detail by multiple methods (by co-localization single-molecule spectroscopy: Salomon et al., *Cell* 2015; by massively parallel ensemble methods: Becker et al., *Molecular Cell* 2019; by RBNS: McGeary et al., *Science* 2019). We note that fly Ago1 is the direct homolog of mouse AGO2.

2. The importance of the GTSF-cofactor for mismatch tolerance

The hypothesis that GTSF1 enables slicing without base-pairing across t10/t11 is intriguing and should be tested: mismatch tolerance of different PIWI proteins should be tested with and without GTSF1 in vitro. In an orthogonal approach, reporter assays could be performed in the existing ex vivo model for piRNA-guided slicing (BmN4). If the authors' hypothesis is true, PIWIs should only tolerate mismatches at t10/t11 in the presence of GTSF.

In our original manuscript, we did not propose that GTSF1 is required for PIWI protein tolerance to mismatches at t10/t11. To the contrary, without GTSF1, even perfectly complementary targets are sliced very slowly (Arif et al., *Nature* 2022), and GTSF1 is only required to accelerate the otherwise slow PIWI cleavage. Our hypothesis is that intrinsic differences between the catalytic centers of AGO and PIWI proteins underlie the remarkable tolerance of PIWI catalysis for unpaired target nucleotides. We have revised the corresponding part of the manuscript to clarify our model.

The results of the experiments requested by the Reviewer provide strong support for our proposal that mismatch tolerance is an intrinsic property of PIWI proteins and is not a consequence of GTSF1 function. The new CNS experiments for MILI, MIWI, and EfPiwi in the presence and the absence of GTSF1 are presented in Extended Data Figs. 9a and 9b. For all three PIWI

proteins, GTSF1 accelerated the cleavage of fully complementary and mismatched targets to a similar extent. We conclude that tolerance for guide:target mismatches is an intrinsic property of PIWI proteins unaffected by GTSF1.

Drosophila Piwi would be a good control because it does not rely on its slicer activity for function in vivo, but still associates with GTSF in vivo (Donertas et al., and Brennecke, Genes Dev 2013).

We and others have not been able to produce *Drosophila* Piwi, Aub, or Ago3 as recombinant proteins that can be loaded with a defined piRNA guide and do not think that revisiting these failed experiments would be productive.

Was GTSF also added to the AGO experiments? If so, is there any effect? If not, it should be tested for comparable experimental conditions.

GTSF1 has no effect on the binding or rate of cleavage by AGO2 (Arif et al., *Nature* 2022). We have now performed CNS experiments for mouse AGO2 in the presence and absence of mouse GTSF1 (Extended Data Fig. 9a). Our results confirm the originally published observation that GTSF1 does not alter the binding affinity or rate of target cleavage by AGO2.

3. Experimental robustness and statistical significance

All experiments should be performed in triplicates to allow for appropriate statistical analyses. It is hard to know whether the differences described in Fig.1 (for example Fig.1c g2-8) are statistically significant because they are only based on two replicates with rather extensive variability. There is really no reason for not showing three replicates for these in vitro assays.

As we note above, we have added an additional replicate (i.e., $n = 3$) for all RBNS and CNS assays. Figure 1 now displays mean \pm SD for all RBNS data.

4. Interpretation of Degradome seq data

The authors calculation of slicing kinetics based on sequencing of degradation intermediates from mouse testes (Degradome-seq; Fig. 3 and 4) rely on the assumption that degradation kinetics are the same for all targets and constant over time. Is there any evidence?

Our analysis does not calculate the in vivo kinetics of slicing, for the exact reason pointed out by the Reviewer. Instead, we use genetics only to identify cleavage sites that were likely (1) cleaved or (2) not cleaved; we thus estimate the fraction of cleaved sites for each piRNA:target pairing pattern. We require that for a degradome read to be considered the cleavage product of piRNA-directed slicing, its abundance must be reduced at least 8-fold when the corresponding piRNA is removed genetically. Our analytical method is therefore unaffected by differences in stability among cleavage products because the abundance of each cleaved target is compared only to itself in wild type vs. mutant.

Minor points:

Fig.1: It looks like L1MC is used as second guide RNA in this in vitro assay. Why is it only used with MIWI?

Our revised manuscript now includes three independent trials of RBNS experiments for L1MC-guided MILI. As seen from RBNS data for piRNA #1, MIWI and MILI have essentially indistinguishable binding preferences when programmed with the same guide.

Do the differences in KD measurements between MIWI-piRNA#1 and MIWI L1MC piRNA indicate additional sequence specific effects in different positions?

We used piRNA #1 (4GC/3AU seed) and L1MC piRNA (1GC/6AU seed) to compare how the GC-content of the seed sequence influences the target-binding affinity of PIWI. The binding affinity of MILI, MIWI, and EfPiwi all correlated linearly with predicted base pairing energy (Fig. 1c, right), suggesting that all positions of the piRNA seed behave similarly.

Fig. 1b could be moved to the supplement.

We would prefer to leave it in the main text to provide readers with a direct summary of the numerical results.

Fig. 2b: The authors should show quartiles and statistics in addition to the median kcut. A box plot would be a more traditional depiction. 2C could be moved to the supplement.

We now provide the interquartile range in Figure 2b and the results of statistical tests in Extended Data Fig. 5b. We note that, as in the original manuscript, all data points shown with boxplots are in Extended Data Fig. 5a. We prefer to keep Fig. 2c in the main text as it directly compares the slicing preferences of the three PIWI proteins used in this study.

Fig. 3. Different in vivo degradation rates of different cleavage fragments make the interpretation of this experiment difficult. The authors could complement it with direct reporter assays in BmN4 cells (a well-established ex vivo system for piRNA-guided slicing).

Our analysis is unaffected by in vivo degradation rates of individual cleavage products because we measured the fraction of cleavage products explained by each possible pairing configuration, not the rate of cleavage in vivo. As we describe above, we required degradome fragments to decline at least 8-fold in the absence of the piRNA that directs target cleavage at a particular site; this threshold serves only to filter the data. The stability of a cleavage fragment determines only the depth of sequencing required to detect it. Currently, we use 1 billion reads per data point, which allows us to detect up to one-hundred thousand piRNA:target RNA pairs (Supplementary Table 2).

As we note above, we do not think that reporter assays in an immortalized cell line would provide an experimental test of our in vivo data.

Fig.4b: Additional AGO-clade and PIWI-clade proteins should be tested (see major points). Are models for MIWI and AGO2 representative of the entire subfamilies?

Our new experiments examining EfPiwi demonstrate that the properties of PIWI-clade proteins have been conserved for >900 million years of evolution. Previously published studies (see above) have exhaustively demonstrated that the properties of mammalian AGO2 are conserved to at least the last common ancestor of flies and mice.

The code for all computational experiments should be provided in the supplement.

Access to both our code and raw sequencing data was provided to the Reviewers and will, of course, be made publicly available either in supplemental data or hosted on our lab website as the journal prefers.

Referee #2

Animals express two separate types of Argonaute proteins, AGOs and PIWIs. Whereas AGO proteins utilize siRNAs or miRNA (21-22 nt long) to repress target mRNAs, PIWI proteins have a specialized, yet essential, function in counteracting transposable elements in the germline of most animals.

Gainetdinov et al. systematically analyze the target RNA binding and cleavage rates of two mouse PIWI proteins, MILI and MIWI. They do so for thousands of RNAs in vitro and in vivo. Moreover, the authors then compare their results to human Argonaute-2, a bona fide member of the AGO clade of Argonaute proteins. The authors demonstrate that, in defying the rules that were established for human Argonaute-2 (hAgo2), the mouse PIWI proteins obey relaxed rules of piRNA:target RNA complementarity for catalyzing target RNA cleavage. In more detail, the authors uncover that PIWI proteins do not rely on 5'-seed:target pairing for selecting RNA targets. Instead, guide:target mismatches (even across the scissile phosphate at target nucleotides 10-11) are tolerated at literally any position across the piRNA:target RNA duplex, as long as at least 16 contiguously paired nucleotides are formed. Importantly, these relaxed targeting rules allow the authors to rationalize the importance of PIWI proteins for fighting their ever-mutating targets, transposable elements. Last, the authors also offer a rationale why 16 contiguous base pairs do not trigger significant off-target cleavage events that would have detrimental effects on mRNA surveillance in PIWI-expressing cells.

I congratulate the authors on their exciting data coupled to a convincing and insightful interpretation. It was a pleasure to go through their story. Because of its outstanding technical quality and the intriguing new insights, which will be relevant to a broad scientific audience, I strongly recommend publication of this manuscript in Nature after my few concerns have been adequately addressed.

We thank the Reviewer for their kind words!

Major points:

1. Given the reliance of the authors' entire story on highly active and well-behaved protein preparations, I urge the authors to include quality control measures to convince

the reader of the high quality of their protein preparation, e.g. SDS gels etc. in the extended data section. In my view this is important as PIWI proteins are known to be prone to precipitation and aggregation. The manuscript would also profit from knowing what percentage of purified PIWI is actually active in target RNA binding and cleavage.

Throughout the manuscript, PIWI protein concentration is reported as active concentration determined by pre-steady-state cleavage assays: all biochemical measurements relied only on the concentration of active piRISC. Extended Data Fig. 1c now presents the assays used to measure the concentration of active protein for each piRISC and a detailed description of these methods now appears in the experimental methods (“Determination of Active Fraction of piRISC”). The purity of EfPwi, MIWI, and MILI, determined by SDS-polyacrylamide gel electrophoresis, now appears in Extended Data Fig. 1d. A representative Coomassie-stained SDS-PAGE gel for *EmGtsf1* and mouse GTSF1 is in the new Extended Data Fig. 1e.

2. Fig. 3c: I was struck by the significant difference between g2-g18 and g2-g19. To me the visible drop in cleaved fraction when incorporating mismatches between nucleotides 2 and 9 looks fairly reminiscent of what I would expect for hAgo2. Can the authors please explain this drop and the difference between g2-g19 and g2-g18?

The Reviewer notes an important aspect of the data in Fig. 3c and Extended Data Fig. 11a. These analyses show that cleavage *in vivo* is impacted by mismatches in the piRNA 5' region to a greater extent (i.e., median $f_{cleaved}$ is smaller) compared to mismatches to other piRNA regions:

- for g2–g17 sites,

$$\text{median } f_{cleaved}^{mismatches\ to\ g^3-g^8} = 0.03 \text{ vs. } f_{cleaved}^{mismatches\ to\ g^9-g^{17}} = 0.09;$$

- for g2–g18 sites,

$$\text{median } f_{cleaved}^{mismatches\ to\ g^3-g^8} = 0.10 \text{ vs. } f_{cleaved}^{mismatches\ to\ g^9-g^{18}} = 0.18;$$

- for g2–g20 sites,

$$\text{median } f_{cleaved}^{mismatches\ at\ g^3-g^8} = 0.06 \text{ vs. } f_{cleaved}^{mismatches\ at\ g^9-g^{19}} = 0.21.$$

Because the data in these analyses incorporate slicing by both low (e.g., ~10 pM) and high (> 500 pM) abundance piRNAs, the lower fraction of cleaved sites for targets with mismatches to piRNA 5' sequence likely reflects the slower on-rates for piRNAs with low intracellular concentration. These results agree well with our analyses in Fig. 3e—where we binned data by piRNA concentration—and with the outcome of the logistic regression analysis. The revised manuscript now includes a discussion of these observations in the Results section.

By contrast, Cleave-‘n-Seq experiments were conducted using 1,000 pM PIWI protein. Such high piRISC concentration likely compensates for the lower frequency of productive collisions for targets with mismatches to piRNA 5' sequence.

3. Whereas the manuscript greatly profits from the close comparison of the authors' PIWI data with data on hAgo2 in Figs. 1+2, I do miss such comparative analysis for the logistic regression classifier model in Fig. 4. Here, I ask the authors to compute such model for hAgo2 as well, which would, once more, highlight the important functional differences between AGO and PIWI proteins despite their - on protein level - astounding similarity.

The only model for AGO2 slicing of large numbers of targets using a highly diverse set of siRNA guides is the unusual mechanism of transposon silencing during oogenesis in mice and rats, but not in other mammals. Unfortunately, sequencing and analyzing 5' monophosphorylated RNAs from mouse oocytes is unlikely to provide data usable in logistic regression analyses, because the highly repetitive nature of transposon-derived siRNAs and targets prevents the unambiguous assignment of siRNAs to putative cleavage sites, making it impossible to calculate the fraction of cleaved sites for each guide-target pairing pattern. Our studies of in vivo cleavage by mouse PIWI proteins were made possible by the unique nature of pachytene piRNAs, which are depleted of repetitive sequences. Our logistic regression classifier analysis of in vivo PIWI slicing data, however, included the features that are known to favor binding and slicing by mouse and human AGO2 (e.g., t1 identity, location of target site in transcript; Agarwal et al., *eLife* 2015; Shin et al., *Mol Cell* 2010).

Minor points:

1. Page 4/5: It remained unclear to me from where the hAgo2 data for comparison were taken? Please add this information both to the text and to all appropriate figure legends. In case they were experimentally added for this manuscript, please describe them in the Methods section.

All mouse AGO2 RBNS data are from Jouravleva et al., *Cell Rep Methods* 2022, except for the third trial of let-7a, which was generated for this manuscript. All human AGO2 RBNS data are from McGeary et al., *Science* 2019. Mouse AGO2 CNS data for let-7a and miR-21 are from (Becker et al., *Mol Cell* 2019). We generated mouse AGO2 CNS data for the L1MC guide for the revised manuscript. The corresponding Methods sections were revised to add this information.

2. Fig. 1 and accompanying text: I was missing information regarding RNA cleavage by the studied PIWI proteins. Was this activity somehow suppressed, e.g. by EDTA? Please add this information to the manuscript. I am aware of the fact that GTSF1 was not added here, but nonetheless slicing will happen to some degree, which could hamper data interpretation.

Because piRNA binding by PIWI Argonaute proteins may require divalent cations, we did not suppress target cleavage using EDTA. Instead, we took advantage of (1) the very slow rate of target cleavage in the absence of GTSF1; and (2) the limitation of RBNS that when sequencing input RNA library at ~50 million read-depth, RBNS can only identify binding sites ≤ 12 nt long. Thus, our

RBNS analyses do not interrogate sites that are long enough to be cleaved by piRISC. Our revised Methods now describe this.

3. P.10: Please add information regarding GTSF1 expression in primary spermatocytes. Which GTSF1 variant is expressed there? What is the evidence of the protein being present there?

By immunofluorescence, GTSF1 protein is expressed in primary spermatocytes (Yoshimura et al., *Developmental Biology* 2009), whereas GTSF1I and GTSF2 are detected only in spermatids (Takemoto et al., *PLoS ONE* 2016). RNA-seq detects *Gtsf1* mRNA in primary and secondary spermatocytes and round spermatids (Arif et al., *Nature* 2022). Western blotting using a 3XFLAG-GTSF1 knock-in strain showed that GTSF1 is readily detected in both primary and secondary spermatocytes (Arif et al., *Nature* 2022). We now include this information in the revised manuscript.

4. Fig. 3e: Please name all bins and the area of abundance they encompass. If this is confusing in the figure directly, please incorporate into the figure legend.

Figure 3e was revised to indicate the range of piRNA concentrations for each bin.

5. Page 16: The authors note that for $k > 16$ fewer than 3% of mRNAs shared a k -mer with a transposon sequence. However, how does that value develop for values $k < 16$? Is there a sudden drop e.g. at $k = 15$?

Analyses for $k = [6, 30]$ (Extended Data Fig. 14a) demonstrate that the fraction of shared k -mers declines from ~ 1 to < 0.05 for $8 \leq k \leq 15$.

6. As mentioned in major point #1 for MILI and MIWI, please also convince the reader of the quality of the recombinant GTSF1 protein purification.

A representative image of a Coomassie-stained SDS-PAGE gel of the purified, recombinant *Ephedratia muelleri* and mouse GTSF1 preparations used in this study is now shown in Extended Data Fig. 1e.

Referee #3 (Remarks to the Author):

In this study, Zamore and colleagues explore the sequence complementarity requirements for target recognition and cleavage by the piRNA associated MIWI and MILI proteins in mice. Through an elegant and exhaustive analysis using complementary biochemical and in vivo approaches they identify that target cleavage by both MIWI and MILI is surprisingly permissive to mismatches between the piRNA and its target, even at the exact bases where cleavage occurs. The results change are fundamental understanding of how Piwi class proteins recognize and silence their targets relative to their sibling Argonautes and will have important implications in our understanding how piRNAs and siRNAs function in genome defense. Indeed, the authors demonstrate that piRNAs possess greater capacity than siRNAs to silence transposable elements because of relaxed complementarity requirements for target

cleavage, without compromising their ability to distinguish and avoid silencing self-genes.

Thank you!

Comments

This is a rigorous study with sound conclusions supported by a combination of sophisticated and superbly executed biochemical and in vivo experiments.

Thank you!

I had some concerns initially when reading through the in vitro results because of the limited number of piRNA sequences analyzed and the variability in their effectiveness in the different target contexts but these were alleviated by the more exhaustive in vivo data that followed.

Suggestions (all issues noted are minor, I saw no need for additional experiments)

Figure 1. It's clear what the reported pairing includes, but It wasn't clear what it excludes. For example, does g2-g10 mean that there is no complementarity or no extended complementarity outside of that region? Perhaps it's irrelevant given how much sequence depth you have but I was curious if in these assays 9 nt is sufficient for target recognition or if it means 9 nt of continuous complementarity as well as maybe 25% on average additional scattered random complementarity.

Our revised Methods section now explicitly defines what binding site are and are not included in the analyses:

“The sequencing depth of the input library ($\sim 50 \times 10^6$ reads) allowed measurement of input frequencies for ≤ 12 -nt motifs. To interrogate non-overlapping target sets, each ≤ 10 -nt contiguous binding site was required to be flanked by nucleotides not complementary to the guide: e.g., a g4–g12 contiguous target site did not pair to guide positions g3 and g13. Each 11-nt long contiguously complementary site was required to be flanked by a non-matching nucleotide only at its 5' end: e.g., a g4–g14 contiguous target site did not pair to guide position g3.”

Since ≥ 4 bp are required to form a stable RNA helix, the probability of a fortuitous ≥ 4 -nt stretch of complementarity outside the interrogated sites is ≤ 0.004 ($0.25 \times 0.25 \times 0.25 \times 0.25$).

Figure 3a. How many piRNA-target pairs are included in this assessment? Also, what are these targets? It seems like this would be a useful resource for follow up studies identifying the functional roles of pachytene piRNAs.

New Supplementary Table 2 contains information about the number of target sites, and new Supplementary Data 1 provides the identity of pachytene piRNA target sites used for analyses for Figs 3b, 3c, 3e, and Extended Data Figs. 11a and 12c. One of the peculiar features of the overwhelming majority of pachytene piRNAs is that although they cleave a considerable number of different mRNAs, lncRNAs, and piRNA precursors, they affect the steady-state abundance of a tiny

number of mRNAs. For example, *pi6* produces >3,500 piRNAs predicted to be present in every pachytene spermatocyte. None of these piRNAs are detected in *pi6*^{-/-} mutants, yet the steady-state abundance of only six mRNAs (targeted by seven piRNAs) are significantly altered (Wu et al., *Nature Genetics* 2020). Choi et al. (*PLoS Genetics* 2021) made similar observations for a *pi18*^{-/-} mutant. Our ongoing analysis of the *pi2*^{-/-}; *pi9*^{-/-}; *pi17*^{-/-} triple mutant data concur with the findings published for *pi6*^{-/-} and *pi18*^{-/-} mutants and, when complete, will be published separately.

Figure 3c. It's interesting that both g2-g18 and g2-g19 seem to weakly favor mispairs around the cleavage site but mispairs in the 5' and 3' end regions have a seemingly substantial negative impact on cleavage, particularly for g2-g19, which differs somewhat from the in vitro results.

Please see our response to a similar comment by Reviewer #2 (their Major Point 2). Briefly, data in these analyses incorporate slicing by both low (e.g., ~10 pM) and high (> 500 pM) abundance piRNAs. The smaller fraction of cleaved sites for targets with mismatches to piRNA 5' sequences likely reflects the slower on-rates for piRNAs of low intracellular concentration. These results agree well with our analyses in Fig. 3e—where we binned data by piRNA concentration—and with the outcome of the logistic regression analyses of in vivo cleavage data. The revised manuscript now includes a discussion of this matter in the Results section.

By contrast, Cleave-n-Seq experiments were conducted using 1,000 pM PIWI protein. Such high piRISC concentration likely compensates for the slower on-rate for targets with mismatches to piRNA 5' sequence.

It's surprising to me also that targets representing every possible mononucleotide mismatch exist but presumably that's due to the sheer abundance of piRNA sequences as noted in Extended Data Figure 7c. In Wu et al I believe only 6 mRNAs were identified as cleaved targets of pi6 piRNAs, which are I assume at similar abundance to those produced from the loci you examine here.

Wu et al. identified six mRNAs whose steady-state abundance differed significantly between wild-type and *pi6*^{-/-} mutants. Thousands of RNAs are cleaved by the >3,500 *pi6* piRNAs predicted to be present in every pachytene spermatocyte, yet these cleavage events have no detectable effect on the steady-state abundance of the target mRNAs. This is one of the most puzzling features of pachytene piRNAs: although the overwhelming majority of pachytene piRNAs are molecularly functional, cleaving a large number of different mRNAs, lncRNAs, and piRNA precursors, they affect the steady-state abundance of a tiny number of mRNAs.

It would be useful to know for each mispair how many unique target-piRNA pairs (n=?) are included in the analysis to help put these results in context.

New Supplementary Table 2 and Supplementary Data 1 now contain these data.

Figure 3d. Does MIWI still cleave between g10-11 when base pairing doesn't begin until g11-13?

To answer this question, we conducted a modified Cleave-'n-Seq experiment in which the identity of the scissile phosphodiester bond was established by sequencing 3' cleavage products. (In these experiments, a barcode in the 3' cleavage product allowed us to infer the complete sequence of the cleavage site.) These new data, presented in revised Extended Data Fig. 8c, demonstrate that MIWI cleaves targets between nucleotides t10 and t11 even when both g10 and g11 are unpaired or when contiguous pairing does not start until g11. We thank the Reviewer for encouraging us to do these experiments!

Reviewer Reports on the First Revision:

Referees' comments:

Referee #1 (Remarks to the Author):

Fig. 1: I am still having difficulty understanding the interpretation of Fig. 1. The calculated dissociation constants (Kds) for two individual guide-RNA sequences show a significant difference, which could be attributed to the instability of the assay for PIWIs or different loading of PIWIs with different guide-RNAs in vitro. It is also possible that piRNA-independent interactions are producing unexpected target profiles. Given the variability in Kds for different PIWIs and guide sequences, it seems difficult to generalize for all piRNAs and all PIWIs.

Fig. 2: Does the wide range of cut rates between replicates and individual guide sequences truly support a general conclusion?

Fig. 3: The interpretation of the "degradome seq" data is challenging to understand. The authors assume that all 5' phosphorylated RNA fragments originate from piRNA-guided slicing, but these fragments also include exonucleolytic degradation intermediates. Varying decay intermediates could be misinterpreted as 'relaxed' piRNA targets. The knockout of entire piRNA clusters (pi2/9/17) could indirectly result in varying general decay fragments. The authors' hypothesis of 'relaxed target rules' is based on highly variable in vitro results and correlational sequencing data, with no attempt for direct experimental verification. My suggestion of reporter assays in silkworm cells to test the hypothesis was not well-received by the authors, but some direct orthogonal experiment should be performed.

Fig. 4: Considering the mild changes for mismatches at individual pairing positions and the lack of changes in target RNA levels in vivo (Wu and Zamore, 2020), the biological importance of this research is questionable.

In summary, while this manuscript could be published elsewhere, it does not seem suitable for publication in Nature. There is no direct experimental evidence for 'relaxed targeting-rules' for individual PIWI proteins, and the hypothesis is based on correlation and highly variable in vitro experiments. The generalization that targeting rules might be different for the entire PIWI subfamily compared to all AGOs is purely speculative. The interpretation that 'relaxed targeting-rules' have anything to do with transposon restriction is unfounded, given that both MIWI and mouse AGO2 participate in both transposon and gene silencing in vivo. Furthermore, the authors' comments during revision highlight the lack of biological relevance, indicating that even if there was some more 'relaxed target cleavage' for one or the other piRNA and PIWI protein, it may not be functionally significant.

Referee #2 (Remarks to the Author):

All points I raised before have been adequately addressed by the authors. I thus recommend

publication of this manuscript at this time.

Referee #3 (Remarks to the Author):

Zamore and team further refine what was already an impressive study by demonstrating that target slicing by a distantly related Piwi protein from a sponge is also permissive to mismatches at each position of the target, suggesting that this is an ancient feature of Piwi proteins that distinguishes them from other Argonautes. They also added additional replication and repetition in their experiments making for an exceptionally rigorous study. They thoroughly addressed the reviewers' concerns with additional experiments, figures, and datasets.

(Additional remarks after cross-consulting):

Reviewer 1 continues to have concerns about variation within the experiments and interpretation of the results presented in Figure 1. After considering the reviewer's concerns, the response from the authors, and reviewing the manuscript, I find the critique unsubstantiated. There is indeed considerable variation in the Kds for the two different piRNAs tested but as noted by the authors, this is to be expected, particularly given the large difference in GC content in the piRNA sequences. Furthermore, both piRNAs tested show a similar trend that is reproducible across three experiments and with three different Piwi proteins, all of which behave differently than Ago2. I believe the authors' conclusions are sound.

Additionally, reviewer 1 questions the variability observed in cleavage rates for the various piRNAs, as well as variation among biological replicates and the extent to which the results support the conclusions drawn from the data in Figure 2. The authors show (Figure R1) that the biological variation is modest and consistent with what would be expected in this or any similar biochemical experiment, which I believe should satisfy the concern about biological variation. While there is certainly some variation among the different piRNA guides, there is a clear and consistent trend that supports the conclusions reported in this figure. Conclusion from the data that continuous pairing across positions 9-13 is not as critical for slicing by the Piwis as it is for Ago2 is also supported by statistical analysis in Extended Data Figure 5. However, the data shown in 2a - that extended pairing at the 3' end of the piRNA with the target can compensate for mismatches between position 2-20 - would benefit from statistical analysis.

The reviewer expresses concern with the degradome sequencing data in figure 3, particularly the prevalence of a high background of decay products not associated with piRNA-guided cleavage. It's well known that degradome sequencing is messy so this is to be expected, but the authors apply a high level of stringency in controlling for background fragments, exceeding the level of rigor typically used in degradome sequencing experiments. I'm satisfied with their rebuttal.

The reviewer also believes an additional in vivo system is necessary to support the biochemical data. The authors rebut this criticism by pointing to the mouse genetics data in figures 3 and 4, which I believe is highly complementary support of the in vitro data. The reviewer does not suggest a specific experiment. A piRNA reporter assay in which a piRNA target site can be manipulated to

contain mispairs at specific positions and then tested for the ability of the various Argonautes to cleave it would be valuable, although it would have a more limited scope than the approach taken by the authors. I would support the addition of such an experiment, however, this has already been done and was recently published (Dowling et al), as noted by the authors. And the results were consistent with the authors' conclusions, so I don't see a need for the authors to do a similar experiment. They cite this work in support of their conclusions.

Finally, the reviewer questions the biological relevance of the study, again taking aim at the quality of the data and support for the conclusions. I strongly disagree with the reviewer's overall assessment. This is an incredibly important study that challenges the dogma in the small RNA field that Argonautes require perfect pairing across positions 10-11 for target cleavage and benefit from pairing within a canonical seed sequence for target recognition. The small RNA field is guided by this dogma and thus this manuscript changes the way we think about small RNA-target interactions and their outcomes. While the manuscript does not unequivocally demonstrate the biological importance of the so-called relaxed targeting rules of Piwis, the results of simulations strongly support a model in which Piwi are better adapted than AGO Argonautes for keeping the ever changing transposon landscape in check to protect animal genomes. Overall, I think the results are sound and that the conclusions are supported by multiple lines of evidence from well-controlled and complementary biochemical and genetic experiments. This is certain to be an impactful study that will without a doubt provide a foundation for further studies aimed at untangling the biological roles of piRNAs in gene regulation and transposon silencing.